# Fractionated degradation and valorization of polypropylene waste into sulfonate surfactants

Zhen Xu [1,7] ✉, Yang Zhang[1,7], Tao Wang[2,7], Rong Yang[1], Hao Sun [3], Meiling Chen[4], Feng Liu[5], Jianjun Xu[6], Kai-Jie Chen [1], Qikun Zhang[4] ✉ & Fuping Pan[1] ✉

Tandem upcycling presents a promising approach to converting plastic waste into valuable chemicals. Controlling the chain length distribution (CLD) of hydrocarbons in plastic degradation is crucial for the downstream synthesis of high-quality and high-value functional chemicals. However, controlling the CLD remains a significant challenge due to the high randomness and poor regulation of polymer chain scission. Herein, we introduce a fractionated degradation approach to tune the CLD of hydrocarbons derived from polypropylene (PP) degradation, thereby facilitating effective downstream valorization. By designing a fractionated degradation reactor that manipulates latent heat, we achieve size-selective degradation of PP into $C_6$-$C_{15}$ and $C_{15}$-$C_{28}$ α-alkenes with narrower and more tunable CLDs compared to traditional methods. Modeling and ASPEN simulations reveal a quasi-linear relationship between chain length and fraction temperatures, as well as a positive correlation between CLD and number of fractions. Scale-up experiments with 10 kg of real-life PP waste validate the effectiveness of the fractionated degradation. The PP-derived α-alkenes with narrow CLDs exhibit superior performance in sulfonation, resulting in foaming, detergency, and biodegradability, comparable to commercial sulfonate surfactants. Techno-economic and life cycle analyses demonstrate improved economic and ecological benefits of PP-to-surfactant upcycling over existing petrochemical-based methods.

Global polypropylene (PP) production rose sharply from ~ 62 million tons in 2015 to ~ 80 million tons in 2023 (~ 20% of total plastics; Supplementary Fig. 1a, b). Driven by disposable packaging[1], rapid PP turnover exacerbated plastic waste and GHG emissions (Supplementary Fig. 1c). The recycling of polyolefins remains challenging due to inefficient logistics, severe contamination in plastic waste (Supplementary Fig. 2), and poor profitability, leading to low recycling rate of PP and other plastics[1–4]. Chemical recycling mitigates environmental impacts and valorizing waste[5–8], converting polyolefins into high-grade fuels[9], monomers[7,10], aromatics[11,12], and functionalized hydrocarbons[13,14]. Among these, surfactants stand out for their high market volume and value (Supplementary Fig. 2). Catalytic oxidative

[1]School of Chemistry and Chemical Engineering, Northwestern Polytechnical University, Xi'an, China. [2]Institute of Carbon Neutrality, Tongji University, Shanghai, China. [3]School of Fintech, Dongbei University of Finance and Economics, Dalian, China. [4]Department of Chemistry, Chemical Engineering and Materials Science, Ministry of Education Key Laboratory of Molecular and Nano Probes, Shandong Normal University, Jinan, China. [5]School of Management Science and Engineering, Dongbei University of Finance and Economics, Dalian, China. [6]Institute of Supply Chain Analytics, Dongbei University of Finance and Economics, Dalian, China. [7]These authors contributed equally: Zhen Xu, Yang Zhang, Tao Wang. ✉e-mail: zhen1@nwpu.edu.cn; zhangqk@sdnu.edu.cn; fupingpan@nwpu.edu.cn

degradation, biological fermentation, and tandem strategies can produce synthetic fatty acids from polyolefins[15–18]. However, consumer preference for natural fatty acids restricts synthetic alternatives from body care, food, and medical markets[19], which dominate global demand (> 90 $%)[20].

Synthetic sulfonate surfactants have dominated the surfactant market since the late 1940s[21,22], propelled by their superior performance in hard water. Hydrocarbyl sulfonates currently make up ~ 50% of the surfactant market (~ 10 million tons), which is equivalent to 20% of the annual global production of PP, positioning surfactants as an ideal destination for polyolefin upcycling[23]. Via tandem strategies, PP can theoretically be converted into sulfonates with two steps: thermal degradation followed by sulfonation[24]. Practically, PP-to-sulfonates remains challenging due to limited control over the chain length distribution (CLD). The distinct reactivity of short- and long-chain olefins leads to various side reactions[24], as demonstrated by our exploratory studies using α-alkenes with broad CLD (Supplementary Fig. 3). The sulfonation of α-alkenes with a broad CLD resulted in severe coking, producing low-value products due to their unaesthetic appearance and poor performance. Therefore, tuning the CLD is crucial for controllable manufacturing of surfactant with optimal properties.

As suggested by population balance models[25–27], polymer random scission spontaneously yields a broad CLD until it approaches complete C−C bond cleavage, finally producing short hydrocarbons. Therefore, regulating the random scission events is vital for controlling the CLD; simply put, this means boosting the cleavage of certain hydrocarbons while avoiding the over-degradation of others. Such size-selective degradation can be achieved by regulating diffusion behaviors and phase changes[28]. For instance, larger hydrocarbons are confined in the pores of catalysts (e.g., molecular sieves and metal-organic frameworks)[28–32], promoting further cleavage and selective

releasing of preferred hydrocarbons. However, the resulting hydrocarbons either have a surfactant range ($C_8$–$C_{45}$) with broad CLD or are dominated by light hydrocarbons, with limited applicability in the downstream synthesis of fine chemicals. Moreover, the catalytic approach requires sophisticated catalysts and suffers from durability challenges[33]. Alternatively, reactor engineering can manipulate phase changes for size-selective degradation[13,26,27,34]. Recently, thermal-gradient degradation of polyolefins in non-fraction (NT) reactors was demonstrated, in which the polyolefins are degraded at specific temperatures (Supplementary Fig. 4), followed by complete collection in quenching modules[13]. However, the light hydrocarbons are often mixed with heavy hydrocarbons, resulting in broad CLD (from $C_{10}$ to ~ $C_{90}$), attributed to the inefficiency for size differentiation and separation in the reactors (Supplementary Fig. 4, see the Supplementary Discussion)[13,34].

Hypothetically, regulating randomness requires precise size selectivity and efficient separation, which can potentially be achieved using fractionated devices consisting of degradation, fractionators, and collection units (Fig. 1). The fractionator exploits size-dependent properties (e.g., latent heat, osmotic pressure, viscosity, etc.) to classify and separate products by sizes for further degradation or collection. Among the size-dependent properties, latent heat can be exploited using cost-effective column frameworks. Herein, as a proof-of-concept, fractionated reactors are designed for the size-selective degradation of laboratory-grade PP and real-life PP wastes (Supplementary Fig. 5). The process generates α-alkenes with tunable and narrow CLD (Fig. 2). Subsequent sulfonation valorizes the α-alkenes into light-color surfactants with enhanced foaming, detergent, and biodegrading performance, comparable to commercial sulfonate surfactants. The fractionated degradation demonstrates improved practicability, sustainability, and economic benefits in the upcycling of waste polyolefins into synthetic detergents.

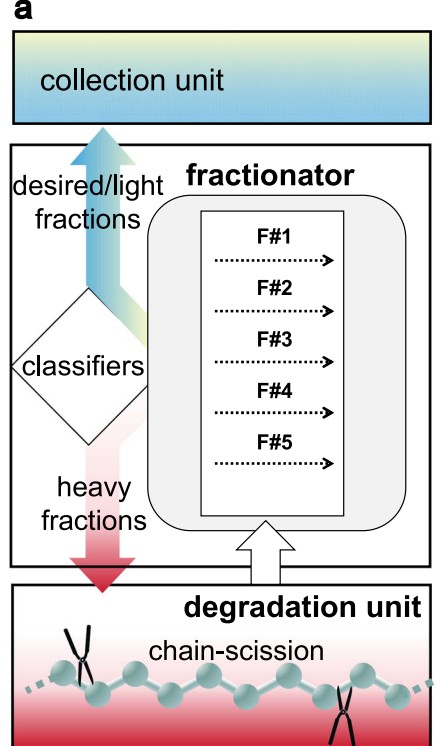

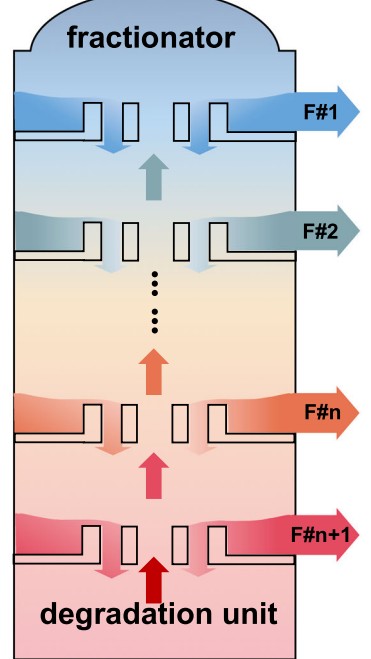

**b** Fractionated reactor based on latent heat

**Fig. 1 | Diagrams illustrating the fraction-enabled size-selective degradation. a** The degradation unit cleaves polymers by catalytic or non-catalytic reactions, yielding a mixture of shorter chains. A fractionator then differentiates chains based on a classifier using size-dependent physical properties as size criterion. Target hydrocarbons are separated into collection units, while heavier chains recirculate to the degradation unit for further degradation. This process repeats until the reactant is depleted. **b** Schematic showing the configuration of fractionated reactors based on latent heat.

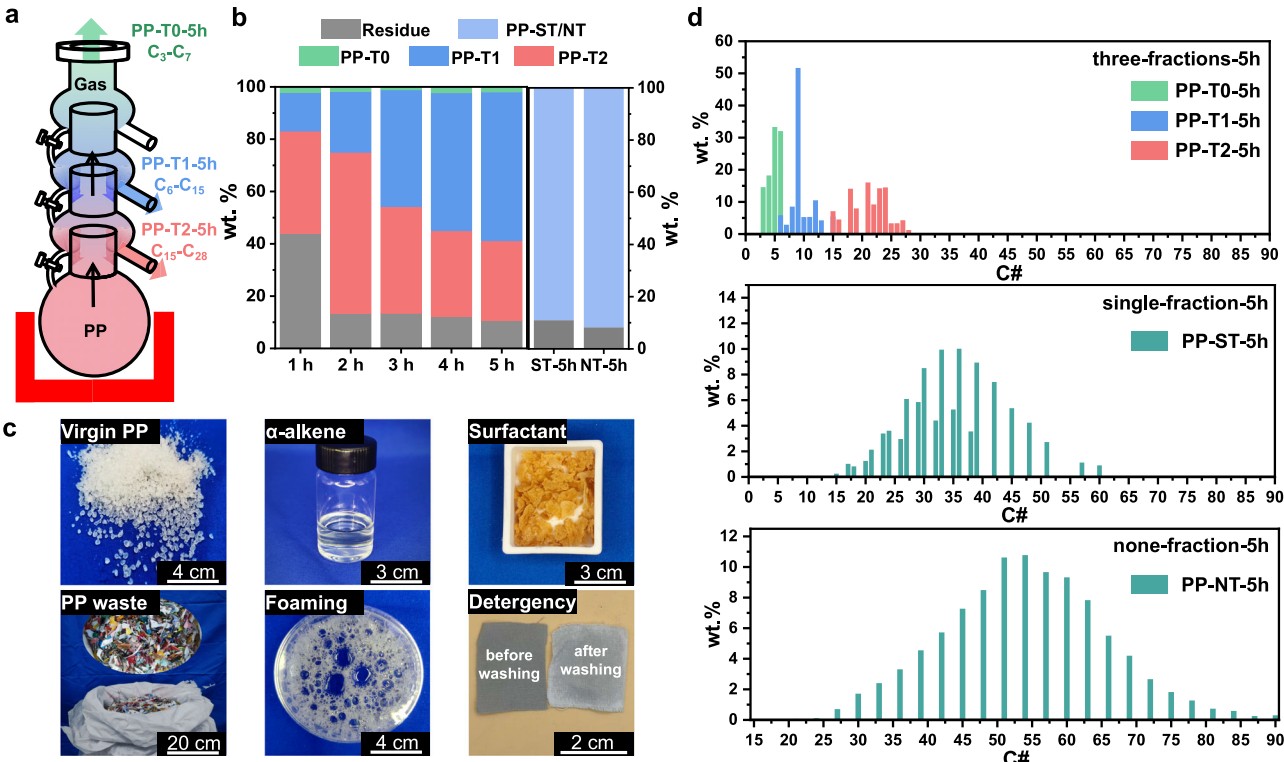

**Fig. 2 | Upcycling of PP to sulfonate surfactants using fractionated reactors.** **a** Schematic of the fractionated reactor used in PP degradation, with the bottom of the reactor heated to induce degradation. Three trays were equipped to control the separation of products, after 5 h, yielding PP-T0-5h ($C_3$–$C_7$), PP-T1-5h ($C_6$–$C_{15}$), and PP-T2-5h ($C_{15}$–$C_{28}$) with controllable CLD. **b** Distribution of pyrolysis products over 5 h. **c** Photographs of PP feedstocks (virgin PP and 10 kg of real-life PP waste), α-alkenes, sulfonated surfactants, and foaming and detergency tests. The detergency test shows sebum-stained fabrics before and after washing using PP-derived surfactants. **d** Product carbon number distributions for three-fraction, single-fraction, and none-fraction setups after 5-h degradation at 400 °C.

## Results

### Size-selective degradation of polypropylene

A custom-designed fractionated reactor with three temperature fractions (T0, T1, and T2), was manufactured to achieve size-selective degradation of PP (Fig. 2a). Under inert atmosphere, pulverized PP (10.0 g) was heated to ~400 °C (50 °C min⁻¹) in a heating mantle, generating hydrocarbon vapors fractionated by the trays. Strong entrainment was observed during the initial stage (0–0.5 h) due to reactor pressure buildup, which rapidly declined after ~0.5 h, leaving only minimal entrainment near the base of T2 (Supplementary Fig. 5). After 5 h, hydrocarbons yields in the trays were determined. The hydrocarbons in T0 (PP-T0-5h) primarily consisted of $C_3$–$C_7$ gases (Supplementary Fig. 6), with a fraction temperature ($T_f$) of ~35 °C (Supplementary Table 1). As PP-T0-5h is minor in the product, it was not sulfonated or characterized further. The hydrocarbons from T1 and T2 (PP-T1 and PP-T2) were liquids, with the yields increasing rapidly as the polyolefins underwent breakdown (Fig. 2b). After 5 h, the yields of PP-T1-5h and PP-T2-5h stabilized at 53 wt.% and 32 wt.%, respectively, with corresponding $T_f$ values of 107 °C at T1 and 208 °C at T2 (Supplementary Table 1). In control experiments, under the same reaction conditions (400 °C, 1 atm), PP (~10 g) was degraded in single-fraction (ST) and NT reactors (Supplementary Figs. 4 and 5b). The yields of hydrocarbons after 5 h (PP-ST-5h and PP-NT-5h) were similar in both reactors (89 wt.% and 92 wt.%, respectively; Supplementary Table 1).

To properly describe the CLD of hydrocarbons, calibrated variance (CV) was utilized, defined as the ratio of distribution variance to the number average carbon number ($C\#_n$; Supplementary Eq. 1-d). Compared to the commonly used polymer dispersity index, CV considers both the variability of mean and distribution range, making it a reliable parameter for reflecting distribution characteristics (See Supplementary Discussion, Supplementary Fig. 7, and Table 2). To accurately determine the distribution and number-average molar mass ($M_n$), hydrocarbons were characterized by both GC-mass spectrometry (GC-MS) and atmospheric pressure chemical ionization mass spectroscopy (APCI-MS; Supplementary Figs. 8 and 9 and Supplementary Eqs. 2 and 3). GC provided precise composition information for volatile mixtures, though its sensitivity was limited for low-volatility hydrocarbons due to column condensation[35]. Therefore, heavy fractions (boiling point > GC column temperature) were quantified using APCI-MS[13].

The initial composition of PP-T1 at 1 h (PP-T1-1h) showed a broad distribution due to entrainment and back mixing (Supplementary Figs. 5, 8, and 9), from $C_9$ to ~$C_{40}$ (CV ~ 1.57, Supplementary Tables 1 and 2). After 5 h, PP-T1-5h displayed a narrowed range, $C_6$ to $C_{15}$ (CV = 0.502, $C\#_n$ = ~$C_9$, Supplementary Tables 1 and 2). APCI-MS confirmed the distribution upper bound of PP-T1-5h terminated at $C_{15}$, with no detectable signals above 210 m/z (Supplementary Fig. 9). PP-T2-1h also exhibited a broad distribution range, $C_6$–$C_{57}$ (CV = 5.07, Supplementary Table 2 and Supplementary Fig. 9), and quickly narrowed to $C_{15}$–$C_{28}$ after 5 h (CV = 0.555, $C\#_n$ ~ $C_{21}$, Supplementary Tables 1 and 2). GC-MS determined the lower bound of PP-T2-5h, confirming it to be $C_{15}$ (Supplementary Fig. 8). In contrast, PP-ST-5h, obtained using the ST reactor was liquid (Supplementary Fig. 9d) and exhibited a broader distribution ($C\#_n$ ~ $C_{30}$, CV = 2.37, ranging from $C_6$ to $C_{60}$) than PP-T1-5h and PP-T2-5h, yet it remained narrower than PP-NT-5h (CV = 2.95, ranging from $C_{15}$ to $C_{90}$, Supplementary Fig. 9a and Supplementary Table 1).

GC-MS and APCI-MS analyses confirmed that PP-T1-5h, PP-T2-5h, and PP-ST-5h contained unsaturated hydrocarbons. Characterization

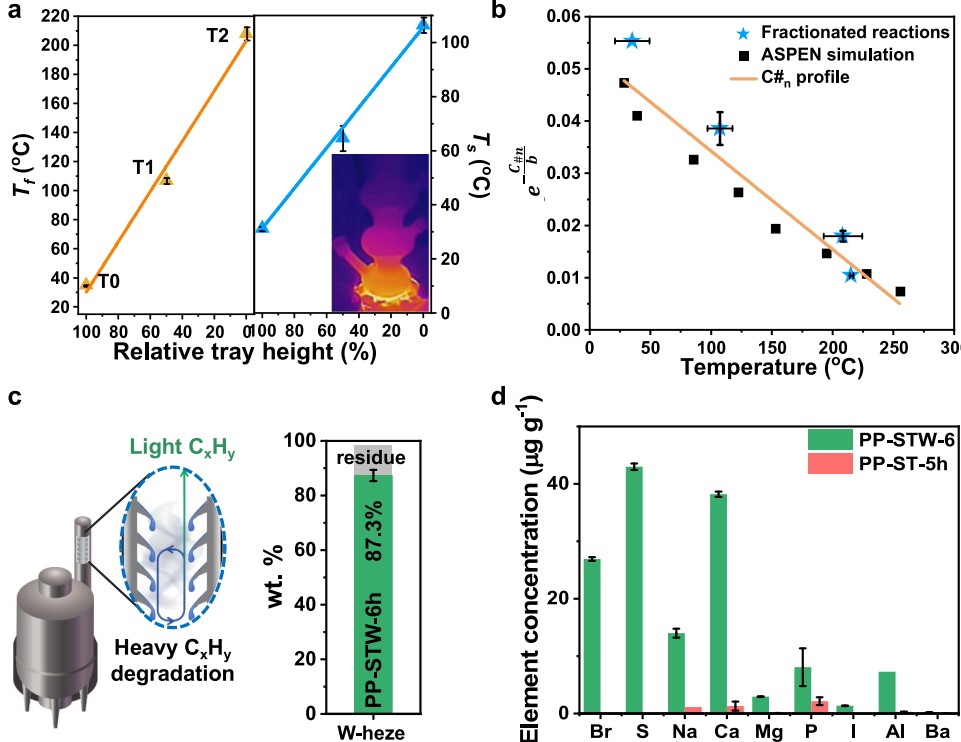

**Fig. 3 | Modeling, simulation, and scale-up experiments. a** $T_f$ and $T_s$ distribution at varying tray heights in the three-fraction reactor. All temperature values are an average of five points per fraction (Supplementary Fig. 13). **b** Correlation between $T_f$ and $C\#_n$. The $C\#_n$ profile showed a linear correlation between $\ln(T_f)$ and $e^{\frac{C\#_n}{14B}}$, as validated by experimental and ASPEN simulated data. **c** Schematic of a scale-up reactor (left) and product yield using W-heze. **d** Elemental analysis of PP-STW-6h and PP-ST-5h. All error bars represent standard deviation (SD) of at least 2 parallel experiments.

by heteronuclear multiple bond correlation NMR (HMBC-NMR) revealed the prevalent of α-alkenes (Supplementary Fig. 10, $^1$H NMR δ 4.7, $^{13}$C NMR δ 110 and δ 144), with minor internal alkenyl and aryl structures ($^{13}$C NMR δ 115–130) due to radical side reactions (see Supplementary Discussion). Quantitative $^1$H NMR analysis of α-alkenyl concentration showed a linear and near-unit correlation with the $1/M_n$, indicating the dominance of mono-α-alkenyl species (Supplementary Fig. 11 and Supplementary Eq. 4). The high α-alkene selectivity arises from methyl stabilization of PP radicals, which facilitates β-scission by reducing both Gibbs free energy and activation energy[13], favoring α-alkenyl formation (see Supplementary Discussion).

## Modeling and simulation

To elucidate the $C\#_n$ regulation in fractionated reactors, a $C\#_n$ profile model was developed, supported by ASPEN HYSYS simulation (Fig. 3, Supplementary Figs. 12–17, Supplementary Eqs. 5–13, and Supplementary Tables 3–8). Polyolefin degradation released vaporized fragments that migrated across reactor trays. Under near-equilibrium and constant molar flow conditions, the interplay of heat/mass transfer and chain scission kinetics produced a linear temperature profile (Fig. 3a) along the hydrocarbon transfer axis, corroborated by $T_f$ and external surface temperature ($T_s$). Therefore, a direct correlation between temperature and hydrocarbon $C\#_n$ can be developed (Fig. 3b) to bypass computational-intensive vapor-liquid equilibrium (VLE) calculations and avoid potential parametric inaccuracy, such as intrinsic kinetic parameters and unknown physiochemical properties. The $C\#_n$ profile unveiled a linear relationship between $\ln(T_f)$ and $e^{\frac{C\#_n}{14B}}$, and validated by experimental data cross different fractionated reactors (Fig. 3b). The ASPEN simulated temperature and $C\#_n$ in an eight-fraction reactor (True Boiling Point and Engler distillation modules; Supplementary Fig. 17a) showed similar linearity by the $C\#_n$ profile. The minor nonlinearity can be attributed to non-ideal VLE and varying

chemical composition throughout the reactor. The robust correlation between $T_f$ and $C\#_n$ demonstrated that $T_f$ serves as the primary indicator for $C\#_n$ profile in fractionated degradation.

Fraction number (F#) governs the CLDs by increasing VLE stages, selectively enriching ascending vapor with lighter hydrocarbons while concentrating heavier components in descending liquid. Therefore, the CLDs of hydrocarbon mixtures on T1 and T2 progressively sharpened from CV ~ 2 to CV < 0.6 (Supplementary Figs. 8 and 9 and Supplementary Tables 1 and 2), an analogous principle to oil refinery[36,37]. When fractions are uniformly distributed, reactors with higher F# enhanced VLE, improving isolation of light and heavy hydrocarbons per tray. Consequently, the three-fraction reactor (F# = 3) yielded product with the narrowest CLDs, outperforming both PP-ST-5h from the ST reactor (F# = 1) and the least narrow PP-NT-5h from NT reactor (F# = 0). Downward liquid flow also recirculated heavy fractions to the degradation unit for re-degradation, increasing average number of scission (s; Supplementary Table 1) and compressing the distribution range, leading to lower upper bound for the fractionated degradation products than PP-NT-5h. The effectiveness of CLD control can be quantified by separation efficiency (η; Supplementary Eqs. 14–16). The three-fraction reactor exhibited higher η and the lowest CV (geometric average η ~ 51%, CV < 0.6) than the ST (η ~ 16%, CV = 2.37) and NT reactors (η ~ 0.05%, CV = 2.95), due to enhanced efficiency of VLE and re-degradation (Supplementary Tables 2, 9, and 10). Plotting CV against η revealed a logarithmic correlation across reactor configurations (Supplementary Fig. 18). The CV-η correlation confirmed that narrowing CLDs requires enhancing η, achievable through tray engineering and hydrodynamic optimization.

## Degradation of real-life plastics and scale-up experiments

Compared to virgin PP plastics, real-life plastic wastes in supply chains exhibit dynamic characteristics (see Supplementary Discussion,

Supplementary Figs. 19 and 20). The levels of impurity (e.g., polymers, biomass, organic additives, and inorganic materials) vary between plastic wastes (Supplementary Figs. 19–22 and Supplementary Tables 11 and 12), influencing product yields, structures, and elemental compositions (Supplementary Fig. 22 and Supplementary Table 12). For instance, the fractionated degradation of real-life PP wastes (W1, W2, and W3) obtained from a single vendor exhibited reduced hydrocarbon yields and increased side reactions at lower purity, resulted in more internal alkene and aromatic byproducts.

Hence, to simplify feasibility evaluation, the scale-up experiments employed a single real-life PP waste sourced from Heze city (W-heze; Supplementary Fig. 19c). This feedstock roughly consisted 87 wt.% PP, 10 wt.% PE, 1.7 wt.% biomass, 0.7 wt.% rubber, and 1.4 wt.% others (PET, PVC, PS, dust, etc.). We processed 10-kg batches to enable meaningful lumped assumptions while ensuring valid comparisons to homogeneous laboratory-grade PP. We designed and manufactured a specialized scale-up reactor with a cost-effective, ST prickly column (STW, I.D. 40 mm; Fig. 3c and Supplementary Fig. 19a) to balance separation efficiency and manufacturing practicality. Fractionated degradation of W-heze generated comparable amount of hydrocarbon (PP-STW-6h, 87 wt.%), solid (12 wt.%), and gases (~1 wt. %) to the cases at the laboratory scale (Fig. 3c and Supplementary Table 1). PP-STW-6h was primarily composed of α-alkenes, with minor alkanes (3.5 mol%) originated from PE (~10 wt.%) and biomass (1.7 wt.%), which are typical polymeric impurities in the real-life waste (Supplementary Figs. 19–24). The $C\#_n$ of PP-STW-6h was ~27 (Supplementary Table 1). The distribution of PP-STW-6h (CV = 2.66, Supplementary Table 1) was similar to that of PP-ST-5h.

Elemental analysis revealed elemental diversity in hydrocarbons derived from W-heze than those obtained from virgin plastics (Fig. 3d, Supplementary Fig. 24, and Supplementary Table 12, see Supplemental Discussion), induced by plastic additives (Supplementary Fig. 21) and supply-chain contaminants. For PP-STW-6h, Ca (~40 mg kg$^{-1}$) and S (~43 mg kg$^{-1}$) were the most abundant elements, followed by Br (~27 mg kg$^{-1}$), Na (~15 mg kg$^{-1}$), P (~5 mg kg$^{-1}$), and others (mostly <1 mg kg$^{-1}$). Elemental profiles, however, varied across wastes. In the degradation products of W1, W2, and W3, levels of Al, Fe, Ba, Cr (100 mg kg$^{-1}$ to 1800 mg kg$^{-1}$) were more dominant than Ca, P, and S (<70 mg kg$^{-1}$). Crucially, these metal levels showed no direct correlation with purity (Supplementary Table 12), underscoring the chemical complexity of real-life plastic waste.

## Upcycling to sulfonate surfactants

The PP-derived α-alkenes were converted into surfactants through the sulfonation-hydrolysis process. Chlorosulfuric acid/1,4-dioxane (ClSO$_3$H/Diox) was utilized for sulfonation in round flasks, replacing the industry-preferred yet aggressive SO$_3$ sulfonation in thin-film reactors[24]. ClSO$_3$H/Diox gently released SO$_3$ during sulfonation[38,39], minimizing side reactions, and keeping the reaction safe in common flask[39]. The sulfonation of α-alkenes potentially went through unstable β-sultone intermediates, producing sulfonic acid, HCl, and sultones intermediates (Supplementary Figs. 25 and 26)[40]. After hydrolysis and purification, the resulting sulfonates (labeled as PP-T1-S, PP-T2-S, and PP-STW-S) were lemon yellow, presenting a high aesthetic quality. The PP-STW-S was slightly darker than the others, probably due to impurities and minor side reactions during sulfonation (Supplementary Fig. 26). X-ray photoelectron spectroscopy (XPS) and Fourier transform infrared spectroscopy (FTIR) confirmed the formation of alkenyl and sulfonate groups (Supplementary Fig. 25). High-temperature HMBC-NMR spectra (50 °C) further identified olefinic sulfonates structures (Fig. 4a and Supplementary Fig. 27), with minor shifts near δ 5.2 ($^1$H NMR) indicating other internal alkenyls, originating from side reaction during sulfonation (Supplementary Fig. 26).

Stationary defoaming experiment of PP-T2-S showed slow coalescence of small bubbles into larger bubbles (Fig. 4b). To rigorously determine the performance of the surfactants, foaming, detergency, and biodegradability were analyzed using standardized methods. Foaming heights in hard water were assessed using the Ross-Miles method (Fig. 4c and Supplementary Fig. 28), which were 14.5 and 13.7 cm for PP-T2-S and PP-STW-S, respectively. The values were higher than PP-T1-S, attributed to the longer average chain length[41]. In contrast, due to the interference on monomer alignment by the branched methyl groups[42], the foaming heights of PP-derived surfactants were lower than that of the commercial sodium dodecyl sulfonates standard (17 cm). Despite this, the branched structure facilitated wetting and permeation to textures (Supplementary Fig. 29), and thus improved detergency (ISO 2267:1986; Fig. 4d). Notably, PP-T2-S and PP-STW-S with longer chain length exhibited better detergency against sebum and protein stains than PP-T1-S and SLS (C$_{12}$) in hard water (200 mg L$^{-1}$), comparable to SDBS (C$_{18}$). The detergency of PP-STW-S was slightly lower than that of PP-T2-S, probably due to the impurities and non-active components induced by side reactions.

Traditional branched alkyl sulfonates (BAS), such as tetra-propylbenzene sulfonate, are resistant to biodegradation due to branched structures inhibiting ω-hydroxylation. Despite their superior detergency, BAS was banned and replaced by biodegradable linear alkyl sulfonates (e.g., US Clean Water Act, 1972). To evaluate the ecological safety of PP-derived surfactants in natural water bodies, biodegradability was assessed using natural sludge in the Gaoguan River at the Qinling National Reserve (sampled at Sep. 2024). *Proteobacteria* strain dominated the sludge (> 70%, Supplementary Fig. 30) and degraded surfactants following pseudo-first-order kinetics (Figs. 4e), probably through enzymatic β-oxidation and decarboxylation pathway (Supplementary Fig. 25h)[43]. PP-T1-S and PP-T2-S demonstrated more efficient biodegradation than commercial SDBS, leaving 23.0% and 38.5% of residues, respectively, after 7 days (25 ± 3 °C, Fig. 4e and Supplementary Fig. 31). This rapid biodegradation of PP-derived sulfonates can be ascribed to internal alkenyls groups (Supplementary Fig. 22e–h), which could accelerate C−H oxidation and bond cleavage by enzymes[44,45].

## Valorization process design and viability analysis

The yields of scale-up experiments served as a foundation for the process design at a scale of 10,000-tons per annum (10 tons and 8 h per batch; Fig. 5a and Supplementary Fig. 32, and Supplementary Eqs. 17–25). The valorization process included degradation, sulfonation, hydrolysis, and drying. Economic viability and ecological impacts were analyzed using techno-economic analysis (TEA) and life cycle assessment (LCA), based on ASPEN simulations and system boundary assumptions. The base scenario (B) assumed fractional degradation in tandem with SO$_3$ drop-film sulfonation and hydrolysis[24]. Degradation-distillation (D) scenario distilled PP degradation products to isolate target fractions, followed by sulfonation and hydrolysis (Fig. 5a). Degradation-distillation could not selectively produce PP-derived α-alkenes with tunable chain lengths at high yield. Nonetheless, for conservative analysis, the material flows of B and D were assumed equivalent, which overestimated the benefits of D scenarios (Fig. 5b). The drying process was optimized using solar drying (-SD), as opposed to the energy-intensive spray drying (-P), producing dried surfactants (~13,000 tons, ~95% purity) and by-products (365 tons, a mixture of surfactants and salts).

Fractionated degradation and solar drying improved both sustainability and economic viability. Under the defined market conditions, B-SD exhibited the lowest capital expense (CAPEX) and operating cost (OPEX) at $2.4 M and $11.8 M, respectively (Fig. 5c, d and Supplementary Tables 13–16), while also achieving the highest CO$_2$ reduction (1841 kgCO$_2$/ton). The annual revenue of B-SD was ~$14.8 M, with an annual net profit of $2.1 M, equating to an internal rate of return (IRR) of 70.1% and a payback period of 1.24 year (Supplementary Table 16). The stability of B-SD against market fluctuations was evaluated by sensitivity analysis (Supplementary Fig. 33), showed low

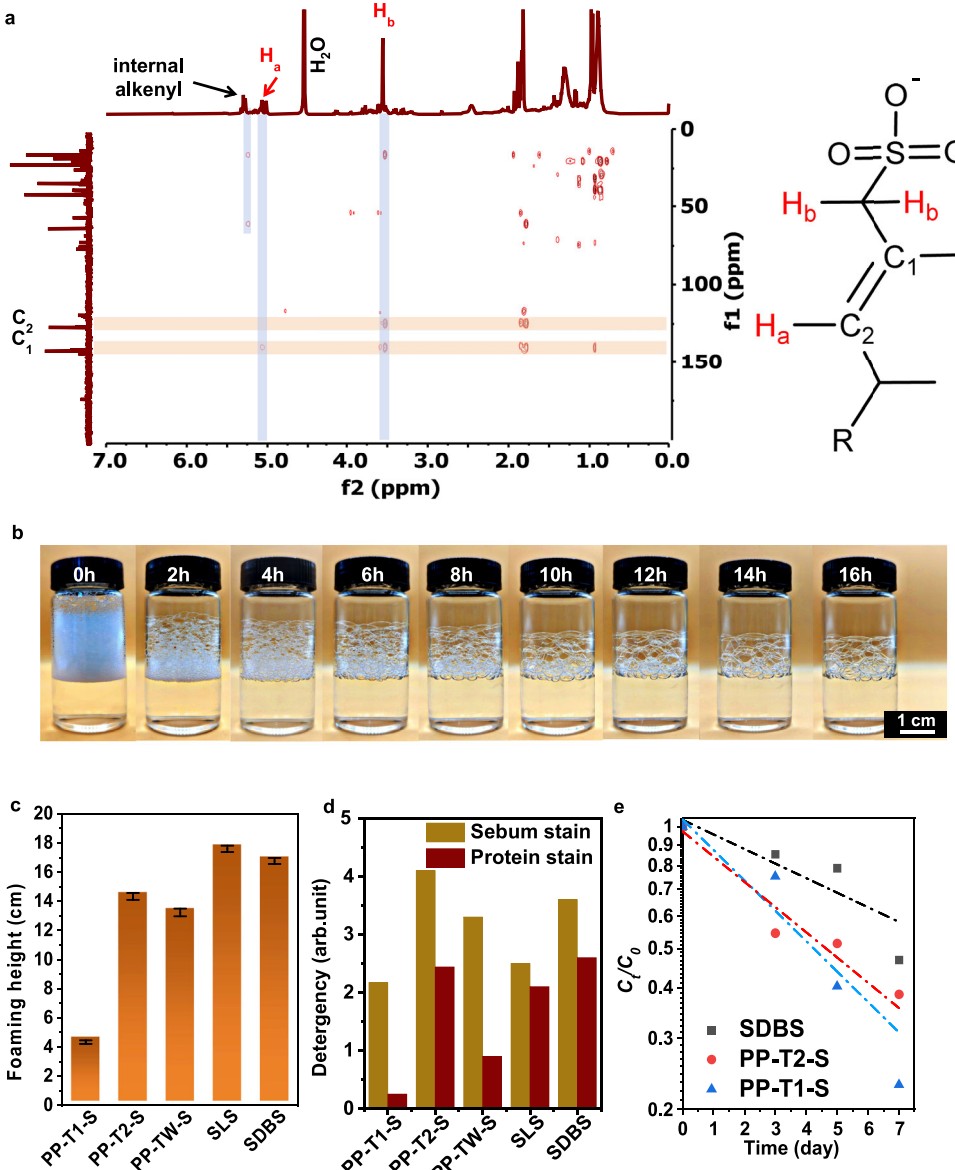

**Fig. 4 | Chemical structure and performance of PP-derived surfactants. a** HT-HMBC-NMR spectrum of PP-T2-S in $D_2O$ at 50 °C. **b** Time-lapse images show the stability of PP-T2-S foam from 0 to 16 h. **c** Foaming heights determined by Ross-Miles methods and **d** detergency against sebum and protein stains on standard cotton fabrics. The PP-derived surfactants are compared against commercial surfactants SLS and SDBS. All error bars represent SD of at least 5 parallel experiments. **e** Biodegradation of PP-T1-S, PP-T2-S, and commercial SDBS in natural water collected from Qinling National Reserve.

sensitivity to primary market. In contrast, D-SD reduced IRR to 32.7% and emitted 40% more $CO_2$ than B-SD, confirming the superiority of fractionated degradation in sustainability and profitability than degradation-distillation. Solar drying is more sustainable than spray drying (Supplementary Table 16), yet, the turnover rate of solar drying is slow (~ 3 work days, mild wind, and 50% relative humidity)[46]. Therefore, B-P could serve as an efficient and flexible substitute. The economic and ecological performance of B-P was similar to those of B-SD, with 20% higher fixed capital investment and 7% less $CO_2$ reduction (Supplementary Table 16). D-P exhibited the lowest benefits, yet still twice the profitability (IRR = 22%) than the sulfonation industry in China (~10%)[47], with 1260 kg$CO_2$/ton $CO_2$ reduction benefit, highlighting the economic and ecological benefit of PP waste valorization.

## Discussion
In summary, this work demonstrated a fractionated degradation, leveraging size-dependent latent heat, for tuning hydrocarbon chain length and CLD during the degradation of polyolefins. Experiment, modeling, and simulation showed a $T_f$-$C\#_n$ correlation, while F# modulated CLDs through VLE and re-degradation, progressively narrow the distribution per tray. Consequently, narrow α-alkenes distribution (CV < 0.6) was obtained using a three-fraction reactor, with CLD outperformed those obtained using reactors with lower F# and in literature. The fractional degradation was further validated by scaling-up to 10 kg using real-life plastic waste, showing product yields, chemical structures, and elemental profile affected by plastic waste purity. The α-alkenes derived from virgin or real-life plastic waste were further valorized into sulfonates via sulfonation and hydrolysis. The resulting sulfonates exhibited enhanced foaming, detergency, and biodegradability performance comparable to commercial surfactants. Industrial-scale TEA and LCA showed that the upcycling of PP into sulfonates possesses improved sustainability and profitability than the industrial preparation of sulfonate surfactants.

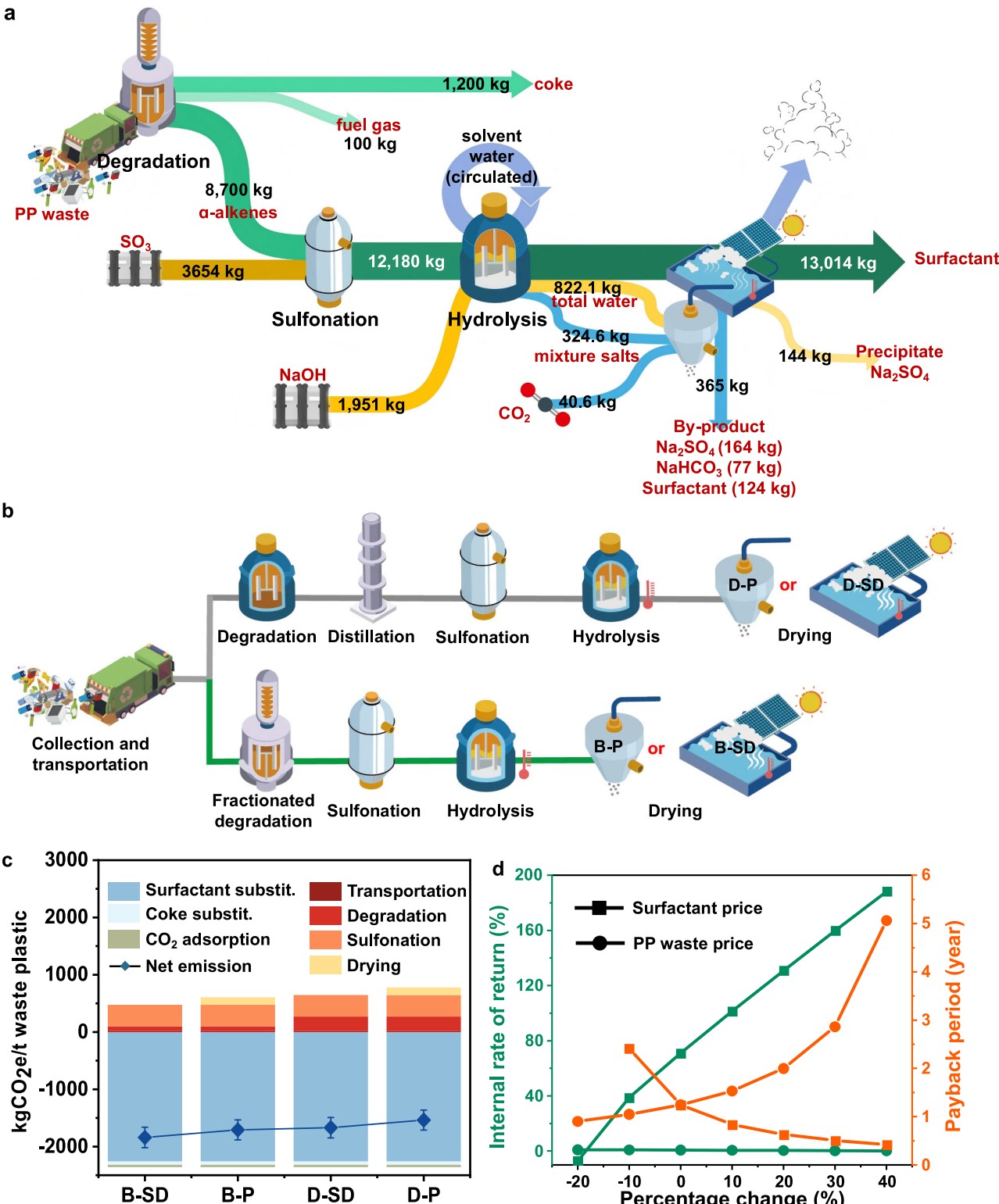

**Fig. 5 | LCA and TEA of PP fractionated degradation and upcycling. a** Sankey diagram depicts the material flow for real-life PP upcycling into surfactants at a scale of 10,000 kg of PP waste. **b** Schematics of four representative scenarios: B-P, B-SD, D-P, and D-SD. The material flows for scenarios B and D are assumed to be equivalent for conservative analysis. **c** LCA and sensitivity analyses comparing $CO_2$ emissions of the four scenarios. The range bars represent the emissions at 80%, 87%, and 90% yields, with the lowest emissions corresponding to the highest yield (90%, top bar) and the highest emissions to the lowest yield (80%, bottom bar) (Supplementary Table 16). **d** Sensitivity analysis of profitability concerning PP waste and surfactant prices. Surfactant prices at −20% discount led to invalid payback period, and thus not shown.

The work herein provides a basic reactor framework, quantitative distribution descriptor, and a promising green surfactant for general cleaning purposes. Fractional degradation represents a advantageous methodology for controlling $C\#_n$ and CLD in polyolefin degradation (Supplementary Table 17), improving energy efficiency, product selectivity, yields, and sustainability than direct distillation of pyrolysis-oil. We anticipate the process to be amendable to other plastics, and can be potentially improved by integrating with catalytic processes, tray engineering, and operation optimization, enabling finer manipulation in adjusting chain length and distribution to promote real-life plastic upcycling.

## Methods

### PP degradation in fractionated reactors
Laboratory-grade PP (10 g) or real-life waste PP (W1, W2, W3) was loaded into a fraction reactor consisting of a degradation unit (~ 100 mL) and trays (~ 20 mL each, Supplementary Fig. 5). The reactor was seamlessly molded to ensure air-tightness and was set up in an Ar-filled glovebox to avoid oxygen contamination. Inside the glovebox, the tray was connected to a total condenser via a frosted neck and sealed with vacuum grease. The condenser was plugged with a fluorinated stopper and connected to a gas bag. The setup was moved out of the glovebox and placed in a heat mantle. To optimize thermal insulation and concentrate heat within the degradation unit, an thick asbestos layer was installed atop the electric heating jacket. The system maintained a controlled heating rate of ~ 50 °C min$^{-1}$. Temperature monitoring was achieved through a thermocouple integrated into the heating assembly, providing continuous verification that operational parameters remained within the prescribed range throughout the reaction duration. After a few hours, the reactor was cooled to room temperature. The setup was cleaned in a muffle furnace and then washed with aqua regia. The cleaned reactor was rinsed with DI water and stored in a desiccator for next use.

### Upcycling scale-up using real-life plastic wastes
The fractional degradation process was scaled up to 10 kg using W-heze. Approximately 10 kg of pulverized waste was loaded into a ST reactor through a flange structure. The flange was then sealed with a steel cover and secured with a hoop. The setup was purged with high-purity nitrogen (~ 100 mL s$^{-1}$) for 15 min. The heating and stirring units were activated (~ 3 °C min$^{-1}$ and ~ 50 rpm) to initiate the reaction. After ~ 2–4 h, the reaction chamber reached 400 °C. Entrainment was observed in the reactor chamber, while no entrainment was detected in the collector. During the reaction, the tray temperature rapidly rose from r.t. to ~ 220 °C. The hydrocarbon flowed into the collecting chamber, exhibiting a light yellowish color. After 6 h, the reaction was halted, and the products (PP-STW-6h) was drained into a 10-L sample bottle. The hoop on the flange was removed, and any solid residue adhering to the inner wall was scraped off using a shovel. The bottom flange was opened to collect the residue. The yields were measured using a platform balance with an accuracy of 0.1 kg.

### Sulfonation by ClSO₃H/Diox
A flat-bottom flask and a constant-pressure separation funnel were dried at 120 °C for at least 12 h. The hot glassware was instantly transferred into an Ar-filled glovebox and sealed with fluorinated stoppers. After cooling to room temperature, PP-T2-5h (1.0 g, $n_{C=C}$ = 3.97 mmol g$^{-1}$), dissolved in 10 mL chloroform, was transferred to the flask using a 10 mL Teflon® syringe. Simultaneously, ClSO₃H/Diox (Diox 0.70 g, 7.94 mmol, chlorosulfonic acid, 0.56 g, 4.76 mmol) and 10 mL chloroform, were transferred to the separation funnel using a Teflon® syringe equipped with a ceramic needle. The flask was then cooled with an ice-salt bath. The sulfonation reagent was added dropwise into the flask. The pressure of the sulfonation setup was released through an outlet in the separation funnel. The gas was absorbed using an AgNO₃ aqueous solution (0.1 mol L$^{-1}$). The precipitate was characterized by XRD.

Aging: The sulfonation was stopped after 30 min, and the mixture was allowed to age at room temperature for another 30 min.

Hydrolysis: An aqueous NaOH solution (1 mol L$^{-1}$) was added into the mixture under vigorous stirring, forming a yellowish emulsion. Subsequently, the hydrolysis of the sulfonated mixture was conducted in a 100 mL hydrothermal reactor (170 °C, ~ 1 MPa) for 30 min. After cooling to room temperature, the sulfonated products were harvested by cooking-off water using a rotary evaporator and then extracted using CHCl₃. Insoluble salts were removed by centrifugation. The CHCl₃ solution was evaporated under reduced pressure, leaving light yellow solid surfactants. The product was scraped off and stored in a desiccator.

### Sulfonation by SO₃
The sulfonation setup (Supplementary Fig. 3) was dried at 120 °C for 12 h. After drying, the setup was rapidly purged with an Ar and allowed to cool in an ice bath. Under an Ar atmosphere, hydrocarbon (~ 10 g, PP-NT-5h or PP-T2-5h) was added to the setup and diluted with chloroform at a 1:2 volume ratio. The gas inlet was connected to a buffer bottle, which was in turn linked to an SO₃ tank and an air tank. The SO₃/air mixture (2 mL s$^{-1}$) was introduced into the chloroform solution under vigorous stirring. The SO₃ concentration in the SO₃/air was controlled by adjusting the gauge pressure, with a preferred volume ratio of 2–10 vol.%[24].

## Data availability
The source data generated in this study are deposited in the Zenodo database under accession code 10.5281/zenodo.17390225 [https://zenodo.org/records/17390225] and are available in the Supplementary Information. All data are available from the corresponding authors upon request.

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

## Acknowledgements

We appreciate the financial support from the Northwestern Polytechnical University Aoxiang (foreign) scholarship (0652024GH0201287, Z.X.) and Fundamental Research Funds for the Central Universities (G2024KY05107, Z.X.). This research is partially supported by National Natural Science Foundation of China (22309146, F.P., 72371059, 72293563, J.X., 72372019, F.L., 12401337, H.S.), Key Research and Development Projects of Shaanxi Province (2024GX-YBXM-418, F.P.), National Key R&D Program of China (2024YFE0207600, K.J.C.), and CIRP Open Fund of Radiation Protection Laboratories (CIR-P2024OF03, K.J.C.). The authors acknowledge generous donation and helpful assistance in the design and operation of the scale-up reactor from engineer Zhiquan Yang, who persistently pursues chemical engineering in his lifetime career with curiosity and enthusiasm. The authors also acknowledge the professional comment and suggestion by Jianmin Xu, ACCA certified accountant on the calculation and modeling in TEA. We are also grateful to Jinliang Liu of XZL Bio-technology Co., Ltd., and Shengti Cao and Xiaocheng Liu of the China Research Institute of Daily Chemical Industry for their assistance with $SO_3$ sulfonation. Finally, we thank Mr. Tong Gao for his help in preparing the experiments. The authors also acknowledge the NMR and Spectroscopy Laboratories in Analyses and Testing Center at Shandong University and Shandong Normal University, as well as Eceshi (www.eceshi.com) for APCI and elemental analyses.

## Author contributions

Z.X. conceived and supervised the project with F.P. Z.X. and F.P. designed the project. Z.X., Q.Z., and F.P. designed the fractionated reactors for laboratory-scale experiments. Z.X. and Y.Z. conducted degradation-upcycling experiments, GC, APCI-MS, FTIR, XPS, and UV–vis experiments. Z.X. and Q.Z. performed modeling and ASPEN simulation. Z.X. and Y.Z. performed NMR experiments. Y.R. and K.J.C. performed PXRD experiments. Z.X. and F.P. performed scale-up experiments. M.C. and Y.Z. performed surfactant performance characterizations. F.L., H.S., and Y.Z. developed distribution descriptor. Z.X. and T.W. performed life cycle analysis. Z.X. and J.X. conducted techno-economic analysis. Z.X., R.Y., and F.P. analyzed the data, interpreted the results, and wrote the manuscript. All authors proofread the manuscript.

## Competing interests

Z.X., Y.Z., F.P., and Q.Z. are inventor on a patent (ZL 202411310504.3) pertaining to the fractional degradation of polyolefins to tune hydrocarbon chain lengths reported in this work. F.P., Z.X., and K.J.C. are inventor on a patent application (CN 2024100370498) related to the sulfonation of polyolefins-derived hydrocarbons into sulfonates detergent reported in this work. The other authors declare no competing interests.
