## [Transparent Peer Review file · Nature Communications]

Fractionated degradation and valorization of polypropylene waste into sulfonate surfactants

Corresponding Author: Professor Zhen Xu

Version 0:

Reviewer comments:

Reviewer #1

(Remarks to the Author)

The manuscript presents the fractionated pyrolysis of PP and waste PP, coupled with the valorization of the resulting pyrolysis products into sulfonate surfactants. The authors demonstrate a reactor design that enables the fractionated degradation of PP using three separation units (T0, T1, and T2) over a 5-hour operation window. Comparisons between three-fraction, single-fraction, and non-fractionated setups show significant differences in carbon number distribution, validating the effectiveness of the fractionated approach. A scaled-up version of the reactor system was applied to 10 kg of mixed waste PP using a single-fraction column, and the α -olefin-rich products were upcycled into sulfonate surfactants. These were benchmarked against commercial sodium dodecylbenzenesulfonate (SDBS), demonstrating comparable foaming, detergency, and biodegradability. The study is further supported by ASPEN-based techno-economic analysis (TEA) and life cycle assessment (LCA), showing favorable cost and environmental metrics for the proposed process. Mechanistic discussions are minor and not sufficient. Please see below for more detailed questions.

1. Lines 140 – 141: Can the authors clarify the operational conditions for PP-ST and PP-NT? How do the yields of PP-ST, PP-NT, and fractionated compare?
2. Figure 1.d: Can the authors add the experimental time at which the product carbon number distribution was detected?
3. Lines 159 – 161: CV = 1.57 cannot be found in Table S1, as the text mentions. The value is in Table S2.
4. Lines 159 – 161: Can the authors add the product carbon number distribution for PP-T1 1 hour? Why did the product distribution narrow after 5 hours?
5. Line 165: What does having no significant signal beyond 210 m/z about the carbon number distribution mean?
6. Lines 170 – 174: Can the authors further explain the differences in the carbon number distribution observed for PP-ST, PP-NT, PP-T1, and PP-T2? Why did PP-ST have a narrower carbon number product distribution than PP-NT? Can the authors clarify if the pyrolysis oil obtained from PP-ST was liquid at room temperature?
7. Lines 221 – 222: How does the separation efficiency affect the decomposition of PP in terms of carbon number distribution, residue, and yield of fractions?
8. Lines 233 – 234: What type of polymers and additives can be found in the waste used?
9. Lines 238 – 240: Is the yield of hydrocarbons for PP-STW statistically different from the yield of hydrocarbons for PP-ST?
10. Lines 240 – 241: What is the difference between the yield of the PP-STW (87 wt.%) and the yield of liquid hydrocarbons for regenerated plastic bags (50 wt.%)? Please add standard deviation.
11. Lines 244 – 245: Can the authors verify if the C#n is shown in Table S1?
12. Lines 247 – 250: The authors compared the impurities of PP-STW with the impurities of PP-T2. However, the first corresponded to the degradation of PP waste with a single fraction, while PP-T2 corresponded to the degradation of PP using three fractionation column. As shown in Figure 1, these are different processes that lead to different carbon number distributions. Then, can the authors compare the impurities of PP-STW and PP-ST?
13. Lines 252 – 253: Can the authors comment on what type of impurities can be found in the waste used that lead to high concentrations of Ba, Sr, Pb, Fe, etc.?
14. Lines 310 – 311: What is the typical degradation of SDBS?
15. Table S1: Can the authors add the time at which this data was obtained?
16. Table S1: What is the standard deviation for PP-NT?
17. Table S11: What is the difference between experiments 1, 2, and 3 in PP-STW? Why can the elements from Ba to In not be detected in Exp3 but can be detected in Exp1 and Exp2?
18. How were the heat transfer coefficients validated during the simulations? Were there any deviations observed at scale due to non-uniform heating?

19. Was there any entrainment or back-mixing observed between trays that could influence CLD narrowing?
20. What was the Murphree tray efficiency assumed or measured for the ASPEN simulations?
21. Was residence time distribution (RTD) analyzed? If so, did the setup approximate plug flow or was there evidence of tailing/mixing?
22. Could trace metals (Fe, Zn, Ca) catalyze unwanted side reactions or impact sulfonation yield?
23. Was a sensitivity analysis run on the vapor-liquid equilibrium parameters? What assumptions were made regarding phase behavior?
24. Were any oligomerization or cross-linking side products detected at higher residence times?
25. Was there fouling or residue accumulation in the tray or collector units over repeated runs?
26. How does reactor performance change over time with waste PP (e.g., increased pressure drop, reduced vapor flow)?
27. Was external heat loss accounted for in the ASPEN simulation?
28. No discussion of non-idealities, such as entrainment, foaming, or pressure drops.
29. Side reactions are not discussed.
30. There is a vague discussion on β -scission control. Provide DFT or kinetic modeling.
31. At 400 C under inert gas, radical reactions are expected, but the authors do not discuss any related reactions.
32. The impact of contaminants on the degradation pathway is not explained.

Reviewer #2

(Remarks to the Author)

Reviewer #3

(Remarks to the Author)

This is a relevant study that pursues surfactants production from waste plastics. The proposed strategy consists of two steps, selective conversion of plastics in suitable hydrocarbon fractions (olefins of different chain lengths), then, these hydrocarbons are further transformed by sulfonation. The paper is in general clear and well written, moreover, it has some original aspects as the polymer degradation couples with products fraction. However, there is a need for improvement and clarification before considering publication.

The process was studied feeding PP, however, the properties and typical product distribution obtained in different polyolefins degradation is similar. The authors should discuss the applicability of this approach to polyolefins mixtures, in fact, it could be easier to find waste polyolefins mixtures than pure PP.

The authors should clarify how were determined the carbon distributions shown in Figure 1, the explanations are unclear. Results discussion is weak, the main contribution of this study is the incorporation of the fractionation system, accordingly, its role on products composition should be further discussed. Moreover, the authors should clearly explain the real advantages of this fractionation strategy, is it better than a typical products separation after oil condensation (i.e., a conventional distillation).

What is the interest and relevance of the modeling section? What about the consistency of the kinetic parameters of this model? The use of suitable kinetic parameters is a critical aspect in that kind of models, sometimes, this is very challenging as long as reaction rate is controlled by heat transfer and not by reaction rate.

The overall process proposed (Figure 4) is interesting; however, the applicability of this process under real process conditions should be commented. Thus, real waste plastics contains several different impurities, what is their potential role on different reaction steps and on the quality and application of final product (surfactants). Please, clarify.

The authors should provide additional details regarding the reactor operation. How was the reactor heated? what about temperature control? What is the heating rate used?

The degradation temperature of 400 °C is quite low, accordingly, reaction rate is low and the process requires very long reaction times. Why not using higher temperatures?

The pilot scale process operates in batch regime, moreover, the heating rate is very slow and therefore the time required to complete a reaction is of several ours. This approach is not suitable for the process scale up as long as the efficiency and process throughput are low. Can this process be feasible at full scale? Plastics thermochemical valorization processes should operate in continuous regime to ensure profitability. Please, comment.

A weak point of the paper is the lack of comparison with previous literature. The authors should provide a complete comparison of product yields and their composition with those reported in the literature with other plastics pyrolysis technologies and approaches.

Version 1:

Reviewer comments:

Reviewer #1

(Remarks to the Author)

The authors resolved the issues previously reported.

Reviewer #2

(Remarks to the Author)

Reviewer #3

(Remarks to the Author)

The authors have revised the paper and soundly responded the comments and doubts of the reviewers. Consequently, the paper has been improved and clarified, and in the reviewer opinion it is now publishable as it is

Reviewer #1

The manuscript presents the fractionated pyrolysis of PP and waste PP, coupled with the valorization of the resulting pyrolysis products into sulfonate surfactants. The authors demonstrate a reactor design that enables the fractionated degradation of PP using three separation units (T0, T1, and T2) over a 5-hour operation window. Comparisons between three-fraction, single-fraction, and non-fractionated setups show significant differences in carbon number distribution, validating the effectiveness of the fractionated approach. A scaled-up version of the reactor system was applied to 10 kg of mixed waste PP using a single-fraction column, and the α -olefin-rich products were upcycled into sulfonate surfactants. These were benchmarked against commercial sodium dodecylbenzenesulfonate (SDBS), demonstrating comparable foaming, detergency, and biodegradability. The study is further supported by ASPEN-based techno-economic analysis (TEA) and life cycle assessment (LCA), showing favorable cost and environmental metrics for the proposed process. Mechanistic discussions are minor and not sufficient. Please see below for more detailed questions.

Author reply: We wish to express our sincere gratitude to the reviewer for their thoughtful and kind comments on our manuscript. Below, we provide our point-by-point reply to each of the valuable and constructive comments. Below, we respectfully offer our point-by-point reply to each of the valuable suggestions provided.

Comment #1: *Can the authors clarify the operational conditions for PP-ST and PP-NT? How do the yields of PP-ST, PP-NT, and fractionated compare?*

Author reply: The authors thank the reviewer for the valuable comments. We have elaborated the operational conditions and yields below, as well as in the revised Manuscript and revised Supporting Information.

Operation conditions:

PP thermal degradation in both single-fraction (ST) and non-fraction (NT) reactors employed identical conditions: 400°C, 50 °C min⁻¹ heating rate, and ~1 atm pressure. Minor procedural variations stemmed solely from reactor configuration differences. All systems underwent the same temperature calibration protocols using identical thermocouple thermometers. Specifically, reactor temperatures were calibrated with thermocouple probes positioned at reactor bottoms (**Supplementary Figures 4 and 5**), implementing gradual heating until stabilization at target temperatures for 30 minutes. Given these highly similar reaction conditions, reactor configuration presents the only variable enabling parallel comparisons between reactors.

Yields of PP-ST, PP-NT, and three-fraction reactors:

Yields for the three-fractions (PP-T0-5h, PP-T1-5h, PP-T2-5h), ST, and NT are similar at ~87 wt.%, 89±1.4 wt.%, and 92±1.5 wt.%, respectively. Although, it seems climbing with reduction of tray number, yet considering the standard deviation, the difference is insignificant. For clarity, we have thoroughly modified

Figure 1 to include yields of PP-ST-5h and PP-NT-5h. The manuscript is revised with detailed experimental procedures and operating conditions for all reactors.

Figure. 1. Upcycling of PP to sulfonate surfactants using fractionated reactors. (a) Schematic of the fractionated reactor used in PP degradation, with the bottom of the reactor heated to induce degradation. Three trays were equipped to control the separation of products, after 5 h, yielding PP-T0-5h (C₃-C₇), PP-T1-5h (C₆-C₁₅), and PP-T2-5h (C₁₅-C₂₈) with controllable CLD. (b) Distribution of pyrolysis products over 5 h. (c) Photographs of PP feedstocks (virgin PP and 10 kg of real-life PP waste), α-alkenes, sulfonated surfactants, and foaming and detergency tests. The detergency test shows sebum-stained fabrics before and after washing using PP-derived surfactants. (d) Product carbon number distributions for three-fraction, single-fraction, and none-fraction setups after 5-h degradation at 400°C.

Main text, Page 5: In control experiments, under the same reaction conditions (400 °C, 1 atm), PP (~ 10 g) was degraded in single-fraction (ST) and NT reactors (Supplementary Figures S4 and 5b). The yields of hydrocarbons after 5h (PP-ST-5h and PP-NT-5h) were similar in both reactors (89 wt.% and 92 wt.%, respectively; Supplementary Table 1).

SI, Page 1:

1. Reaction temperature calibration and pressure monitoring.

The reaction temperature of reactors with three-, single-, and none-fraction were calibrated using a thermocouple thermometer. Specifically, the thermocouple probe was placed at the bottom of the reactor

(**Supplementary Figures 4 and 5**). The heating mantle temperature was gradually increased, and the thermocouple readings were monitored until the temperature stabilized at the target value for 30 minutes. The reactor pressure during the reaction was directly measured using a digital pressure meter equipped with needle probes.

3. PP degradation in single-fraction reactors.

The ST reactor consists of three main components: a heating unit (150 mL volume), a tray, and a total condenser. The reactor body was precision-molded to ensure complete airtightness. Approximately 10 g of polymer sample was loaded into the reactor, which was then transferred to an argon-filled glovebox to prevent oxygen contamination. Inside the glovebox, the condenser was connected to a grease-sealed frosted glass flask for liquid product collection. The assembled system was placed in a heating mantle and gradually heated to 400 °C (heating rate: ~ 50 °C min^{-1} ; pressure: ~ 1 atm). After 5 hours, the reactor was cooled to ambient temperature, and the condensed liquid products were collected for subsequent analysis. For reactor maintenance, the setup was first calcined in a muffle furnace, followed by aqua regia washing. Finally, the reactor was thoroughly rinsed with deionized (DI) water and stored in a desiccator for next use.

4. PP degradation in none-fraction reactor.

The experimental setup replicated our previously reported system,¹ with similar reactor geometry and operational parameters. A minor difference is inclusion of bay structures on the condenser for larger scale of reactions (**Supplementary Figure 4**). Approximately 10 g of polymer was loaded into the reactor, which was subsequently transferred into an argon-filled glovebox. The reactor assembly was sealed using a quartz glass plate secured by stainless steel clamps and a fluorinated O-ring for enhanced pressure resistance. Positioned in a heating mantle, the reactor was heated to 400 °C (heating rate: ~ 50 °C min^{-1} ; pressure: ~ 1 atm). During pyrolysis, evolved vapors condensed on the water-cooled inner surfaces of the attached condenser. A specially designed bay structure effectively captured heavier condensates through gravitational draining. After 5 hours of continuous operation, condensed products were recovered via either (i) mechanical scraping or (ii) solvent extraction using refluxing hexanes, chloroform, or tetrahydrofuran.

Supplementary Figure 4. (a) Technical drawing of a typical thermal-gradient non-fraction degradation reactor. (b) Digital image showing strong entrainment within the reactor. The red laser was directed near the product outlet of the thermal gradient reactor. The light intensity of the trail significantly decreases due to scattering caused by the high density of droplets from entrainment. This observation was corroborated by the broad distribution of PP-NT-5h (Figure 1 and Supplementary Figure 9) and previous research.^{1,20}

Comment #2: *Can the authors add the experimental time at which the product carbon number distribution was detected?*

Author reply: The authors thank the reviewer for this helpful suggestion. We apologize for the inconvenience caused by this missing information. The carbon number distribution was analyzed after 5 h degradation. We have now added the specific experimental time behind each sample name throughout the manuscript. For example, products collected at T2 after 5h is labeled as PP-T2-5h; products collected in NT reactor after 5h is labeled as PP-NT-5h.

Comment #3: $CV = 1.57$ cannot be found in Table S1, as the text mentions. The value is in Table S2.

Author reply: The authors thank you for the correction. We apologize for this careless mistake. The reference to the calibrated variance ($CV = 1.57$) was wrongly correlated to **Supplementary Table 1**. We have fixed this error in the revised manuscript as follows.

Main text, Page 6: The initial composition of PP-T1 at 1 h (PP-T1-1h) showed a broad distribution due to entrainment and back mixing (**Supplementary Figures 5, 8, and 9**), from C_9 to $\sim C_{40}$ ($CV \sim 1.57$, **Supplementary Tables 1 and 2**). After 5h, PP-T1-5h displayed a narrowed range, C_6 to C_{15} ($CV = 0.502$, $C\#_n = \sim C_9$, **Supplementary Tables 1 and 2**).

Comment #4: Can the authors add the product carbon number distribution for PP-T1 1 hour? Why did the product distribution narrow after 5 hours?

Author reply: The author appreciate the reviewer for the constructive comments. We have rewritten the manuscript to improve clearance. A discussion of distribution reduction is included in the supporting information. To improve the convenience for reading, a panel b was added in **Supplementary Figure 9** to include distributions for PP-T1-1h and PP-T2-1h. A brief discussion about distribution narrowing is provided below, with more detailed discussion in the revised Supporting Information

Discussion of distribution reduction. Three stages are anticipated in the fractionated degradation: initiation, equilibrium, and termination, as indicated by the kinetic curves (**Supplementary Figure 22a**). The narrowing of the product distribution after 5 hours can be attributed to the establishment of vapor-liquid equilibrium (VLE) after the reaction initiation. In the initiation stage, PP is introduced and degraded. The vapor of fragments rise upward along the direction of mass transfer. At this initial point, the VLE is not yet established and comes with entrainment phenomena (**Supplementary Figure 5d**), together leading to the broad distributions of PP-T1-1h and PP-T2-1h at the first 1 h (**Supplementary Figures 8-9**). In the equilibrium stage, three factors are taking effect to narrow down the distribution.

1. **Re-degradation:** Separation of heavy fraction (oligomers, heavy hydrocarbons, undesired long chain) and re-degradation in the degradation unit is a key to decline the distribution upper boundary. Without or with limited re-degradation (*Science*, **2023**, 381, 666; *Nat. Sustain.*, **2024**, 7, 1681-1690), ultra-heavy hydrocarbons would be dominant in the products (**Figure R2**), leading to inclusion of ultra heavy hydrocarbons.

2. **Vapor liquid equilibrium (VLE).** Reaching VLE equilibrium requires time due to finite rates of mass and heat transfer. Therefore, with continuous reflux and re-vaporization, countercurrent contact of vapor and

liquid on each tray ascend volatile component and descend the heavy fractions, gradually forming a gradient distribution of molecules by sizes and volatility (**Figure 2b**).

3. **Fractionation:** Increasing fractionation number improve the separation efficiency to promote more efficient re-degradation and VLE establishment.

Following revisions were made to make the manuscript being more accessible.

Main text, Page 10: Fraction number ($F\#$) governs the CLDs by increasing VLE stages, selectively enriching ascending vapor with lighter hydrocarbons while concentrating heavier components in descending liquid. Therefore, the CLDs of hydrocarbon mixtures on T1 and T2 progressively sharpened from $CV \sim 2$ to $CV < 0.6$ (**Supplementary Figures 8-9, Supplementary Tables 1-2**), an analogous principle to oil refinery.^{36,37} When fractions are uniformly distributed, reactors with higher $F\#$ enhanced VLE, improving isolation of light and heavy hydrocarbons per tray. Consequently, the three-fraction reactor ($F\# = 3$) yielded product with the narrowest CLDs, outperforming both PP-ST-5h from the ST reactor ($F\# = 1$) and the least narrow PP-NT-5h from NT reactor ($F\# = 0$). Downward liquid flow also recirculated heavy fractions to the degradation unit for re-degradation, increasing average number of scission (s ; **Supplementary Table 1**) and compressing the distribution range, leading to lower upper bound for the fractionated degradation products than PP-NT-5h. The effectiveness of CLD control can be quantified by separation efficiency (η ; **Supplementary Equation 14**). The three-fraction reactor exhibited higher η and the lowest CV (geometric average $\eta \sim 51\%$, $CV < 0.6$) than the ST ($\eta \sim 16\%$, $CV = 2.37$) and NT reactors ($\eta \sim 0.05\%$, $CV = 2.95$), due to enhanced efficiency of VLE and re-degradation (**Supplementary Tables 2, 9, and 10**). Plotting CV against η revealed a logarithmic correlation across reactor configurations (**Supplementary Figure 18**). The CV- η correlation confirmed that narrowing CLDs requires enhancing η , achievable through tray engineering and hydrodynamic optimization.

References

- 36 Soave, G., Gamba, S. & Pellegrini, L. A. SRK equation of state: Predicting binary interaction parameters of hydrocarbons and related compounds. *Fluid Phase Equilib.* **299**, 285-293 (2010).
- 37 Cai, T. J. & Chen, G. X. Liquid back-mixing on distillation trays. *Ind. Eng. Chem. Res.* **43**, 2590-2597 (2004).

SI, Page 18:

Fractionated degradation process and advantages

1. Reaction Process. Fractionated degradation may progress through three stages: initiation, equilibrium, and termination, evidenced by chain scission kinetic curves (**Supplementary Figure 15d**). During initiation, reaction kinetics were constrained by polymer melting rates but subsequently accelerated. This acceleration

induced pressure buildup, rapid vapor flow, and entrainment phenomena (**Supplementary Figure 5d**), producing broad chain-length distributions at the reaction start (PP-T1-1h and PP-T2-1h; **Supplementary Figures 8-9**).

In the equilibrium stage, VLE and re-degradation serve as the fundamental processes narrowing distribution, which, macroscopically, can be tuned by adjusting F#.

Vapor-Liquid Equilibrium (VLE): molecular redistribution driven by VLE is fundamental for narrowing CLDs of hydrocarbons. The VLE governs the bidirectional, thermodynamic separation of the hydrocarbon mixture on a tray. During operation, lighter hydrocarbons preferentially vaporize from the liquid phase due to the low vaporization enthalpies, when heavier hydrocarbons in the vapor phase undergo condensation, migrating downward to lower fractions. Each tray, thus, continuously strips heavy components from ascending vapor and light components from descending liquid, progressively sharpening molecular weight distributions within individual fractions. The process establishes well-defined temperature and C#_n gradient along the mass transfer pathway (**Figure 2b**).

Re-degradation: selective breakdown of heavy fractions, specifically oligomers, long-chain hydrocarbons, and macromolecular byproducts is critical for reducing hydrocarbon chain lengths and lowering the molecular weight upper boundary. Re-degradation is intrinsically linked to VLE driven descending flows. The efficiency of re-degradation was quantified by the scission number s ($s = M_0/M_t$, where M_0 represents initial molecular weight and M_t denotes post-scission molecular weight).²¹ Fractionated systems (*e.g.*, three-fraction and single-fraction reactors) enable recirculation of heavy hydrocarbons to degradation zones. In contrast, non-fractionated configurations accumulate ultra-heavy species due to inefficient or suppressed VLE descending flows (**Supplementary Table 1**).^{1,20}

Fractionation: Adjusting F# is the straightforward strategy to enhance VLE efficiency. Each additional tray establishes a discrete VLE stage, cumulatively narrowing CLDs to yield progressively sharper, more discrete C#_n bands across fractions, as evidenced by the logarithmic correlation across reactors (**Supplementary Figure 18**).

Process termination may occur through reactant depletion and competitive side reactions. During termination, diminished pressure impedes mass transfer. Constant reflux within high-temperature zones preferentially drives side reactions (**Supplementary Scheme 1 and 2**) rather than chain scission events.

Supplementary Figure 5. Technical drawing of (a) a three-fraction reactor and (b) a single-fraction reactor. (c) Schematic showing laser positions used to examine entrainment: 1 - bottom of T2, 2 - middle of T2, and 3 - middle of T1. (d) Digital images of fractionated degradation in a three-fraction reactor at the initiation stage and after equilibrium. During initiation (0-0.5 h), pronounced entrainment phenomena were evident across all trays. After equilibrium, entrainment effects were largely mitigated, persisting only minimally at the lower section of T2. (e) Pressure monitoring in the reactor at three tray levels.

Supplementary Figure 9. (a) APCI-MS analysis of PP degradation products. The spectra exhibit the molecular weights and distributions of PP degradation products in three-fraction reactors after 1 hour (PP-T1-1h and PP-T2-1h), after 5 hours (PP-T1-5h and PP-T2-5h), PP-ST-5h, and PP-NT-5h (Supplementary Tables 1 and 2). The spectrum of PP-T1-5h shows minimal signals, indicating that PP-T1-5h predominantly consists of hydrocarbons with chain lengths less than 15 ($m/z = 210$), which was reliably characterized by GC-MS (Supplementary Figure 8). A grid is used to facilitate comparison and reading across the spectra. The solvent baseline has been subtracted from the spectra. (b & c) Product carbon number distributions for PP-T1-1h and PP-T2-1h after 1-h degradation at 400 °C, estimated based on GC-MS and APCI-MS. (d) Digital images of PP-T1-5h, PP-T2-5h, and PP-ST-5h.

Supplementary Figure 18. Empirical correlations of CV with η . Logarithmic fitting of η with CLD descriptor, showing that η is a critical parameter controlling the distribution range. Data were extracted from **Supplementary Table 8** and **9**.

Comment #5: *What does having no significant signal beyond 210 m/z about the carbon number distribution mean?*

Author reply: The author deeply thanks the reviewer for the kind comment. In this study, we employed a complementary analytical approach using both GC-MS and APCI-MS to reveal the full carbon number distribution and determine the distribution upper and lower bounds. GC-MS is highly sensitive for volatile, low-molecular-weight hydrocarbons, and is therefore ideal for defining the lower boundary of the distribution. In contrast, APCI-MS is better suited for detecting low-volatility, higher-molecular-weight species, and was used to establish the upper bound of the product spectrum.

The lack of significant signal beyond m/z 210 in APCI-MS reflects that the product molar mass falls below 210. The absence of higher molecular weight species indicates that the degradation process primarily yields

light to mid-range hydrocarbons. To facilitate the understanding, we revised the manuscript accordingly as follows.

Main text, Page 6: To accurately determine the distribution and number-average molar mass (M_n), hydrocarbons were characterized by both GC-mass spectrometry (GC-MS) and atmospheric pressure chemical ionization mass spectroscopy (APCI-MS; **Supplementary Figures 8-9** and **Equations 2-3**). GC provided precise composition information for volatile mixtures, though its sensitivity was limited for low-volatility hydrocarbons due to column condensation.³⁵ Therefore, heavy fractions (boiling point > GC column temperature) were quantified using APCI-MS.¹³

Main text, Page 6: After 5h, PP-T1-5h displayed a narrowed range, C₆ to C₁₅ (CV = 0.502, C#_n = ~ C₉, **Supplementary Tables 1** and **2**). APCI-MS confirmed the distribution upper bound of PP-T1-5h terminated at C₁₅, with no detectable signals above 210 m/z (**Supplementary Figure 9**).

Comment #6: *Can the authors further explain the differences in the carbon number distribution observed for PP-ST, PP-NT, PP-T1, and PP-T2? Why did PP-ST have a narrower carbon number product distribution than PP-NT? Can the authors clarify if the pyrolysis oil obtained from PP-ST was liquid at room temperature?*

Author reply: We sincerely appreciate your thoughtful inquiry. Below, we provide a concise explanation of the differences in PP-ST-5h, PP-NT-5h, PP-T1-5h, and PP-T2-5h, with particular analysis on the CLD narrowing. To enhance manuscript accessibility, we have included detailed explanations in the Supporting Information (as noted in our response to **Comment #4**).

Differences in carbon number distribution.

The carbon number distributions exhibit a clear hierarchical trend: PP-T0, PP-T1, and PP-T2 demonstrate the narrowest distributions, followed by PP-ST with intermediate width, while PP-NT displays the broadest distribution (**Figure 1d**). This difference correlates directly with reactor configurations: (1) PP-T0, PP-T1, and PP-T2 were synthesized using a three-fraction reactor (**Supplementary Figure 5a**); (2) PP-ST was produced in a single-fraction (ST) reactor (**Supplementary Figure 5b**); (3) PP-NT originated from a modified none-fraction (NT) reactor (**Supplementary Figure 4**). Although these reactors share fundamental structural similarities (composed by a high-temperature degradation unit tandem with a cooling unit), their tray numbers differ significantly. This critical variation directly induces the size-selectivity differences, where increased tray count enables sharper fractionation, better tray separation efficiency, and thus narrower carbon distribution.

In contrast, in the NT reactor, the produced hydrocarbons were “frozen” after vaporization. Although, it may produce limited reflux (*Nat. Sustain.*, **2024**, 7, 1681-1690), the broad chamber configuration cannot stop

entrainment unless fractionated (see Supporting Discussion, “Disadvantage of the non-fraction reactors”), eventually lead to broad CLD.

Why did fractionated degradation produces narrower CLD.

As have addressed in our reply to **Comment #4** and in the revised manuscript, the narrower CLD observed in fractionated degradation products versus PP-NT-5h derive from the synergistic re-degradation and VLE, progressively classifying and separating hydrocarbon in each tray. Crucially, this classification capability is absent in NT reactors.

VLE: VLE-driven molecular redistribution is a key mechanism of the CLD narrowing. The VLE enriches ascending vapor with lighter components while concentrates descending liquid with heavier hydrocarbons. The “classifier” thus functions as a molecular filter, selectively stripping heavier hydrocarbons from rising vapors and lighter fractions from falling liquids, sharpening CLDs across trays.

Re-degradation: At the bottom tray, descending heavy fractions recirculate to the degradation unit, subjecting repeated chain scission. The scission number s can be a quantitative indicator of re-degradation. For PP with initial M_n of 87 kDa, experimental data yield $s \sim 210$ for PP-ST-5h ($M_n = 420$ Da), compared to $s \sim 130$ for PP-NT-5h ($M_n = 676$ Da), indicating that fractions recirculation enhances scission times, consequently reducing the upper boundary of the CLDs.

Mechanistically, increase in F# introduces additional trays, thereby augmenting the number of VLE stages, progressively narrowing the CLDs to produce sharper and more discrete distribution bands on each trays. Therefore, improving the F# enhances separation efficiency and depresses upper carbon limits (*e.g.*, PP-ST-5h terminating at $\sim C_{60}$ when PP-NT-5h terminating at $\sim C_{90}$).

Pyrolysis oil color. The PP-ST-5h is in liquid form at room temperature. This physical characteristic is visually demonstrated in **Supplementary Figure 9** of the Supplementary Information.

Supplementary Figure 9d. Digital images of PP-T1-5h, PP-T2-5h, and PP-ST-5h.

Main text, Page 7: In contrast, PP-ST-5h, obtained using the ST reactor **was liquid (Supplementary Figure 9d)** and exhibited a broader distribution ($C_{\#n} = \sim C_{30}$, $CV = 2.37$, ranging from C_6 to C_{60}) than PP-T1-5h and

PP-T2-5h, yet it remained narrower than PP-NT-5h (CV = 2.95, ranging from C₁₅ to C₉₀, Supplementary Figure 9a and Supplementary Table 1).

Comment #7: *How does the separation efficiency affect the decomposition of PP in terms of carbon number distribution, residue, and yield of fractions?*

Author reply: We deeply appreciate the reviewer for the insightful comment about separation efficiency (η). Indeed, in the previous manuscript, the discussion and analysis of η was insufficient. Following the comment, we re-examined our experimental data, and have expanded the discussion section to provide deeper mechanistic insights. These updates are discussed and presented in replies to **Comment #4** and **#6**. Below, we offer a brief discussion about effect of η on yields and CLD.

Effect of η on Hydrocarbon and Residue Yields. To systematically evaluate η -yield relationships across fractionated and non-fractionated systems, we recalculated separation efficiencies using an corrected O'Connell correlation (Chemical Engineering Progress, 2018, July, Predict distillation tray efficiency). For the three-fraction system, section efficiencies were determined using weight-average vapor pressure and viscosity, with geometric means applied for overall η of the reactor (Supplementary Table 9). These recalculated η were subsequently plotted against hydrocarbon and residue yields. Although the reactor apparently generates more hydrocarbons at lower η , the minor yield differences (<5 wt.%) and their standard deviations (1.5-4 %) indicate an insignificant effect of η on reaction yield (**Figure R1**). This outcome is not surprising. While higher efficiency typically facilitates component isolation and potentially improves yields, concurrent processes, particularly feedstock refluxing, may elevate resident time and induce undesirable side reactions (*e.g.*, coking, crosslinking, dehydrogenation). These competing mechanisms may result in system-dependent tradeoffs in net η -yield relationships.

Distribution. In hydrocarbon distillation, η fundamentally governs both component purity and molecular distribution width. As established by distillation principles, η typically exhibits positive correlation with theoretical tray number due to enhanced VLE staging and suppression of flooding and back mixing phenomena. This mechanistic relationship explains the significantly higher η values in three-fraction reactors (geometric average $\eta \sim 51\%$) than both single-fraction (16%) and non-fractionated systems (0.05%; Supplementary Tables 9-10). Crucially, our analysis of CV versus η (Supplementary Figure 18) revealed a logarithmic correlation across reactor configurations, demonstrating that increased η enables narrower molecular distributions. These findings empirically establish η as a quantifiable indicator of fractionation quality. Consequently, achieving narrower distributions necessitates enhancement of η (*e.g.*, advanced tray design and vapor-liquid interphase contact optimization).

Figure R1. Effect of η on reaction yields. An geographic average η was used for the three-fraction reactors.

Supplementary Figure 18. Empirical correlations of CV with η . Logarithmic fitting of η with CLD descriptor, showing that η is a critical parameter controlling the distribution range. Data were extracted from **Supplementary Table 8** and **9**.

Comment #8: *What type of polymers and additives can be found in the waste used?*

Author reply: We express our sincere appreciation to the reviewer for this significant comment. As noted, real-life waste samples exhibit considerable complexity, containing impurities and additives (**Supplementary Figures 19-22**). Recognizing the importance of quantitative characterization of real-life wastes for process insights, we have analyzed the composition of our real-life plastic waste (designated as W-heze, W1, W2, and W3) using the standardized methodologies (CJ/T 313-2009, ISO 19219) to identify polymer impurities and potential contaminants. Potential additives in the waste were evaluated based on elemental composition, with references to literature sources (*Nature*, **2025**, 643, 349; *Polym. Test.*, **2024**, 136, 10848; Hans Zweifel, *Plastics Additives Handbook*, Hanser Publications, 2009). A concise discussion of these findings is presented below. The manuscript is revised to be more comprehensive.

Potential polymer impurities. Composition characterization (**Supplementary Figures 19 and 22**) revealed biomass, primarily cellulose and lignin from plants. These biomasses originated from tags and supply chain contamination. Additionally, animal hair and rubber were identified, suggesting the presence of proteins, polyisoprene, and polyurethanes. For example, sample W3 contained 1 wt.% rubber, 1.2 wt% biomass, and 1.3 wt.% other polymers (including PET and PVC) as shown in **Supplementary Figure 22c**. Presence of PE in W-heze was experimentally confirmed. The content of PE can be estimated from alkane yields (pure PE degradation yields ~40 mol% alkanes at 400 °C; *Science*, **2023**, 381, 666). In our manuscript, APCI-MS revealed significant alkane quantities (~3.5 mol%, **Supplementary Figure 23b**), indicating ~10 wt.% PE in the W-heze PP waste.

Potential additives and others. To identify potential additives within the mixture, we employed a deductive approach correlating elemental composition with specific additive sources. For instance, post-consumer polyolefin waste from packaging and household goods contains metallic components derived from stabilizers, pigments, and fillers, including BaSO₄, TiO₂, AO-series, S-series, P-series, and EPDM (*Polym. Test.*, **2024**, 136, 10848; Hans Zweifel, *Plastics Additives Handbook*, Hanser Publications, 2009), corresponding to elements (Ca, P, Ba, Co, S, etc.) detected in our elemental analysis (**Supplementary Table 12 and Supplementary Figure 22**). A detailed analysis is updated in Supporting Discussion (SI, Page 25)

In summary, the PP waste used in our study contain polymer impurities, including PE, biomass, rubbers, PET, and PVC. Additive components were inferred based on elemental composition and degradation product characterization.

Accordingly, the manuscript has been revised to incorporate these analyses.

Main text, page 9: Compared to virgin PP plastics, real-life plastic wastes in supply chains exhibit dynamic characteristics (see Supplementary Discussion, **Supplementary Figures 19-20**). The levels of impurity (*e.g.*,

polymers, biomass, organic additives, and inorganic materials, **Supplementary Figure 21**) vary between plastic wastes (**Supplementary Figure 20, Supplementary Tables 11-12**), influencing product yields, structures, and elemental compositions (**Supplementary Figure 22 and Supplementary Table 12**). For instance, the fractionated degradation of real-life PP wastes (W1, W2, and W3) obtained from a single vendor exhibited reduced hydrocarbon yields and increased side reactions at lower purity, resulted in more internal alkene and aromatic byproducts.

Main text, Page 10: PP-STW-6h was primarily composed of α -alkenes, with minor alkanes (3.5 mol%) originated from PE (~ 10 wt.%) and biomass (1.7 wt.%), which are typical polymeric impurities in the real-life waste (**Supplementary Figures 19-23**).

SI, Page 25:

2. Complicity of real-life plastics wastes.

Real-life plastic waste exhibits greater complexity than virgin or regenerated plastics. Additive and contaminant often vary significantly between batches (**Supplementary Figure 22d**). This variability introduces uncertainty in determining precise chemical structures and concentrations within waste streams, making conventional analytical frameworks time-consuming and of limited practical value.

To characterize the composition of these waste materials, we employed standardized analytical methods based on visual examination (**Supplementary Figure 22**).^{7,8} The visual methodology is widely adopted due to its efficiency and reliability in characterizing heterogeneous materials. In this work, we categorized impurities into two categories: mixed impurities (macroscopically heterogeneous) and blended impurities (microscopically heterogeneous).

3. Composition analysis of real-life PP wastes.

Mixed impurities. Mixed impurities are foreign materials occupying distinct domains within plastic mixtures (**Supplementary Figure 20**). These macroscopic and phase-separated impurities source from contamination during transportation and are typically immiscible under ambient conditions. Compositional analysis of W3 (**Supplementary Figure 22d**) identified natural polymers and paper (biomass) as a prevalent polymeric impurities at 1.2 wt.%. Visual inspection further detected trace quantities of animal-derived materials (<0.1 wt.%, categorized as “Others”) and rubber components (1.0 wt.%). Other plastics including PET and PVC were also identified through visual inspection at minimal levels (<0.1 wt.%, categorized as “Others”). Although PE shares spectral (visual light and infrared) similarities with PP, their degradation pathways differ fundamentally. PP degradation yields predominantly alkenes, while PE degradation produces ~40 mol% alkanes.¹ Our quantification of PP-STW-6h revealed alkane formation (~3.5 mol%, **Supplementary Figure**

23b), indicating PE contamination. Using APCI spectral analysis and assuming 40% alkane yield from PE degradation, we estimated the PE fraction at ~10 wt.% (**Supplementary Figure 19c**), establishing it as the dominant polymer impurity in W-heze.

Other than polymeric components, metals emerged as significant mixed impurities. These metals may not be introduced during manufacturing, and are likely introduced through supply-chain contamination. In the W3 sample, metal content reached ~5.8 wt.%, potentially introduced through supply-chain contamination. Characterizing the precise mineral composition presents challenges due to heterogeneous distributions and particulate variability at sub-sampling scales.

Blend impurities: Blended impurities, conversely, consist of additives integrated or dissolved within materials during manufacturing, including stabilizers, pigments, antioxidants, and plasticizers.^{44,45}

To identify potential additives within the mixture, we employed a deductive approach correlating elemental composition with specific additive sources. For instance, post-consumer polyolefin waste from packaging and household goods contains metallic components derived from stabilizers, pigments, and fillers, including BaSO₄, TiO₂, AO-series, S-series, P-series, and EPDM,⁴⁴ corresponding to elements (Ca, P, Ba, Co, S, etc.) detected in our elemental analysis (**Supplementary Table 12** and **Supplementary Figure 22**). Following, potential impurities are discussed by element, with example compounds listed in parenthesis.^{44,45}

Ca primarily originates from stabilizer additives like calcium stearate, flame retardant (calcium magnesium hydroxide oxide), and pigment (CaCO₃ and calcium resinate).

Mg typically derives from mineral fillers such as talc, stabilizer (magnesium carbonate), and pigment (magnesium ferrite, magnesium sulfate).

P primarily originates from phosphite stabilizers (P-1, P-2, and P-3, triisotridecyl phosphite) and flame retardant ((2-cyclohexylphenyl) phosphite, Tris (3-chloropropyl) phosphate).

S is introduced through thiol-based antioxidants like S-1 (DSTDP), S-2, and S-3 (DLTDP). Other sources include stabilizer (methanesulfonate, pentalead tetraoxide sulfate), pigment (zinc sulfide), and antistatic agent.

Na, Al, and Si may come with each other.

Na is sourced from pigment, such as sodium docusate and ultramarine blue (Na₈Al₆Si₆O₂₄).

For Al, in addition to pigment, it may also come from flame retardant (sodium aluminate, sodium aluminosilicate, sodium aluminum phosphate), stabilizer (Al₂Ca₆O₆(SO₄)₃, aluminum hydroxide), and pigment (Al, Al₂O₃).

Si sources are vary. Other than pigment, many silicates (Pb, Mg, Zn, Zr) are plastic stabilizers.

Due to the dominance of Na, Al, and Si in lithosphere and ocean, these element can also be introduced through contamination during transportation.

Br is predominantly associated with flame retardants such as brominated phosphates, hexabromocyclododecanes, and tetrabromobisphenol A.

Ba is primarily from barium stabilizers (stearate, acetate), colorants (1-naphthalenesulfonate), monomer (barium nonylphenolate), and fillers (element or oxidate) in plastics.

Sr is commonly attributed to Sr-Zn stabilizers, strontium chromate colorants, and SrTiO₃-based electronic fillers in e-waste.

Pb is frequently introduced as stabilizers, as element lead, chloride, naphthenate, phthalate, phosphite, or oxides. Other application includes colorant (lead chromate), crosslinker, and hardener (tetraethyl lead).

Zn often comes with other metals. For example, Zn is a common element in lubricants (Al-Mg-Zn carbonate hydroxide) and pigments (Zn chromate, Zn oxide).

Zr is well-known a key element of Ziegler-Natta catalyst, and may come with Al, Zn, Ti, and Fe.

Fe is sourced from many applications. As an additive, Fe is mostly utilized as colorants (*e.g.*, iron chromate, iron manganese trioxide, and iron oxide). Yet, it can also be introduced into real-life waste by rust during transportation and mechanical wear from pulverization equipment.

Cu, a non-ferrous metal, is often used as pigment or an ingredient of pigment.

Inorganic fillers sometime constitute the most predominant additives (30-40 wt.%), added to enhance mechanical properties of regenerated plastics. For example, the filler material was observed during the fractionated degradation of W-heze at 10 kg scale, producing ~12 wt% gray solid residue and ~87 wt.% of PP-STW-6h (**Figure 2b**). PXRD analysis revealed diffraction patterns corresponding to CaCO₃ (**Supplementary Figure 23a**).

Supplementary Figure 19. Scale-up experiments using real-life plastics. (a) Schematic diagram of the scale-up reactor, featuring flanges for inlet and outlet (left), a fractionated reactor body equipped with an agitator, inspection window, and prickly column (center), and a 7-channel graphite condenser connected to a glass collector (right). Dimensions and primary components are labeled accordingly for clarity. The reactor body is constructed with a #304 stainless steel shell and includes a solid residue outlet at the base. The scale-up reactor was cleaned using ethanol before use. (b) The source and time of procurement of real-life PP wastes: images show PP wastes collected from a plastic waste trading site in Heze, Shandong Province, China, collected on June 24, 2023. The weights were measured using a platform balance (± 0.1 kg). (c) Pulverized real-life PP wastes from the collection site in (b) and its composition. The mass ratio of PE was roughly estimated based on **Supplementary Figure 23b**. (d) Compositional analysis of real-life PP wastes: images show various waste components including fibrous polymers, colored PP, metal debris, paper tags, PP composites, and biomass.

Supplementary Figure 22d. (d) Standardized purity analysis of Jieshou samples W1-W3 and their yields in laboratory-scale fractionated degradation using three-fraction reactors.⁷ The error bars represent standard deviations. The “Other” category includes minor PET plastics, PVC plastics, and undetermined components. *NOTE, PP or PE plastics are PP polymer with additives.*

Supplementary Figure 21. Examples of potential additives.^{44,45}

Supplementary Figure 23. Characterization of PP-STW-6h and solid residue. (a) PXRD pattern of solid residue and simulated pattern for CaCO_3 . Insert shows digital image of solid residue after degradation. (b) APCI-MS spectrum of PP-STW-6h, where "C xx" and "C' xx" denote alkenes and alkanes, respectively. The mass ratio of PE can be roughly estimated based on the spectrum and the following assumptions: 1. molar ratio of alkane is 3.5 mol%; 2. yields of alkane from PE degradation was assumed to be 40 mol%; 3. the average molar mass of the alkane portion is ~ 350 kg/mol; 4. the average molar mass of the alkene is ~ 350 kg/mol; 5. other impurity are inert. (c) HMBC-NMR spectrum of PP-STW-6h.

Supplementary Table 11. Potential additives in PP plastic wastes.^{44,45}

Additive Category	Primary Function	Inference basis	Potential Compounds
Antioxidants	Prevent thermal oxidation	P, S	•Hydroxyphenyl compounds: AO-13
			•Thiodiphenol: AO-27
			• Thioethers: S-1, S-2, S-3
			• Phosphites: P-1, P-2, P-3
Lubricants	Improve processing flow	Ca	• Internal: Calcium stearate
Impact Modifiers	Enhance toughness	C, S	• Elastomers: EPDM (vulcanization)
Fillers/Reinforcements	Increase rigidity, reduce cost	Ca, Mg	• Mineral fillers: CaCO ₃ , talc (20-40%)
			• Reinforcing fibers: Glass fiber (30%)
Colorants	Provide color/opacity	S, Na	•Cadmium Yellow •Cadmium Red •Ultramarine Blue
Functional Additives	Specialized properties	P, S, Br	• Flame retardants: Brominated/phosphorus: Thiobromodiphenoxyethane, TBBPA-BDBPE, Phosphorus oxide, red phosphorus, etc.

Comment #9: *Is the yield of hydrocarbons for PP-STW statistically different from the yield of hydrocarbons for PP-ST?*

Author reply: We appreciate the reviewer for pointing out the inaccuracy. Based on our experimental data, the hydrocarbon yields for PP-STW, using real-life PP waste, and PP-ST, using virgin plastics are 87% and 89%, respectively. The values are slightly different but may not be statistically different. The yield of PP-STW was measured using a platform balance which has lower accuracy (± 100 g) than the analytical balance utilized for PP-ST (± 0.0001 g). To determine the statistically difference, the data uncertainty must be calculated considering the system accuracy. Therefore, we recalculated the standard error of PP-STW using **Equation R1**.

$$\sigma = \sqrt{\sigma_{\text{data}}^2 + \sigma_{\text{balance}}^2} \quad (\text{R1})$$

The σ of PP-ST yield was calibrated to be 1.400; while σ of PP-STW was calibrated to be 2.507. The sample number are 3 for each data set. Therefore, the degree of freedom is ~ 4 . With significance value of 0.01, the critical t value can be found to be 8.6.

Using the calibrated standard error and average number, the t value can be found to be 0.75, smaller than

the critical t value of 8.6. Therefore, the yields of PP-STW and PP-ST has no significance difference.

We have corrected our inaccurate expression in the manuscript accordingly. The standard errors were updated in **Supplementary Table 1**. Samples measured by analytical balance were not corrected because of the minor effect of analytical balance on standard error.

Main text, Page 10: Fractionated degradation of W-heze generated comparable amount of hydrocarbon (PP-STW-6h, 87 wt.%), solid (12 wt.%), and gases (~ 1 wt. %) to the cases at the laboratory scale (**Figure 2c** and **Supplementary Table 1**).

Comment #10: *What is the difference between the yield of the PP-STW (87 wt.%) and the yield of liquid hydrocarbons for regenerated plastic bags (50 wt.%)? Please add standard deviation.*

Author reply: We sincerely appreciate the reviewer's kind feedback. The degradation study of regenerated PP was conceived as a parallel investigation to underscore the complexity of real-life PP waste. As highlighted in our findings, the observed yield variations were intended to demonstrate the critical impact of compositional variability in plastic wastes on reaction outcomes. About the statistical analysis, we acknowledge the reviewer's valid point. Since this particular experiment was conducted as a single trial to establish fundamental trends rather than quantify reproducibility, standard error calculations were not included in the initial manuscript.

In direct response to the reviewer valuable concerns regarding data robustness, **we have removed the regenerated PP bag from the discussion, as we have conducted more systematic and robust experiments using other real-life plastic wastes (W1, W2, W3, Supplementary Figure 22) in three-fraction reactor.** To comprehensively address **Comment #32**, we conducted systematic experiments using real-life PP wastes of varying purity levels. All materials were sourced from the same plastic circulation industry park at the same time to eliminate potential spatiotemporal effects. These investigations revealed a consistent empirical relationship between material purity and reaction outcomes, including both product yields and chemical structures. The yields for the diverse real-life plastic samples are presented in **Supplementary Figure 22d**.

Comment #11: *Can the authors verify if the C_n is shown in Table S1?*

Author reply: We sincerely appreciate your suggestion of tabulation of **Supplementary Table 1**. Since C_n is directly calculated from the M_n , we initially omitted it. We have now updated C_n values into **Supplementary Table 1**, as follows.

Supplementary Table 1. Degradation products in fractionated and non-fraction reactors.

	Yields (wt. %)	T _f (°C) ^e	T _s (°C) ^e	M _n (g/mol)	M _w (g/mol)	C# _n	C _{C=C} (mmol/g) ^d	s ^g
PP-T0-5h	2.0 (± 0.6)	35.0 (± 14)	31.0(± 0.76)	67	71	4.8	-	
PP-T1-5h ^a	53 (± 1.2)	107 (± 2.0)	65.0 (± 4.7)	129	136	9.2	5.40 (±0.11)	530
PP-T2-5h ^c	32 (± 4.0)	208 (± 15)	106 (± 2.8)	290	297	21	3.97 (±0.07)	
PP-ST-5h ^c	89 (±1.4)	215 (± 1.6)	-	420	454	30	4.54 (±0.02)	207
PP-NT-5h ^b	92 (±1.5)	-	-	676	700	48	-	128
PP-STW-6h ^{cf}	87 (± 2.5)	221 (± 12)	-	379	416	27	4.25 (±0.03)	229

^a The molecular weights were evaluated by GC using Supplementary Equation S2;

^b The molecular weights were evaluated by APCI-MS using Supplementary Equation S3;

^c The molecular weights were evaluated by combination of APCI-MS and GC-MS;

^d The alkenyl concentration was determined using Supplementary Equation S4;

^e The T_f and T_s were measured using thermocouple and IR camera, respectively.

^f The standard error was corrected with balance accuracy.

^g The average number of scission (*s*) were calculated as M_0/M_t ,²¹ where M_0 is initial molecular weight of PP (87 kDa, Supplementary Figure 15a), and M_t is M_n as calculated in Supplementary Table 1. The M_t of three-fraction reactor was estimated as weight average of M_n .

Comment #12: *The authors compared the impurities of PP-STW with the impurities of PP-T2. However, the first corresponded to the degradation of PP waste with a single fraction, while PP-T2 corresponded to the degradation of PP using three fractionation column. As shown in Figure 1, these are different processes that lead to different carbon number distributions. Then, can the authors compare the impurities of PP-STW and PP-ST?*

Author reply: We sincerely appreciate the reviewer's insightful comment and apologize for the inaccuracy. In direct response to this concerns, we conducted an elemental analysis on PP-ST-5h to enable more rigorous elemental comparisons between PP-STW-6h and PP-ST-5h. These new datasets have been incorporated into the updated Supplementary Table 12 and Figure 2d.

The results in the revised table reveal that the elemental compositions of PP-ST-5h do not differ significantly from PP-T2-5h. The level of metal and non-metal elements were significantly lower than PP-STW-6h. This consistency supports our original conclusion regarding metal impurities in real-life wastes contributing to degradation product contamination. Accordingly, we have revised the manuscript to strengthen

the scientific precision of our analyses.

Main text, Page 11: Elemental analysis revealed elemental diversity in hydrocarbons derived from real-life plastic wastes than those obtained from virgin plastics (**Figure 2d**, **Supplementary Figure 24**, and **Supplementary Table 12**, see Supplemental Discussion), induced by plastic additives (**Supplementary Figure 21**) and supply-chain contaminants.

Supplementary Table 12. Element profile of degradation products of real-life PP wastes and laboratory-grade PP via various degradation methods.^a

Elements	PP-STW-6h			PP-T2-5h		Blank ^a	PP-ST-5h		PP-W1-5h	PP-W2-5h	PP-W3-5h
Comprehensive scan quantification											
Ba	0.195	0.193		0.0311	0.0320	0.0048	0.0153	0.0148	1.07	247	502
Sr	0.159	0.205		0.0210	0.0190	0.0019	0.0018	0.0055	2.81	23.4	46.5
Na	13.2	18.5		ND	3.81	0.4137	1.13	1.13	0.570	81.1	76.0
Al	ND	7.28		0.510	0.210	0.0160	0.257	0.324	72.6	393	1798
As	0.00700	0.0500		0.0012	0.00560	ND	0.0022	0.0001	0.0600	3.91	1.22
Sb	0.0100	0.0110		0.0140	0.0440	0.0001	0.0012	0.0014	0.0800	8.20	12.9
Cd	0.00270	0.00380		0.0016	0.0330	ND	ND	0.0003	0.01	0.290	0.480
Cr	0.028	0.79		0.039	0.0850	0.0022	0.584	0.519	3.46	20.2	105
Pb	0.458	0.820		0.120	0.280	ND	0.0210	0.0342	3.36	6.22	8.82
Fe	2.08	7.55		1.81	2.06	0.0045	0.493	0.825	63.3	223	1749
Cu	0.570	0.700		0.083	0.140	0.0138	0.0355	0.0441	2.69	6.52	54.8
Zn	0.860	4.50		0.110	0.450	0.0013	0.245	0.506	10.2	80.9	104
In	0.350	0.0130		0.004	0.006	ND	0.001	0.003	0.0100	0.0100	0.0500
Standard element quantification^b											
Br	27.3	26.9	26.5	-	-	-	-	-	-	-	-
I	1.39	1.36	1.30	-	-	-	-	-	-	-	-
Ca	38.3	38.6	37.7	9.86	8.88	0.0454	0.740	1.85	1.75	57.7	62.9
Mg	3.01	2.86	2.96	0.680	0.860	0.0021	0.0560	0.0905	36.1	2.48	2.09
S	43.2	42.2	43.6	4.43	5.28	ND	ND	ND	0.0300	0.190	1.07
P	5.89	5.48	12.8	12.8	1.57	0.0032	2.63	1.67	15.9	24.5	43.5
Si	0.246	1.14	0.240	ND	ND	ND	0.0289	0.0163	0.156	2.18	8.18

^a ND = not detected; “-” = not analyzed

^b The blank sample was DI water

^c The elemental compositions were measured using standard element calibration curves (**Supplementary Figure 24**)

Figure 2d. Elemental analysis of PP-STW-6h and PP-ST-5h.

Comment #13: Can the authors comment on what type of impurities can be found in the waste used that lead to high concentrations of Ba, Sr, Pb, Fe, etc.?

Author reply: The author thanks the reviewer for this thoughtful question. As comprehensively addressed in our reply to **Comment #8**, the complexity of the real-life wastes challenged direct determination of compounds and concentrations. Therefore, we adopted a deductive methodology, which involve inferring potential compounds through elemental analysis, supplemented by thorough cross-referencing with established literature (*Nature*, 2025, 643, 349; *Polym. Test.*, 2024, 136, 10848; Hans Zweifel, *Plastics Additives Handbook*, Hanser Publications, 2009).

The origin of impurity elements can be sourced to many routes, such as contamination during transportation and additives. Some examples (Ba, Sr, Pb, Zn, Zr, Fe, and Cu) were cited from our revised supporting information as follow. More detailed discussion can be found in our reply to **Comment #8** and revised manuscript (SI, page 26).

SI, Page 27:

Ba is primarily from barium stabilizers (stearate, acetate), colorants (1-naphthalenesulfonate), monomer (barium nonylphenolate), and fillers (element or oxidate) in plastics.

Sr is commonly attributed to Sr-Zn stabilizers, strontium chromate colorants, and SrTiO₃-based electronic fillers in e-waste.

Pb is frequently introduced as stabilizers, as element lead, chloride, naphthenate, phthalate, phosphite, or oxides. Other application includes colorant (lead chromate), crosslinker, and hardener (tetraethyl lead).

Zn often comes with other metals. For example, Zn is a common element in lubricants (Al-Mg-Zn carbonate hydroxide) and pigments (Zn chromate, Zn oxide).

Zr is well-known a key element of Ziegler-Natta catalyst, and may come with Al, Zn, Ti, and Fe.

Fe is sourced from many applications. As an additive, Fe is mostly utilized as colorants (e.g., iron chromate, iron manganese trioxide, and iron oxide). Yet, it can also be introduced into real-life waste by rust during transportation and mechanical wear from pulverization equipment.

Cu, a non-ferrous metal, is often used as pigment or an ingredient of pigment.

Comment #14: *What is the typical degradation of SDBS?*

Author reply: Thank you for your valuable suggestion. The degradation of SDBS (linear alkylbenzene sulfonate) is typically mediated by microbial communities through an enzymatic process (RSC Adv., **2021**, *11*, 20303). The degradation of SDBS typically begins with the oxidation of the alkyl chain, undergoing ω -hydroxylation to produce hydroxylated alkyl products. This process is mainly catalyzed by alkane 1-monooxygenase (alkB) and cytochrome P450 enzymes, which convert alkanes into alcohols, further oxidize them into aldehydes, and eventually into carboxylic acids. These carboxylic acids then enter the β -oxidation pathway, where they are further broken down into smaller molecules, which can be utilized by microbes as a carbon source (**Supplementary Figure 25h**). The benzene ring in SDBS undergoes oxidation by oxygenases to form hydroxylated benzene derivatives. The hydroxylated benzene derivatives subsequently undergo ortho- or meta-cleavage, leading to the gradual breakdown of the benzene ring into water-soluble small molecules.

In the revised manuscript, we have discussed these degradation mechanisms and predicted the degradation behavior of plastic-derived sulfonates as follows.

Main text, Page 13: *Proteobacteria* strain dominated the sludge (> 70%, **Supplementary Figure 29**) and degraded surfactants following pseudo-first-order kinetics (**Figure 3e** and **S30**), probably through enzymatic β -oxidation and decarboxylation pathway (**Supplementary Figure 25h**).⁴²

Reference

42 Zhang, Q., Li, Y., Song, Y., Li, J. & Wang, Z. Properties of branched alcohol polyoxyethylene ether carboxylates. *J. Mol. Liq.* **258**, 34-39 (2018).

SI, Page 29:

Potential biodegradation pathway

The biodegradation pathways of PP-T1-5h-S and PP-T2-5h-S may align with alkyl sulfonate biodegradation

pathway. Taking sodium dodecylbenzenesulfonate (SDBS) as an example, the most commercially significant sulfonate surfactant, the process typically initiates with alkyl chain oxidation.⁵² This initial transformation involves ω -hydroxylation, where microbial enzymes including alkane 1-monooxygenase and cytochrome P450 systems convert terminal methyl groups into alcohols. These hydroxylated intermediates then undergo sequential oxidation: first to aldehydes, then to carboxylic acids. The resulting derivatives subsequently enter β -oxidation pathways, where they are cleaved into smaller molecular units that serve as carbon substrates for microbial metabolism (**Supplementary Figure 25h**). Concurrently, the aromatic component degradation may take two pathways. Aryl groups could undergo direct enzymatic conversion. Bacterial oxygenases hydroxylate the benzene ring, forming catechol derivatives that undergo either ortho- or meta-cleavage. Aryl groups could also be converted into benzoyl compounds. The β -oxidation consumes the alkyl chain and eventually leaves a carbonyl group on the aryl.⁵² Both processes ultimately yield water-soluble aliphatic acids, thereby significantly reducing the environmental persistence and ecotoxicity characteristic of sulfonate compounds.

Reference.

- 52 Y. Gu, Y. Qiu, X. Hua, Z. Shi, A. Li, Y. Ning, D. Liang, Critical biodegradation process of a widely used surfactant in the water environment: dodecyl benzene sulfonate (DBS), *RSC Adv.*, **2021**, *11* (33), 20303-20312.

Supplementary Figure 25. Characterization of sulfonation intermediates and products. (a) Evolution of sulfonated product structures at ambient temperature, as indicated by ^1H NMR in CDCl_3 . (b) PXRD pattern of precipitates formed in AgNO_3 solution during sulfonation and simulated pattern of AgCl. The composition of the sulfonation exhaust gases was characterized using chemical adsorption method. The exhaust gas was purged into an aqueous AgNO_3 solution, generating white precipitate. The precipitate, characterized by PXRD, displayed spectrum lines corresponding solely to AgCl, indicating that the primary component of the exhaust gas is HCl, with negligible amounts of SO_2 and SO_3 . (c-g) FTIR and XPS spectra of α -alkenes, sulfonate, and a sodium lauryl sulfate reference. The presence of S-OH and S=O signals confirmed the successful conversion of alkenes to sulfonates. In (c), the C=C signals exhibit broadening due to the formation of internal alkenes, as evidenced by HT-HMBC-NMR spectra (Figure 3a and S23). (h) A typical bio-degradation pathway of sulfonate compounds.⁵²

Comment #15: *Can the authors add the time at which this data was obtained?*

Author reply: Thank you for your suggestion. We have added the corresponding experimental time for all samples in our manuscript. We appreciate your attention to this detail. For example, products collected at T2 after 5h is labeled as PP-T2-5h; products collected in NT reactor after 5h is labeled as PP-NT-5h. We have revised the whole manuscript and highlighted these changes in blue. A revision is given below as an example.

Main Text, Page 7: GC-MS and APCI-MS analyses confirmed that PP-T1-5h, PP-T2-5h, and PP-ST-5h contained unsaturated hydrocarbons.

Comment #16: *What is the standard deviation for PP-NT?*

Author reply: We are grateful for the reviewer's meticulous attention to this vital statistical consideration. In direct response to this valuable comment, we have taken additional experimental using the NT reactor. Through rigorous analysis, the standard deviation for the PP-NT-5h was determined to be 1.5. We have accordingly updated this value in **Supplementary Table 1** to ensure accuracy.

Comment #17: *What is the difference between experiments 1, 2, and 3 in PP-STW? Why can the elements from Ba to In not be detected in Exp3 but can be detected in Exp1 and Exp2?*

Author reply: We sincerely apologize for the confusion caused by the poor tabulation of the initial **Supplementary Table 11** (now **Supplementary Table 12**). In our scale-up experiment of real-life PP wastes degradation, primary focus was directed to Ca, Mg, P, S, Br, and I, (Ca-to-I) due to their prevalence in the specific plastic waste streams. To ensure accuracy for Ca-to-I, quantification was performed using certified reference materials *via* inductively coupled plasma (ICP) analysis, when other elements were analyzed through ICP parallel scans to establish general abundance profiles.

SI, Page 7: *The concentration of Ca, Mg, P, S, Br, and I were quantified using standard materials (Tanmo Quality Control: S, Ca, Mg, P, and Br; National Nonferrous Metals: I) and the standard curve method (Supplementary Figure 24 and Supplementary Table 12).*

We have thoroughly re-tabulated **Supplementary Table 12** to clearly delineate these methodological distinctions. The Reviewer's insightful identification has improved the clarity of our work, for which we extend our most profound gratitude.

Supplementary Table 12. Element profile of degradation products of real-life PP wastes and laboratory-grade PP via various degradation methods.^a

Elements	PP-STW-6h			PP-T2-5h		Blank ^a	PP-ST-5h		PP-W1-5h	PP-W2-5h	PP-W3-5h
Comprehensive scan quantification											
Ba	0.195	0.193		0.0311	0.0320	0.0048	0.0153	0.0148	1.07	247	502
Sr	0.159	0.205		0.0210	0.0190	0.0019	0.0018	0.0055	2.81	23.4	46.5
Na	13.2	18.5		ND	3.81	0.4137	1.13	1.13	0.570	81.1	76.0
Al	ND	7.28		0.510	0.210	0.0160	0.257	0.324	72.6	393	1798
As	0.00700	0.0500		0.0012	0.00560	ND	0.0022	0.0001	0.0600	3.91	1.22
Sb	0.0100	0.0110		0.0140	0.0440	0.0001	0.0012	0.0014	0.0800	8.20	12.9
Cd	0.00270	0.00380		0.0016	0.0330	ND	ND	0.0003	0.01	0.290	0.480
Cr	0.028	0.79		0.039	0.0850	0.0022	0.584	0.519	3.46	20.2	105
Pb	0.458	0.820		0.120	0.280	ND	0.0210	0.0342	3.36	6.22	8.82
Fe	2.08	7.55		1.81	2.06	0.0045	0.493	0.825	63.3	223	1749
Cu	0.570	0.700		0.083	0.140	0.0138	0.0355	0.0441	2.69	6.52	54.8
Zn	0.860	4.50		0.110	0.450	0.0013	0.245	0.506	10.2	80.9	104
In	0.350	0.0130		0.004	0.006	ND	0.001	0.003	0.0100	0.0100	0.0500
Standard element quantification^b											
Br	27.3	26.9	26.5	-	-	-	-	-	-	-	-
I	1.39	1.36	1.30	-	-	-	-	-	-	-	-
Ca	38.3	38.6	37.7	9.86	8.88	0.0454	0.740	1.85	1.75	57.7	62.9
Mg	3.01	2.86	2.96	0.680	0.860	0.0021	0.0560	0.0905	36.1	2.48	2.09
S	43.2	42.2	43.6	4.43	5.28	ND	ND	ND	0.0300	0.190	1.07
P	5.89	5.48	12.8	12.8	1.57	0.0032	2.63	1.67	15.9	24.5	43.5
Si	0.246	1.14	0.240	ND	ND	ND	0.0289	0.0163	0.156	2.18	8.18

^a ND = not detected; "-" = not analyzed

^b The blank sample was DI water

^c The elemental compositions were measured using standard element calibration curves (**Supplementary Figure 24**)

Comment #18: *How were the heat transfer coefficients validated during the simulations? Were there any deviations observed at scale due to non-uniform heating?*

Author reply: We deeply appreciate the reviewer's constructive comment and are grateful for this opportunity to elaborate on the heat transfer coefficient employed in our simulations. The values applied to feedstock within the degradation unit were originally referred to established literature (*Science*, **2023**, 381, 666). Specifically the

research utilized stirred-tank reactor configurations for PE and PP cracking. The reliability of the cited values was further rigorously validated in our work. Please allow us to discuss them further.

Heat transfer coefficients validation.

The heat transfer coefficient is calibrated based on a rough value from literature (*Science*, **2023**, 381, 666) using more specific process conditions (stirring rate, paddle size, heat capacity, etc.) and Chilton correlation (**Equation R2**). **The relevant calculation was omitted from our original manuscript for simplicity.**

$$\frac{\frac{hD}{k} \left(\frac{\mu_j}{\mu}\right)^{0.14}}{\left(\frac{c}{k}\right)^{1/3}} = A \left(\frac{L^2 N \rho}{\mu}\right)^B \quad (\text{R2})$$

where h is heat transfer coefficient, D is reactor diameter, k is thermal conductivity, μ_j is the viscosity near the heater, μ is the viscosity of PP at 400°C, c is thermal capacity of the feedstock, A and B are constant and dependent on heating type (jacket reactor, A = 0.36, B = 2/3; internal coil reactor, A = 0.87, B = 0.62), L is length of stirrer paddle, N is stirring rate, ρ is feedstock density.

Using a coil reactor and dimension listed in **Supplementary Table 4**, we assumed viscosity ratio = 1 given minimal viscosity differences among hydrocarbons at 400°C. Other parameters appear below (**Table R1**). Since PP undergoes rapid degradation at 400°C (**Supplementary Figure 16**), the true viscosity of PP at 400°C becomes meaningless. We therefore assumed partially degraded PP as a 70 wt.% solution mixture.

Table R1. Dimension of the simulated reactor and physical properties of PP.

Parameters	Value (unit)
D	9 (ft)
L	7 (ft)
k	0.098 (p.c.u/h ft °C)
c	0.5 (p.c.u/lb °C)
N	18000 (rph)
ρ	56.1 (lb/ft ³)
μ^a	50 (lb/h ft)

^a the viscosity of partially degraded PP is assumed to be ~50 lb/h ft (20 cP), as measured in the literature at 260°C (*ACS Omega*, **2021**, 6, 32832); k, c, and ρ were cited from PP Engineering Properties datasheet by INEOS, olefins & Polymer.

The calculated heat transfer coefficient is ~110 BTU/hr ft² °C), falls within the typical range of heavy oil in industrial practices (80-180 BTU/hr ft² °C; *Fundamentals of Momentum, Heat and Mass Transfer*, 5th ed., Welty et al., John Wiley & Sons). To simplify the calculation, a h = 100 BTU/hr ft² °C was utilized.

Heat transfer coefficient difference between scale-up experiment and simulation.

The core scope of the scale-up experiment is validating molar mass and CLD control in fractionated degradation. Therefore, we did not focus on or measuring the heat transfer coefficients in the scale-up reactor. Actually, heat transfer coefficients necessarily differ between simulated and scale-up reactors, due to dimensional and operational variations (**Table R2**).

Although, the heating in our scale-up reactor is adequately uniform with electronically controlled tubular heater arrays attached to the reactor barrel (**Figure R2a-c**), producing reproducible yield and distributions (**Figure R2b & 2c**), as the reviewer have noted, heating non-uniformity is a significant factor for consideration. For example, creating hot spots by overpowering the heating mantle enhanced heat transfer, raising the reactor outlet temperature. This temperature increases at the column bottom, altered the C_n profile, yielding products with broader distributions and higher M_n than PP-T2-5h (**Figure R2d**). These effects underscore the critical need for precise temperature control in scale-up operations.

To accurately determine heat transfer coefficients, future work must precisely monitor reaction temperatures at key locations (bulk reactant, heater surface, column bottom), viscosity at different operating temperatures, and mixture density across temperature ranges. We are therefore developing an advanced reaction system at larger scale to resolve undetermined physical properties and correlations in polymer degradation in future work (**Figure R2e**).

Table R2. Comparison of the simulated reactor and scale-up reactor.

Parameters	Simulated	10-kg scale up
D	9	1.5 ft
L	7	1 ft
N	18000 rph	9000 rph
Heating method	Coil (internal heating)	Jacket (external heating)
Heat transfer coefficient	~ 109 BTU/hr ft ² ·°C	~ 18 BTU/hr ft ² ·°C

Figure R2. (a) The heating unit in the scale-up reactor. (b) the yields of all three PP-STW samples. (c) APCI-MS of the first and the second PP-STW samples. The third sample was not characterized. (d) PP degradation in three-fraction reactors with varied heat transfer through heating mantle manipulations. The insert is the digital image of reaction with enhanced heat transfer. Solid hydrocarbons dominated. (e) The blueprint of the scale-up reactor for future research. The key dimensions were blurred for commercial interest.

Comment #19: Was there any entrainment or back-mixing observed between trays that could influence CLD narrowing?

Author reply: We sincerely appreciate the reviewer's valuable insights concerning reactor hydrodynamics. Both entrainment and back-mixing represent significant phenomena that can substantially disrupt mass transfer and compromise $C\#_n$ profiles.

Specifically, entrainment is a specialized form of liquid back-mixing occurring between trays. In this process, high-velocity vapor streams transport liquid phases upward, leading to cross-tray contamination. This unintended mass transfer undermines the control over CLD, as carefully reported in our experimental observations (**Supplementary Figure 5**).

In our prototype three-fraction and single-fraction reactors, entrainment was observed. Laboratory-scale observations confirmed the entrainment during the initial degradation stage through light transmittance using a red laser (**Supplementary Figure 5d**). The red light path clearly demonstrated the entrainment phenomenon. As detailed in our supporting information (*Fractionated Degradation Process and Advantages*, SI Page 18), the entrainment directly contributes to the broad CLD of PP-T1-1h and PP-T2-1h (**Supplementary Table 1** and

Supplementary Figure 9).

Importantly, entrainment can be readily addressed through tray modifications. As predicted by the well-established Kister-Haas correlation, hole area and diameters are two critical factors affecting the hydrodynamic behavior in a column. For illustration, consider a scenario with moderate vapor flow (1 m/s) and low foam heights (0.005 m). In the three-fraction reactor, where the pore size of the tray equals the column diameter (0.02 m), calculations using the Kister-Haas correlation indicate entrainment rates of ~ 0.31 kgL/kgV, significantly exceeded the typical threshold of 0.1 kgL/kgV. Under the same hydrodynamic conditions, using sieve layers with pore diameters of 0.001 m would lower the entrainment level to 0.07 kgL/kgV.

Although no visual evidence of liquid or vapor back mixing was observed after the initial 0.5 hours, back mixing theoretically should persist in non-plug-flow reactions. The retention time distribution (RTD) analysis (**Figure R4, please see reply to Comment#21**) indicates that the reactor operates like a stir tank reactor rather than a plug-flow reactor (PFR). Consequently, the Peclet number (Pe) in the fractionated degradation process should be finite (partial back mixing). The slight tailing observed in the RTD curves can be attributed to back mixing phenomena. The phenomena of back mixing and entrainment throughout the reaction period collectively disrupt the mass flow, slow down the VLE, and lead to a broadening of the CLD. However, this broadening induced by entrainment and back mixing can be regulated through molecular size sieving by the "classifier" that separate heavy and light hydrocarbons through VLE and re-degradation processes, progressively refining the $C\#_n$ profile and narrowing the CLD. As a result, the CLD showed significant narrowing from $CV \sim 2.5$ to ~ 0.5 in 5 h (**Supplementary Table 1**).

The causes of back mixing and entrainment, along with other non-idealities, are discussed in detail in our revised manuscript and in our reply to **Comment #28**. Additionally, we have discussed potential solutions for their mitigation.

Comment #20: *What was the Murphree tray efficiency assumed or measured for the ASPEN simulations?*

Author reply: We sincerely appreciate the reviewer's insightful inquiry regarding the Murphree tray efficiency parameterization in our simulation. In the revised Manuscript, we have enhanced methodological description by explicitly specifying our assumption with its technical rationale. A sensitivity analysis and detailed discussion are provided below.

While we fully acknowledge the critical importance of accurate efficiency projections for simulation. However, referential η for fractionated degradation in industrial practice is absent like other newly proposed processes. The laboratory-determined η (**Supplementary Table 9**) should be lower than the actual η in

industrial practices due to the simplified channel configuration (**Supplementary Figures 4 and 5**) for manufacturing viability. The vapor-liquid contact in these reactors are limited, leading to low η .

Given these constraints, we adopted a simplification of 100% tray efficiency for process design and conceptual understanding. We wish to emphasize that this value remains reasonably aligned with typical tray efficiencies (70-90%) observed in commercial systems for hydrocarbons distillation (*AICHE*, 2010, 56, 2323).

To further strengthen the reliability of our assumption, we conducted sensitivity analyses using tray efficiency values of 100% and 70% (**Figure R3**). The resulting vapor-liquid composition profiles exhibited only marginal deviations (<1%), with negligible impact on product specifications at varied stage numbers. Actually, this insensitivity to efficiency variations at high-end is expected. At high η , stages operate near theoretical equilibrium. Minor efficiency drops (e.g., 95% to 85%) rarely alter product purity significantly. Therefore, our assumption of 100% efficiency is valid.

SI, Page 12: Particularly, given the absence of industrial referential efficiency data for fractionated degradation, we assumed $\eta = 100\%$, aligning with near-optimal industrial operation of typical oil refinery (70 - 90%).

Figure R3. Separation effect of a hydrocarbon mixture at 100% and 70% Murphree efficiency.

Comment #21: Was residence time distribution (RTD) analyzed? If so, did the setup approximate plug flow or was there evidence of tailing/mixing?

Author reply: We are deeply grateful for the reviewer's thoughtful question. Residence time distribution (RTD) analysis represents a valuable approach for examining fluid dynamics, non-ideality, processing capabilities, and other reactor system behaviors. In our study, RTD analysis was performed to deepen our understanding, though not included in the original manuscript for the following reasons:

- (1) the presented data sufficiently demonstrate the viability of our method for upcycling PP into surfactants comparable to commercial alternatives;
- (2) the high degradation temperature (400°C) presents significant challenges in identifying stable organic tracer with structural similarity to long-chain hydrocarbons;
- (3) We utilized 1, 2, 4, 5-tetrachlorobenzene (TCB) for the RTD analysis. While TCB provided measurable data, we acknowledge its limitation due to structural dissimilarity to long-chain hydrocarbons.

Detailed discussion of the limitations and RTD data are presented below.

Identifying suitable tracers for RTD analysis during fractionated polymer degradation presents particular challenges. Ideal tracers should similar to the physicochemical properties of long-chain hydrocarbons while maintaining chemical inertness. Yet under polymer thermal degradation conditions, marked by high temperatures (400°C) and radical reactions (**Supplementary Scheme 1**), conventional candidates (long-chain hydrocarbons, halogenated compounds, certain aromatics) are often unstable. Studies demonstrate polytetrafluoroethylene (PTFE) begins degrading near 300°C (*Vacuum*, **1968**, *18*, 437), while polyfluoroalkyl substances (PFS) exhibit only ~4-minute half-lives at 400°C (*Environ. Sci. Technol.*, **2024**, *58*(50), 22417-22430). Chlorinated long-chain hydrocarbons can degrade at temperatures as low as 150°C (*Polymers*, **2019**, *11*, 2080). At our experimental temperatures, analogous organic molecules suffer to degradation, consuming tracers and triggering side reactions. Consequently, unstable tracers distort RTD curves and diminish signal intensity.

Halogenated benzenes generally demonstrate superior thermal stability compared to PFS or long-chain hydrocarbons. Tetrachlorobenzene, for instance, exhibits decomposition temperatures approaching 800 °C (Serban C. Moldoveanu, *Pyrolysis of Organic Molecules*, Ch. 3). Based on this documented stability, we selected TCB as the RTD tracer. We should note, however, that TCB still displays intermolecular interactions through polar forces and π - π stacking. In vapor-phase mixtures with alkanes/alkenes, this self-association tendency promotes non-ideal behavior, manifesting as positive deviations from Raoult's law relative to hydrocarbon mixtures. Such deviations may consequently impact RTD interpretation. Therefore, we took RTD analysis with TCB as a preliminary approach to understand the fractionated degradation.

Method: For the RTD analysis, the tracer (0.1 g) was thoroughly blended with PP powder (10.0 g) using

mortar and pestle grinding. Residual powder on the mortar and pestle were carefully collected and loaded into the reactor. During degradation, samples were collected every 15 minutes, with TCB levels on trays quantified by GC with calibration curves. The resulting distribution patterns at T0, T1, and T2 trays are presented in **Figure R4**.

Results: The RTD curves revealed rapid TCB abundance increase within the first 30 minutes, indicating strong entrainment and back mixing at the initiation stage, as supported by pressure build up and visual observation (**Supplementary Figure 5**). As discussed in reply to **Comment #4** and **#19**, this back mixing burst heavy hydrocarbon to the upper trays, causing initial broadening of PP-T1-1h and -T2-1h. The gradual decay after the peak indicate the end of initiation (0-0.5 h), aligning with the kinetic curves (**Supplementary Figure 22a**). The re-degradation and VLE processes became more prevailing, progressively fixing the CLD distribution disturbed by liquid back mixing. After plateaued, total concentrations of TCB at all trays are clearly lower than the peak level at 15 min, suggesting majority of the TCB are trapped in the degradation unit (since TCB is thermally stable, the tracer degradation can be excluded from consideration). Some minor fluctuation were observed, indicating presence of weak parallel flow paths due to reactor or operation non-ideality. In general, the concentration RTD curves showed exponential decay patterns, aligning closely with stir-tank reactor characteristics. Consequently, we adopted the stir-tank BatchR model in ASPEN.

Figure R4. RTD curves at each trays using TCB as the tracer.

Comment #22: *Could trace metals (Fe, Zn, Ca) catalyze unwanted side reactions or impact sulfonation yield?*

Author reply: We are deeply grateful to the reviewer for raising this point. The presence of metal impurity could pose challenges for downstream sulfonation/hydrolysis processes and subsequent product applications, as trace metals may catalyze unintended side reactions and compromise final product quality. To our knowledge, however, the mechanistic role of metals in SO₃-mediated sulfonation and post-processing remains unexplored in literature. This gap likely stems from two key factors: (1) the extreme reactivity of SO₃ complicates precise mechanistic studies, and (2) industrial feedstock specifications typically restrict metal impurities to below-ppm levels, limiting value of investigation. In direct response to the valuable comment, we have expanded our discussion to address potential effects of metals during sulfonation/hydrolysis, based on fundamental reaction principles and established sulfur chemistry literature.

In principle, the electronegative oxygen atoms in SO₃ create an electron-deficient sulfur center, making the O on SO₃ more susceptible to interactions with cations. Therefore, it is generally believed that in the presence of H₂SO₄ or trace water, SO₃ readily forms SO₃H⁺. Analogously, metallic cations can coordinate SO₃ to generate stable crystalline complexes (*Dalton Trans.*, **2011**, 40, 1209). At material interfaces, SO₃ binds strongly to metal oxides, forming thermodynamically stable structures (*J. Phys. Chem. A*, **2023**, 127, 9541-9549). On metal surfaces, adsorption can even induce S-O bond cleavage, reducing SO₃ to SO₂ and sulfide (*Surf. Sci.*, **1995**, 343, 211). Collectively, these interactions alter SO₃ electron structures, either stabilizing or destabilizing the molecule, and redirecting the reaction pathways. A notable example is inhibition of SO₃ self-polymerization by adding boron (<1.5 wt.%, US Patent 2,458,718), showcasing how electron-deficient centers modulate SO₃ reactivity.

Based on the aforementioned facts and analysis, we anticipate altered reaction kinetics when olefins are contaminated with impurity elements. The presence of metals and metal oxides, combined with the introduction of SO₃ at stoichiometric ratios, would lead to SO₃ absorption and reduction, thereby decreasing the SO₃ concentration and reducing the yield of sulfonates. The generation of SO₂, sulfides, and other sulfur-containing byproducts would result in malodorous products and potentially cause secondary water pollution. Additionally, if SO₃ becomes incorporated into an inorganic framework and coordinates with metals, the reactivity and yields of the sulfonation reaction would be compromised. Conversely, the presence of strongly electron-deficient elements that interact with the oxygen atoms could make the central sulfur atom be more electron-deficient, thereby enhancing the reactivity of SO₃.

Additionally, metal ions may precipitate sulfonates during neutralization, where agglomeration effects might influence product solubility. For instance, Ca exhibits a strong affinity to sulfonate groups (*Chem. Mater.*, **2016**, 28, 6276), forming stable alkaline salts, a chemistry leveraged in ion-exchange resins for deionized water production. In our experiments, virgin PP contains relatively low levels of impurity metals. Consequently, we

observed minimal evidence of coking, product precipitation, or color deterioration in the PP-T2-S and PP-T1-S samples. However, when using PP-STW, the resulting PP-STW-S exhibited slightly darker coloration, which can be attributed to the effect of metal impurities. Due to the metallic cations, the sulfonation reaction might be accelerated, induced minor coking and coloration.

We have expanded the SI to address potential side reactions during PP degradation and the sulfonation, with further discussion provided in our reply to **Comment #29**.

Comment #23: *Was a sensitivity analysis run on the vapor-liquid equilibrium parameters? What assumptions were made regarding phase behavior?*

Author reply: Sensitivity analysis is a key ASPEN tool that systematically varies manipulated inputs to quantify their impact on response outputs, identifying critical variables, elucidating process behavior, enabling optimization, and supporting risk assessment. In this work, sensitivity analysis of VLE parameters, specifically Peng-Robinson (PR) binary interaction parameters (BIPs), was omitted due to their negligible effect on the olefin mixture system in our work. An illustrative sensitivity analysis is given below, with detailed ASPEN simulation discussion.

For fractionated degradation modeling, thermodynamic BIPs (K_{ij}) are paramount, directly quantifying deviations from Raoult's Law. By varying K_{ij} within feasible ranges while monitoring output variations, the robustness can be assessed. However, ASPEN database lacks BIPs for most long-chain hydrocarbons. We therefore conducted data regression and thermodynamic consistency analyses in the “**Step 1: Define components and calculation methods**” to predict reliable BIPs before simulation.

Thermodynamic consistency of VLE data is a prerequisite for reliable regression analysis; inconsistent points must be excluded to avoid erroneous simulations. To validate consistency, we used experimental VLE data covering the operational temperature and pressure ranges. Each data point includes temperature, pressure, liquid phase composition (x), and vapor phase composition (y) (**Table R3**).

We initiated thermodynamic consistency tests by defining pure components and inputting experimental data (**Table R3**) into ASPEN. The consistency assessment employed NRTL or Wilson methods to calculate liquid phase activity coefficients (γ_i). Output data were evaluated against compliance criteria. Due to the pairwise analysis for thermodynamic consistency testing, results would yield 141 figures of interaction pairs, prohibitively large for display. Consequently, our prior submission omitted full results. Below, a C6-C7 T-XY plot (**Figure R5**, screen shot of ASPEN software) is offered, exemplifying satisfactory consistency.

Figure R5. Screenshot of thermodynamic consistency of gas-liquid equilibrium data.

After confirming thermodynamic consistency of the experimental VLE data, we initiated ASPEN physical property regression module to obtain missing BIPs for the PR EOS. We selected the PR-BM method (Peng-Robinson with Boston-Mathias alpha function) and regressed the K_{ij} parameter, the most critical BIP for PR. The regression output (**Table R4**) was accepted and imported into the PR binary parameter library for subsequent simulations. Upon convergence of thermodynamic consistency calculations, we entered the sensitivity module. Regression indicated most BIPs fell between 0 and 0.2, so sensitivity bounds were set to 0 (lower), 0.2 (upper). With response variables defined, the sensitivity analysis of a three-fraction system was conducted. Below, we used the K_{ij} s for PP-T0-5h, -T1-5h and T2-5h to probe the sensitivity of distributions to VLE parameter change.

Sensitivity analysis for PP-T0-5h reveals that varying K_{ij} from 0 to 0.2 minimally impacts product molar yield. Molar fractions of C3–C7 remain nearly constant, demonstrating robustness in the fractionated degradation simulation. This K_{ij} insensitivity was also confirmed to PP-T1-5h and PP-T2-5h (**Figure R6**). Distributions for all cases show negligible change when K_{ij} values are set to 0.1 and 0.2, with most CV variations consistently below 1% (**Figure R6**).

The sensitivity analysis above confirms negligible impact of K_{ij} variation on product CLDs. Actually, such insensitivity is understandable for olefin mixture systems, for their structural similarity among olefinic chemicals and weak intermolecular interactions. Consequently, no additional sensitivity analysis was performed. Simulations using the recommended property method, PR model, satisfied reaction separation requirements.

Assumptions on Phase Behavior

Following assumptions were made:

1. The VLE system used in this work is a mixture of light hydrocarbons (low-carbon-number alkanes and low-carbon-number alkenes) at pressures < 2 atm;

2. The interactions between hydrocarbons are weak;
3. The molecular structures of the hydrocarbons are similar;
4. The vapor-liquid system in this work is close to the components in the petroleum refining process, allowing processing using the polymer system PC-SAFT and the oil-gas system.

Nonetheless, as have shown in the analysis, minor deviation was observed. To correct such deviation for optimal accuracy, future research should focus on more accurate component characterization for accurate regression. **The VLE deviation is discussed as a non-ideality in the Supporting Discussion with potential solution discussed for future process optimization (see Discussion of Non-ideality).**

Table R3. VLE experimental data.

Tray	C#	T (°C)	P (atm)	Y mol%	X mol%
T1	C6	108	1	55.20	8.96
	C7	108	1	24.25	3.81
	C8	108	1	11.01	9.81
	C9	108	1	5.11	52.85
	C10	108	1	2.39	4.85
	C11	108	1	1.12	4.45
	C12	108	1	0.48	8.05
	C13	108	1	0.24	3.01
	C14	108	1	0.12	0.15
	C15	108	1	0.06	3.33
T2	C15	208	1	45.54	9.80
	C16	208	1	28.54	5.85
	C18	208	1	11.21	16.20
	C19	208	1	7.08	8.67
	C21	208	1	2.83	15.85
	C22	208	1	1.80	8.66
	C23	208	1	1.15	12.84
	C24	208	1	0.73	12.48
	C25	208	1	0.49	2.76
	C26	208	1	0.30	2.67
	C27	208	1	0.20	3.29
	C28	208	1	0.12	0.93

	T0		T1		T2	
	CV	variation	CV	variation	CV	variation
Unchanged K_{ij}	0.287	0	0.414	0	0.554	0
$K_{ij} = 0.1$	0.287	0	0.414	0	0.552	0.4%
$K_{ij} = 0.2$	0.288	0.3%	0.387	6.5%	0.554	0

Figure R6. Effect of binary interaction parameters change on carbon number.

Table R4. BIPs regression data.

Before Regression (30 pairs)			After Regression (141 pairs)														
i	j	K_{ajj}	i	j	K_{ajj}	i	j	K_{ajj}	i	j	K_{ajj}	i	j	K_{ajj}	i	j	K_{ajj}
C3	C4	0.0033	C3	C4	-0.0297	C6	C9	-0.3250	C9	C17	-0.00261	C15	C23	-0.00868	C21	C25	0.7425
C3	C5	0.0267	C3	C5	-0.2403	C6	C13	0.5214	C10	C11	0.8418	C15	C24	0.6644	C21	C26	0.8777
C3	C6	0.0007	C3	C6	-0.0063	C6	C14	0.4005	C10	C13	0.5704	C15	C26	0.02764	C21	C27	0.6841
C3	C7	0.0056	C3	C7	0.0616	C6	C15	0.3026	C10	C14	0.1291	C16	C18	1.247	C22	C23	0.8264
C3	C10	0	C3	C10	0	C6	C16	-0.3643	C10	C15	0.3348	C16	C21	-0.6645	C22	C24	-0.350
C4	C5	0.0174	C4	C5	-0.1566	C6	C17	0.2603	C10	C16	-0.4808	C16	C22	2.241	C22	C25	-1.860
C4	C6	-0.0056	C4	C6	-0.0616	C7	C8	0.5351	C10	C17	0.3257	C16	C23	-1.383	C22	C26	-2.021
C4	C7	0.0033	C4	C7	0.0363	C7	C10	1.000	C11	C12	0.6180	C16	C24	-0.3841	C22	C27	-0.0774
C4	C10	0.0078	C4	C10	0.0078	C7	C11	-0.0164	C11	C13	0.1168	C16	C26	3.327	C22	C28	-3.398
C5	C7	0.0074	C5	C7	0.0814	C7	C12	0.6466	C11	C14	0.3696	C16	C27	-3.426	C23	C24	0.2812
C5	C8	0	C5	C8	0	C7	C13	0.1090	C11	C15	0.05728	C18	C19	0.4995	C23	C25	-0.8925
C6	C7	-0.0078	C6	C7	0.2202	C7	C14	1.104	C11	C16	-0.3988	C18	C21	0.2679	C23	C26	-0.4368
C3	C9	-0.0160	C3	C9	-0.0160	C7	C15	-0.3095	C11	C17	0.2961	C18	C22	0.02318	C23	C27	0.6172
C3	C16	0.01461	C3	C16	0.01461	C7	C16	-0.7024	C12	C13	0.3528	C18	C23	0.4271	C24	C25	-0.5508
C3	C28	-0.1098	C3	C28	-0.1098	C7	C17	0.3739	C12	C14	0.1548	C18	C24	0.2067	C24	C26	1.159
C4	C8	0.00841	C4	C8	0.00841	C8	C9	-0.3294	C12	C15	0.2714	C18	C25	-0.1921	C24	C27	-0.2490
C4	C9	-0.002653	C4	C9	-0.00265	C8	C11	0.5502	C12	C16	-0.1564	C18	C26	1.465	C24	C28	6.168
C4	C14	0.001585	C4	C14	0.001585	C8	C13	0.3187	C12	C17	0.1452	C18	C27	-0.00151	C25	C26	-2.094
C5	C6	0.0003009	C5	C6	-0.00270	C8	C14	0.4457	C13	C14	0.4479	C19	C24	0.1320	C25	C27	0.7445
C5	C10	0.006362	C5	C10	0.006362	C8	C15	0.3000	C13	C15	-0.06110	C19	C25	0.1000	C25	C28	-2.156
C5	C16	-0.007109	C5	C16	-0.00710	C8	C16	-0.2524	C13	C16	-0.7808	C19	C26	1.120	C26	C27	0.5600
C5	C28	-0.055283	C5	C28	-0.05528	C8	C17	0.1883	C13	C17	0.3004	C19	C27	0.3676			
C6	C10	-0.003281	C6	C10	0.02953	C9	C10	-0.1015	C14	C15	0.2227	C19	C28	-11.560			
C6	C11	-0.003593	C6	C11	-0.01781	C9	C11	-0.1947	C14	C16	-0.9143	C21	C22	0.6969			
C6	C12	-0.004697	C6	C12	0.04172	C9	C12	-0.1414	C15	C17	0.07425	C21	C23	-0.2521			
C6	C28	-0.06598	C6	C28	-0.06598	C9	C13	-0.1193	C16	C17	-1.502	C21	C24	0.5271			

C7	C9	0.006993	C7	C9	-0.06088	C9	C14	0.04252	C15	C18	-0.6258	C18	C28	4.481			
C8	C10	-0.005396	C8	C10	0.04654	C9	C15	-0.1296	C15	C19	-1.757	C19	C21	0.2739			
C8	C12	-0.008887	C8	C12	0.06143	C9	C16	-0.0593	C15	C21	-0.7262	C19	C22	1.313			
C10	C12	-0.004413	C10	C12	0.03075	C27	C28	0.05759	C15	C22	1.164	C19	C23	0.8550			

Comment #24: *Were any oligomerization or cross-linking side products detected at higher residence times?*

Author reply: The authors thank the reviewer for highlighting the critical point regarding olefin side reactions. Under the original reaction time, polymerization, cross-linking, and coking side reactions were not evident. Following the comment by the reviewer, we extended the times to 24 hours to enhance the potential for oligomerization and cross-linking side products.

Polymerization side reactions are an inherent characteristic of polyolefin degradation, consistent with the fundamental principles of polymerization. As monomer conversion increases, the polymerization rate decreases due to limited diffusion of monomers to active sites, such as radical or ionic centers. During depolymerization, the viscosity of the mixture decreases while monomer concentration increases, leading to a rebound in polymerization rates that competes with the degradation reaction. Kinetic analysis revealed the competition between polymerization and chain scission. The rapid chain scission dominates the initial 4 hours before plateauing due to polymerization (**Supplementary Figure 22a**). Polymerization is expected to dominate at temperatures sufficiently high to promote chain growth but below the PP degradation temperatures (*e.g.*, temperatures at T2 and T1). To assess potential polymerization, we conducted an extended fractionated degradation experiment and a simple liquid-flow test using PP-T2s, comparing the viscosity of the liquid fraction after 5 and 24 h as a proxy for molar mass differences. Although GPC is typically used for polymer and oligomer characterizations, partially cross-linked samples can clog GPC columns. Therefore, we did not perform GPC for these materials.

By extending the degradation period to 24 h, the product in the T2 exhibited significantly increased viscosity in a flowing test (**Supplementary Figure 22c**). Dropping PP-T2-24h and -5h using an automatic pipette (100 μ L) at the same time, after 3 s, PP-T2-5h flowed to the termini, while PP-T2-24h stayed near the start line. This reduced flowability indicating olefin polymerization. Oligomers and potentially partially crosslinked network could have formed.

Supplementary Figure 22c. Flow tests of PP-T2-5h and PP-T2-24h on silicone gel plate. PP-T2-24h shows yellowish color and higher viscosity than PP-T2-5h, suggesting polymerization.

Comment #25: *Was there fouling or residue accumulation in the tray or collector units over repeated runs?*

Author reply: In our repeated experiments (**Figure R7**), the accumulation of residual carbon was observed exclusively in the degradation zone, with no evidence of coke or fouling on the trays. The channel configuration of the three-fraction reactor provides a substantial passage room for vapor and liquid flow, effectively preventing the accumulation of products or residues on the trays. Additionally, the tray temperature ($\sim 200^{\circ}\text{C}$) is insufficient to induce coking. Consequently, after five parallel runs, the fractionated region remained white, while the degradation unit turned black.

Figure R7. Digital picture of three-fraction reactor after 5 repeated runs.

Comment #26: *How does reactor performance change over time with waste PP (e.g., increased pressure drop, reduced vapor flow)?*

Author reply: We express our sincere gratitude for the reviewer's insightful question and apologize for the oversight. In response to this suggestion, we observed pressure changes during a fractionated degradation process.

Our experimental data indicate an initial pressure burst followed by a plateau across the trays. The T2 exhibited a more pronounced pressure increase, reaching 0.37 kPa, before declining to ~ 0.1 kPa. T1 and T0, operating at relatively lower pressures, displayed similar trends. Although the compact design small size of our apparatus precluded direct measurement of vapor flow, vapor flow is correlated with reactor pressure, as correlated by the established fluid dynamics principles, such as Reed-Fenske relationship. Thus, the vapor flow patterns in our system should analogous to the observed pressure changes.

The observed fluctuations can be explained as follows, aligning with our response to **Comment #4, #21, #19**, and the revised SI. The fractionated degradation process progresses through three stages: initiation, equilibrium, and termination. During the initiation stage, as the plastic approaches degradation temperatures, the polymer degrades rapidly, generating hydrocarbon fragments. The accumulation of these volatiles in the degradation unit builds up pressure, causing an initial burst of vapor. The fast vapor flow subsequently carries liquid to the upper trays, inducing back-mixing and entrainment phenomena captured by RTD analysis (**Figure R4**). Subsequently, VLE and re-degradation processes retain most of the liquid in the trays, thereby reducing the pressure and vapor flow.

The determined pressure data is included in the revised manuscript as supporting to the entrainment and reactor hydrodynamics.

Main text, Page 5: Strong entrainment was observed during the initial stage (0-0.5 h) due to reactor pressure buildup, which rapidly declined after ~0.5 h, leaving only minimal entrainment near the base of T2. (**Supplementary Figure 5**).

SI, Page 1:

1. Reaction temperature calibration and pressure monitoring.

The reaction temperature of reactors with three-, single-, and none-fraction were calibrated using a thermocouple thermometer. Specifically, the thermocouple probe was placed at the bottom of the reactor (**Supplementary Figures 4 and 5**). The heating mantle temperature was gradually increased, and the thermocouple readings were monitored until the temperature stabilized at the target value for 30 minutes. The reactor pressure during the reaction was directly measured using a digital pressure meter equipped with needle probes.

Supplementary Figure 5e. Pressure monitoring in the reactor at three tray levels.

Comment #27: *Was external heat loss accounted for in the ASPEN simulation?*

Author reply: In conducting our ASPEN simulations, we intentionally excluded heat loss for simplicity given its minimal contribution to the overall heat duty in a properly designed and insulated equipment. This simplification is well-justified by national and international standards (GB/T 4272-2024, ISO 12241), requiring heat loss of $<145 \text{ W/m}^2$ for industrial equipment operating at 400°C . As documented in **Supplementary Table 4**, the reactor total surface area is $\sim 25 \text{ m}^2$, resulting in a heat loss of $\sim 14 \text{ MJ/h}$, representing merely 0.6% of the total heat duty. Therefore, when equipment is properly installed, designed, and insulated with qualified material and company, heat dissipation through reactor surfaces demonstrates minor impact on process environmental footprint and operational economics. To enhance clarity, we've revised the caption of **Supplementary Table 4**.

SI, Page 51: Supplementary Table 4: ^d To simplify the TEA and LCA, heat loss was not included in the calculations. With GB/T 4272-compliant equipment, the actual impact of heat loss is negligible ($< 1\%$).

Comment #28: *No discussion of non-idealities, such as entrainment, foaming, or pressure drops.*

Author reply: Thank you for highlighting this critical oversight of potential non-idealities. In the revised manuscript, the potential non-idealities are discussed with possible solutions.

SI, Page 22:

Discussion of non-ideality

The proposed fractionated degradation demonstrated favorable size selectivity at both 10-gram and 10-kg scales. However, due to the complex and often unpredictable composition of real-life wastes, non-idealities are anticipated in pilot-scale experiments and industrial production. Consequently, potential non-idealities are outlined and discussed below. Future research targeting system optimization, modification, and pilot testing should consider these non-idealities to achieve optimal performance.

1. Thermodynamic non-Idealities.

a. Azeotrope formation. Azeotrope forms when intermolecular forces (*e.g.*, hydrogen bonding) create thermodynamic non-idealities in vapor-liquid equilibrium. These molecular interactions vary between compounds, resulting in deviations from Raoult's law that produce either minimum- or maximum-boiling azeotropes. Azeotrope systems establish barriers where vapor and liquid phases share identical compositions, preventing further separation through conventional distillation. Mixtures with divergent properties, such as benzene/toluene, benzene/cyclohexane, and water/methanol, are susceptible to form azeotrope.

Fractionated degradation of PP primarily yields α -olefins with similar molecular architectures. Ideally, mixtures composed exclusively of linear molecules would exhibit uniform intermolecular interactions consistent with Raoult's law, avoiding azeotropic phenomena and enabling relatively ideal component separation with sufficient theoretical plate. However, the ideality could be compromised by side products and impurities (**Supplementary Figure 20**), where the introduction of cyclic structures, oxygen contamination, or other impurities, may promote azeotrope or near-azeotrope and reduces separation efficiency. Typically, achieving complete separation of all components in near-azeotropic systems ($\alpha \sim 1$) requires operational adjustments such as increasing stage number, adjusting reflux ratios, installing de-entrainment devices, or implementing other compensatory measures. For true azeotropes ($\alpha = 1$), alternative approaches including impurity pre-separation, pressure-swing distillation, or extractive distillation may become necessary. Critically, the effect of azeotrope on fractionated degradation is currently unquantifiable due to thermodynamic complexity of real-life plastic waste and their degradation products (see discussion "*Characteristic of real-life plastic waste*"). Missing VLE data of fractionated degradation product from varied waste sources is a critical gap. The knowledge gap necessitates focus on chemical properties prediction or measurement, and will be discussed in the section of VLE deviation.

b. Foaming. Foaming occurs when trace surfactants or other polar impurities disrupt surface tension and viscosity. Vapor is trapped in liquid, leading to entrainment and potential flooding. While foaming is common in practices like crude oil distillation and surfactants drying, foaming remains unlikely in operation of pure hydrocarbon mixtures. Specifically, for the fractionated degradation of virgin PP and real-life wastes (W-heze, W1, W2, and W3), no foaming was observed due to low polymer melt viscosity at 400 °C and minor polar contamination after proper cleaning procedure (**Supplementary Figure 2a**). However, polyolefins are frequently used in detergent containers. Improperly cleaned feedstock or accidental supply-chain contamination

may introduce active species. Mitigation strategies include thorough cleaning, adding antifoam reagents, and operating degradation with controlled heating rates to moderate vapor flow.

c. VLE deviation. VLE deviations occur when experimental or predicted phase behavior diverges from actual performance, primarily due to incorrect assumptions or VLE data. Modeling approximations (*e.g.*, Wilson, NRTL, and UNIQUAC equations) require accurate parameters that often unavailable or inaccurate for innovative systems. While recent Aspen Plus V14 updates incorporated polymer pyrolysis databases, viscosity, thermodynamic, degradation kinetics, multi-component interaction parameters of polymers remain absent. Consequently, determining VLE parameters remains a significant challenge for industrial process design and simulation.

Future research aiming to optimize or modify the fractionated degradation system for industrial applications should prioritize minimizing VLE deviations. Regression-based prediction methods offer a effective pathway for obtaining reliable VLE data. By precisely determining key liquid and vapor molar ratio across temperature and pressure ranges, the physical properties can be fitted, either manually or through automated regression tools in ASPEN. In our simulations, the miss BIP parameters were regressed using data in **Supplementary Figures 9 and 10**. In addition, missing VLE parameters can also be estimated through group contribution methods.^{35,36} The group contribution methods enable parameter determination via manual or software computation without experiment. However, group contribution methods typically deliver lower accuracy than experiments or regression fitting, though it remains valuable for system assessments when the other techniques are unavailable. Ultimately, for practical implementation, regression-derived and simulation-based parameters should not be accepted without validation against actual pilot-scale operating records where available.

2. Equipment & Hydrodynamic Non-Ideality.

a. Entrainment. Entrainment occurs when vapor velocity lifts liquid droplets or foam from a lower to the upper tray. This phenomenon arises when vapor energy overcomes gravitational forces and liquid film surface tension, primarily driven by high vapor flow rates and pressure gradient. The upward flow contaminates streams in the upper trays. Consequences include broadened distribution, off-specification products, reduced separation performance, higher heat duty, and even column flooding. In the fractionated degradation of polymer, entrainment poses the critical threat to product uniformity. While largely mitigated in our conceptual design after equilibrium (**Supplementary Figure 5**), the entrainment during system initiation caused distribution broadening (PP-T1-1h and PP-T2-1h; **Supplementary Figures 8-9**).

To suppress entrainment in fractionated degradation systems, the following design and operational modifications could be helpful:

- (1). increase tray spacing and expand reactor diameter to reduce vapor velocity;
- (2). install de-entrainment plate above output trays or within vapor transfer direction;³⁷
- (3). choose proper tray types that compatible with expected vapor-liquid loading ranges;
- (4). for foaming systems, one may extend liquid flow paths and incorporate anti-foam devices;
- (5). reduce reboiler duty and control heating rate to moderate vapor generation rates;
- (6). introduce anti-foam reagents to suppress bubble formation (need compatibility with process to prevent contamination).

b. High pressure drop. Pressure arises from frictional resistance during vapor flowing through trays. Factors causing pressure buildup include equipment flaws (improper internals installation, damaged trays) and operation conditions. Maldistribution of flow exacerbates pressure value and creates localized high-pressure

zones. Potential consequences of high pressure include flooding and increased operation costs in vacuum systems.

Our experimental data showed controlled pressure drop of <1 kPa (**Supplementary Figure 5e**), within normal operational range. However, in industrial implementation, risks of entrainment and flooding caused by high pressure during startup should be considered. Wet feedstock and rapid degradation (caused by catalyst or uncontrolled heating) accelerate pressure buildup at the reactor bottom. To mitigate high pressure, tray engineering is critical for effective management of hydraulic behavior in the reactors. Sieve trays with large open area and small pore, valve trays, and cross flow trays are potential options.

Future work may prioritize scale-up studies and pilot testing to examine tray configurations for industrial fractionated degradation (experiments at laboratory scale is challenging due to difficult tray manufacturing at small scale, diameter ~ 2-3 cm). Anti-fouling properties of tray coating should also be included in relevant study, particularly, for real-life plastic wastes.

c. Back mixing. Back mixing refers to the unintended reversal of liquid or vapor flow, such as entrainment, disrupting the counter-current that essential for efficient separation. Back mixing of liquid occurs on trays due to liquid recirculation, excessive hydraulic gradients, or weeping. Back mixing of vapor arises when vapor jets penetrate liquid layers or when significant vapor bypasses around or through damaged trays. The back mixing disrupts the concentration gradient along the mass transfer direction, causing fluid mixing between adjacent trays and compromising Murphree tray efficiency. The consequences of back mixing include diminished product purity, increased energy consumption, reduced throughput capacity, compromised process controllability, etc.

Back mixing is critical for industrial applications and requires thorough tray engineering investigation.³⁷ Key layout parameters, flow path length, outlet weir height, flow path width, etc., require optimization to minimize back mixing. Crossflow trays and high-performance sieve trays can reduce back mixing, as well as pressure drop, fouling resistance, and entrainment.

3. Kinetic non-ideality.

Intrinsic kinetic. Kinetic parameters from polymer degradation studies often reflect apparent rather than intrinsic kinetic values due to coupled factors, including side reactions, heat transfer limitations, mass transfer effects, etc.³⁸ For example, thermal gradients in the reactor slow down the degradation reaction and could cause deviations from the intrinsic kinetics. Even with minimal heat and mass transfer limitations, kinetic curves may show deviations, attributable to the side reactions (**Supplementary Figure 22a**). In addition, during the fractionated degradation, characterizing molar mass evolution (polymer to small molecules) using a single analytical approach is challenging, while using integrated techniques may introduce systematic errors.

Acquiring intrinsic kinetic parameters presents critical experimental challenges. To approach the intrinsic value of polymer degradation, the kinetic experiments employed judiciously designed setup and processes. A small thin-walled Schlenk flask was heated with thick asbestos insulation under controlled conditions (~400 °C, ~1 atm). Molar mass characterization employed APCI-MS and GC-MS, with kinetic analysis focused specifically on the linear region (0.5-4 hours). Through mitigation of mass/heat transfer limitations, simulation of degradation unit conditions, and rigorous post-model fitting, the derived kinetic parameters can optimally approach the intrinsic values of PP random scission. To better determine intrinsic chain scission kinetics, judiciously designed microreactors featuring sub-mm channels might achieve near-isothermal conditions and further reduce diffusion timescales. Additionally, material forms (thin films, fibers, or nanoparticles) can be optimized to accelerate heat transfer. Computational deconvolution methods may provide valuable insights as

well, for example, iterative solutions of energy/mass balances coupled with reaction equations using computational fluid dynamics.

References

- 35 Peters, F. T., Laube, F. S. & Sadowski, G. PC-SAFT based group contribution method for binary interaction parameters of polymer/solvent systems. *Fluid Phase Equilib.* **358**, 137-150 (2013).
- 36 Soave, G., Gamba, S. & Pellegrini, L. A. SRK equation of state: Predicting binary interaction parameters of hydrocarbons and related compounds. *Fluid Phase Equilib.* **299**, 285-293 (2010).
- 37 Cai, T. J. & Chen, G. X. Liquid back-mixing on distillation trays. *Ind. Eng. Chem. Res.* **43**, 2590-2597 (2004).

Comment #29: *Side reactions are not discussed.*

Author reply: We thank the reviewer for pointing out our missing in the discussion of side reactions. A discussion of potential side reactions has been included in our revised manuscript. In addition, in accordance with the **comment #32**, the chemical effect of contamination in real-life PP wastes were also evaluated and included in the discussion of side reaction. Specifically, the side reactions were classified into two groups, side reactions during fractionated degradation and side reactions during sulfonation, include internal alkene formation, aromatization, polymerization (crosslinking and coking), gasification, etc.

Main text, Page 8: Characterization by heteronuclear multiple bond correlation NMR (HMBC-NMR) revealed the prevalent of α -olefins (**Supplementary Figure 10**, ^1H NMR δ 4.6, ^{13}C NMR δ 110 and δ 144), with minor internal alkenyl and aryl structures (^{13}C NMR δ 115-130) due to radical side reactions (**Supplementary Scheme 1b**, see Supporting Discussion).

Main text, Page 10: Compared to virgin PP plastics, real-life plastic wastes in supply chains exhibit dynamic characteristics (see Supplementary Discussion, **Supplementary Figures 19-20**). The levels of impurity (*e.g.*, polymers, biomass, organic additives, and inorganic materials, **Supplementary Figure 21**) vary between plastic wastes (**Supplementary Figure 20**, **Supplementary Tables 11-12**), influencing product yields, structures, and elemental compositions (**Supplementary Figure 22** and **Supplementary Table 12**). For instance, the fractionated degradation of real-life PP wastes (W1, W2, and W3) obtained from a single vendor exhibited reduced hydrocarbon yields and increased side reactions at lower purity, resulted in more internal alkene and aromatic byproducts.

2. Analysis of side reaction and impact of contamination on fractionated degradation.

NOTE: analysis of impurity effects on fractionated degradation products utilized exclusively real-life PP waste sourced from the plastic circular industry park in Jieshou, Anhui, China (W1, W2, W3). This selection was based on the site's high productivity (~ 8 wt.% of the world). While certain chemistry in the analysis may be applicable to real-life plastic waste from other sites, our analysis must not be construed as universally valid. The findings represent rigorously scientific observations strictly confined to PP waste processed in the Jieshou circular industry park, including materials that underwent standardized collection, sorting, cleaning, and reuse protocols specific to this industrial park.

The investigation of real-life PP wastes with varying purity focused on hydrocarbons collected from the T2 tray. This was based on prior performance data that PP-T2-5h-S demonstrated superior performance than those obtained from the T1 tray. Samples were designated as PP-T2W1, PP-T2W2, and PP-T2W3.

a. Aromatization and internal alkenyl formation. Internal alkenyl formation emerges as a notable side reaction during polyolefin degradation, with internal alkenes potentially serving as precursors to aromatics. Under appropriate conditions, alkenes may undergo cyclization and dehydrogenation leading to formation of aromatics. Non-catalytic thermal degradation of PP in inert atmosphere is initiated through radical reactions, including C-H and C-C homolysis, β -scission, or radical initiators (**Supplementary Scheme 1b**). Subsequent dehydrogenation or disproportionation then generates alkenyl intermediates (**Supplementary Scheme 1c**).²⁵⁻²⁷ These structures capable of rearranging, cyclizing, and dehydrogenating to form cyclic and aromatic structures (**Supplementary Scheme 1d**). For instance, intramolecular cyclization pathways such as Diels-Alder reactions could yield six-membered rings that subsequently aromatize to phenyl groups (**Supplementary Scheme 1d path II**).

The HMBC-NMR analysis of PP-T1-5h and -T2-5h detected minor signals of internal alkenes and aromatics during virgin PP degradation (**Supplementary Figure 10**). In real-life PP waste, impurities enhanced internal alkenyl and aryl formation, probably through both ionic and radical pathways. We observed substantial increases in internal alkenyl signal with higher contamination levels (110-130 ppm), while α -alkenyl signals diminished (**Supplementary Figure 22**). This inverse correlation suggested that some internal alkenyls might derive from α -alkenyl, potentially through α -olefin polymerization, forming internal radicals, then underwent disproportionation or dehydrogenation (**Supplementary Scheme 1c, path III**). Additionally, allylic methylene groups in α -alkenyls have higher reactivity than typical methylenes, which could facilitate hydrogen abstraction, leading to conjugated alkene formation near chain termini.

The formation of internal alkenyl groups corresponded to the reactions of CH₃, CH₂, and CH groups (**Supplementary Figure 22**). Specifically, the CH₃ signal near 20 ppm decreased sharply, likely due to internal alkenyl generation, causing CH₃ signal split with new peaks emerging near 22-23 ppm and at 14 ppm (allylic methyl in conjugated systems). These internal alkenyls appear highly reactive, further converting into aryl structures. This transformation correlated with peak intensification at 125-131 ppm (aryls) and 32-37 ppm (benzylic methylenes), particularly pronounced at the lowest purity levels (**Supplementary Figure 22f**).

b. Polymerization. Polymerization competes with polyolefin degradation according to the fundamental principle of polymerization.²⁸ The polymerization retards at high monomer conversions due to viscosity limiting diffusion to active sites. Under degradation (reduces viscosity and increases monomer concentration), polymerization rate would rebound, causing competition with degradation. Initiation of olefins could be induced

by C-H homolysis or initiator (*e.g.*, oxygen, peroxides, or some impurities). The resulting radicals propagate and trigger intramolecular reactions with internal alkenyl groups (**Supplementary Scheme 1e**), leading to crosslinked networks and cyclic structures. The polymerization and intramolecular reaction can be accelerated at elevated temperatures in the presence of radical-initiating species.

During PP fractionated degradation, polymerization of α -olefins might occur in the degradation unit and T2 tray. The product became more viscous with long reaction period (**Supplementary Figure 22c**). Kinetic analysis further confirmed this dynamic competition between polymerization and chain scission. The rapid chain scission dominated the initial 4 hours before plateauing due to polymerization (**Supplementary Figure 22a**).

c. Gasification. Gasification reaction producing light hydrocarbons (CH_4 , C_2H_4 , C_3H_6 , etc.) typically intensifies at elevated temperatures. Gasification primarily proceed through chain-end scission, or dominant after complete chain scission (**Supplementary Scheme 1f**). Gasification requires temperatures >500 °C for preferable conversion.²⁹ In our fractionated degradation systems, however, two factors suppressed gasification: rapid removal of reactive fragments from the hot regions and the lower operating temperature (400 °C), as evidenced by the detection of only trace C_4 - C_7 alkenes in PP-T0-5h.

d. Branching. During fractionated degradation, chain scission generates free radicals that initiate branching through radical coupling, chain transfer, or other pathways (**Supplementary Scheme 1**). The ^{13}C NMR analysis of PP-T2-5h reveals molecular tacticity variations and three characteristic signal groups at ~ 20 ppm (CH_3), ~ 29 ppm (CH_2), and ~ 45 ppm (CH). Significantly, signals in the 27-31 ppm region correspond to methine (CH) carbons, serving as potential branching indicators.³⁰

For real-life PP waste, impurities in real-life plastic waste complicated the quantification. With high impurity level, the CH intensity reduced significantly due to alkenyl formation, converting saturated tertiary and quaternary carbons to unsaturated species, obscured branching analysis through signal overlap and signal shifts.

Nonetheless, branching represents a potential reaction during fractionated degradation, coexisting with radical coupling and polymerization, processes inherent to olefin radical chemistry. When alkyl radicals couple with other radicals or alkenes, they form tertiary or quaternary branch points (**Supplementary Scheme 1g**). Furthermore, olefin polymerization (**Supplementary Scheme 1g**) also generates branched structures.

Supplementary Scheme 1. β -scission and potential side reactions.

3. Side reactions during the sulfonation

a. Olefin polymerization. In addition to radical-induced polymerization, olefins could undergo cationic polymerization when exposed to acid or cationic species (**Supplementary Scheme 1e, Path II**). These reaction increase oil viscosity, induce color darkening, and ultimately degrade surfactant performance and functionality. Notably, such polymerization becomes particularly dominant under low viscosity, and tend to be dominant when the concentration of acidic initiator is relatively low, inhibiting radical coupling termination and offering reaction opportunity for activated center and monomers.

b. Over-sulfonation. We define over-sulfonation as a series of side reactions involving pyrosulfonations, coking, and anhydrides formation (**Supplementary Scheme 2, Path I**).² Insufficient heat transfer is the primary cause of over-sulfonation in industrial processes. The highly exothermic nature of sulfonation creates localized hot spots, which accelerate side reactions. In our experiments, over-sulfonation may occur through three pathways. First, when using olefins with a broad distribution, the rapid sulfonation of light olefins facilitates over-sulfonation with sulfonation reagents, resulting in a sulfonated mixture with dark coloration (**Supplementary Figure 3b**).

Second, when sulfonation conditions are poorly controlled, such as with slow stirring, overheating, or rapid feeding of the sulfonation reagent, localized hot spots of either temperature or SO_3 are created, causing local over-reaction.

Third, in the case of real-life plastic upcycling, the presence of impurities, particularly cationic species and metallic components, may affect the reaction. Some metals can absorb SO_3 onto its surface, converting SO_3 to SO_2 and sulfides.³¹ Metal oxides can also incorporate SO_3 into the frameworks,³² while some metal ion coordinate with SO_3 , generating stable metal complexes.³³ Collectively, these interactions alter the electron structure on SO_3 , either stabilizing or destabilizing SO_3 and redirecting the sulfonation reaction pathways.

For instance, introducing SO_3 at stoichiometric ratios in the presence of metal and metal oxide particles would lead to SO_3 absorption and reduction, thereby decreasing the yield of products. The generation of SO_2 , sulfides, and other sulfur-containing byproducts would result in malodorous products and potentially cause secondary water pollution. If SO_3 becomes incorporated into a metal oxide framework and coordinates with metals, the reactivity and yields of the sulfonation reaction could be compromised.

c. Sultone. In alkene sulfonation chemistry, as the intermediate of sulfonation, sultones represent a special case. While typically, sultones undergo hydrolysis to form hydroxyl sulfonates without compromising surfactant performance, β -sultones present unique challenges. After hydrolysis, the adjacent hydroxyl and sulfonate induces intramolecular hydrogen bonding, creating stabilized conformations possess poor surfactant activity (**Supplementary Scheme 2, Path II**).

d. Internal alkene sulfonation. It is noteworthy that a portion of internal alkenes have formed through the aforementioned pathways or been induced by SO_3 (**Supplementary Scheme 2, Path III**). The sulfonation of internal alkenes results in internal sulfonates. This side reaction is captured in our HT-NMR-HMBC characterization (**Figure 3a**), showing a minor shifts near δ 5.2 (^1H NMR).

Supplementary Scheme 2. Potential side reactions during sulfonation.^{2,34}

References

- Ruiliang Gao, Shanjun Mao, Bing Lu, Wencong Liu & Wang, Y. Efficient upcycling of polyolefin waste to light aromatics via coupling C—C scission and carbonylation. *Angew. Chem. Int. Ed. Engl.* **64**, e202424334 (2025).
- Song, J. *et al.* Catalytic pyrolysis of waste polyethylene into benzene, toluene, ethylbenzene and xylene (BTEX)-enriched oil with dielectric barrier discharge reactor. *J. Environ. Manage.* **322**, 116096 (2022).
- Li, S., Li, Z., Zhang, F. & Chen, J. Upgrading waste plastics to value-added aromatics. *Chem. Catalysis* **4**, 100928 (2024).
- Odian, G. in *Principles of Polymerization, Radical Chain Polymerization*, 198-349 (John Wiley & Sons, New York, 2004).

- 29 Saebea, D., Ruengrit, P., Arpornwichanop, A. & Patcharavorachot, Y. Gasification of plastic waste for synthesis gas production. *Energy Rep.* **6**, 202-207 (2020).
- 30 Jung, M. *et al.* Analysis of chain branch of polyolefins by a new proton NMR approach. *Anal. Chem.* **88** (2016).
- 31 Sellers, H. & Shustorovich, E. Chemistry of sulfur oxides on transition metal surfaces: a bond order conservation-Morse potential modeling perspective. *Surf. Sci.* **356**, 209-221 (1996).
- 32 Joyner, N. A., Lee, Z. R. & Dixon, D. A. Binding of SO₃ to group 4 transition metal oxide nanoclusters. *J. Phys. Chem. A* **127**, 9541–9549 (2023).
- 33 Schenk, W. A. The coordination chemistry of small sulfur-containing molecules: a personal perspective. *Dalton Trans.* **40**, 1209-1219 (2011).
- 34 Kaneko, M., Kumagai, S., Nakamura, T. & Sato, H. Study of sulfonation mechanism of low-density polyethylene films with fuming sulfuric acid. *J. Appl. Polym. Sci.* **91**, 2435-2442 (2003).

Comment #30: *There is a vague discussion on β -scission control. Provide DFT or kinetic modeling.*

Author reply: We sincerely appreciate the reviewer's suggestion. We would like to clarify that the thermal degradation mechanism of PP in inert atmosphere was already comprehensively investigated (molecular dynamic, thermodynamic, and kinetic) in prior work through density functional tight-binding (DFTB) simulations and group additive methods (*Science*, **2023**, 381, 666-671). Therefore, we did not repeat the simulation in this work. However, the authors admit that some of our research data are not sufficiently discussed and analyzed, which could have revealed more valuable insights. Following the kind comments, the β -scission, PP fractionated degradation side reactions, and chemical effect of contamination in real-life waste were elaborated in the revised manuscript. The discussion of β -scission is presented below, while side reactions and chemical effect of contamination in real-life waste were presented in the reply for **Comments #29** and **#32**.

Main text, Page 7: The high α -alkene selectivity arises from methyl stabilization of PP radicals, which facilitates β -scission by reducing both Gibbs free energy and activation energy,¹³ favoring α -alkenyl formation (see Supporting Discussion).

SI, Page 18:

β -Scission, side reactions, and impact of contamination

1. β -scission.

Previous density functional tight-binding (DFTB) simulations suggested that PP degradation occurs mainly through random β -scission (**Supplementary Scheme 1a**), with disproportionation as a secondary pathway.¹ Our

current observations consistent with this mechanism. Under near-ambient pressure in a Schlenk flask, the evolving molar mass of fragments aligns well with the random scission model (**Supplementary Figure 15**). Moreover, mathematics correlations between alkenyl density (mmol/g) and scission frequency ($s = 1/M_n$), suggesting α -alkene formation may directly relate to chain scission behavior (**Supplementary Figure 11**). Taken together, these findings indicate β -scission plays a significant role in PP degradation, which aligns with observations in other reactor systems.^{23,24}

References

23. Guo, W.; Fan, K.; Guo, G.; Wang, J., Atomic-scale insight into thermal decomposition behavior of polypropylene: A ReaxFF method. *Polym. Degrad. Stab.* **2022**, *202*, 110038.
24. Peterson, J. D.; Vyazovkin, S.; Wight, C. A., Kinetics of the Thermal and Thermo-Oxidative Degradation of Polystyrene, Polyethylene and Poly(propylene). *Macromol. Chem. Phys.*, **2001**, *202*, 6.

Comment #31: *At 400 C under inert gas, radical reactions are expected, but the authors do not discuss any related reactions.*

Author reply:

We are grateful for the reviewer's valuable insight into reaction mechanisms. As the reviewer suggested, radical chemistry is the principal mechanism governing polyolefin degradation at 400 °C under inert atmosphere. Our expanded discussion now encompasses β -scission fragmentation as well as associated radical-mediated side reactions, including alkene formation, aromatization, polymerization, and others (detailed in **Supplementary Schemes 1-2**).

Accordingly, we have substantially enhanced our coverage of radical reactions throughout the manuscript. A dedicated section in the Supporting Information (" *β -Scission, side reactions, and impact of contamination*") addresses radical reactions (β -Scission, intrinsic side reactions, and side reactions induced by impurities), their mechanistic significance, and potential impurity effects. Elaboration on specific reaction can also be found in our responses to **Comments #29, #30, and #32**, as well as in Supporting Information.

Comment #32: *The impact of contaminants on the degradation pathway is not explained.*

Author reply: We sincerely appreciate the reviewer's thoughtful question. Impurity effects on product chemistry complicate plastic chemical recycling and have gained limited effort previously. To provide some

insight, degradation of real-life PP waste samples with diverse purity levels and careful data analysis were conducted. A overview of our study is given below.

Given the compositional complexity of plastic waste, especially considering potential spatio-temporal variations, establishing robust and broadly applicable chemistry would require intensive experiments and data analysis across diverse feedstock sources. **To focus on our research scope (exploring fractionated degradation and valorization of PP waste into surfactants), the study of impurity effect on PP fractionated degradation specifically examines real-life PP waste from the plastic circulation industry park where we conducted field research.** Selecting the Jieshou site instead of Heze site was deliberate and significant: Jieshou site produces ~3 million tons of sorted plastic waste, cleaned plastic waste, and regenerated plastic annually (see Supplementary Discussion of field research), and is the largest plastic circular industry park in China, representing 15% of China's recycled plastic and ~8% of the global total. We therefore believe this location offers valuable insights into China's plastic circulation systems, while also providing meaningful perspectives on global recycling practices.

To ensure our investigation reflected real conditions, we purchased PP waste samples with varying quality levels, named as W1, W2, and W3 (**Supplementary Figure 22d**), respectively. Following established standards (CJ/T 313-2009 and ASTM D523192), we characterized the composition of these materials through visual inspection before pulverizing into fine powder. Pulverization helped to ensure composition uniformity and preserved sample representative. After degradation in the three-fraction reactor, the resulting products from the T2 tray were then collected, named as PP-T2W1, -T2W2, and -T2W3, and characterized to obtain elemental composition and chemical structures (**Supplementary Figure 22** and **Supplementary Table 12**). We provided comprehensive details of these experimental methods in the updated Supplementary Information.

Our analysis revealed interesting correlations between purity and structures when comparing PP-T2W1-5h, -T2W2-5h, and -T2W3-5h with PP-T2-5h. In general, despite minor fluctuations, the intensities of α -alkenyl, methine, methylene, and methyl groups tended to decrease with declined purity, while internal alkenyl and aryl group intensified (**Supplementary Figure 22**). These trends suggest that impurities encouraged unsaturation, leading to the formation of more internal alkenyl and aryl structures. The corresponding reduction in methyl, methylene, and methine groups further hinted that this unsaturation could involve methyl transfer, branching, polymerization, and cyclization along the polymer backbone.

Accordingly, the discovery was incorporated into the main text and supporting discussion, as listed below. Please also refer to our reply to the **comment #29** for the complete discussion of “ *β -Scission, side reactions, and impact of contamination*”.

Main text, Page 10: Compared to virgin PP plastics, real-life plastic wastes in supply chains exhibit dynamic characteristics (see Supplementary Discussion, **Supplementary Figures 19-20**). The levels of impurity (*e.g.*, polymers, biomass, organic additives, and inorganic materials, **Supplementary Figure 21**) vary between plastic wastes (**Supplementary Figure 20, Supplementary Tables 11-12**), influencing product yields, structures, and elemental compositions (**Supplementary Figure 22 and Supplementary Table 12**). For instance, the fractionated degradation of real-life PP wastes (W1, W2, and W3) obtained from a single vendor exhibited reduced hydrocarbon yields and increased side reactions at lower purity, resulted in more internal alkene and aromatic byproducts.

Hence, to simplify feasibility evaluation, the scale-up experiments employed a single real-life PP waste sourced from Heze city (W-heze; **Supplementary Figure 19c**). This feedstock roughly consisted 87 wt.% PP, 10 wt.% PE, 1.7 wt.% biomass, 0.7 wt.% rubber, and 1.4 wt.% others (PET, PVC, PS, dust, etc.). We processed 10-kg batches to enable meaningful lumped assumptions while ensuring valid comparisons to homogeneous laboratory-grade PP. We designed and manufactured a specialized scale-up reactor with a cost-effective, single-fraction prickly column (STW, I.D. 40 mm; **Figure 2c and Supplementary Figure 19a**) to balance separation efficiency and manufacturing practicality. Fractionated degradation of W-heze generated comparable amount of hydrocarbon (PP-STW-6h, 87 wt.%), solid (12 wt.%), and gases (~ 1 wt. %) to the cases at the laboratory scale (**Figure 2c and Supplementary Table 1**). PP-STW-6h was primarily composed of α -alkenes, with minor alkanes (3.5 mol%) originated from PE (~ 10 wt.%) and biomass (1.7 wt.%), which are typical polymeric impurities in the real-life waste (**Supplementary Figures 19-23**). The $C\#_n$ of PP-STW-6h was ~27 (**Supplementary Table 1**). The distribution of PP-STW-6h (CV = 2.66, **Supplementary Table 1**) was similar to that of PP-ST-5h.

Main text, Page 10: Elemental profiles, however, varied across wastes. In the degradation products of W1, W2, and W3, levels of Al, Ba, Cr (100 mg kg⁻¹ to 1800 mg kg⁻¹) were more dominant than Ca, P, and S (< 70 mg kg⁻¹). Crucially, these metal levels showed no direct correlation with purity (**Supplementary Table 12**), underscoring the chemical complexity of real-life plastic waste.

SI, Page 8:

5. Purity estimation of real-life PP wastes.

Following China's national and ASTM standards,^{7,8} we manually sorted real-life plastic waste to determine their compositions. Waste plastics were evenly spread on a 1-meter diameter circular zone (sorting area), in a clean 1 m² platform to prevent contamination. Operators categorized materials by appearance, texture, plastic number, and other properties to separate plastics from impurities. Separated components were classified by type,

weighed using a platform balance (± 1 g accuracy), and purity of plastic waste was calculated as the plastic mass fraction.

SI, Page 19:

NOTE: analysis of impurity effects on fractionated degradation products utilized exclusively real-life PP waste sourced from the plastic circular industry park in Jieshou, Anhui, China (W1, W2, W3). This selection was based on the site's high productivity (~ 8 wt.% of the world). While certain chemistry in the analysis may be applicable to real-life plastic waste from other sites, our analysis must not be construed as universally valid. The findings represent rigorously scientific observations strictly confined to PP waste processed in the Jieshou circular industry park, including materials that underwent standardized collection, sorting, cleaning, and reuse protocols specific to this industrial park.

The investigation of real-life PP wastes with varying purity focused on hydrocarbons collected from the T2 tray. This was based on prior performance data that PP-T2-5h-S demonstrated superior performance than those obtained from the T1 tray. Samples were designated as PP-T2W1, PP-T2W2, and PP-T2W3.

SI, Page 20: The HMBC-NMR analysis of PP-T1-5h and -T2-5h detected minor signals of internal alkenes and aromatics during virgin PP degradation (**Supplementary Figure 10**). In real-life PP waste, impurities enhanced internal alkenyl and aryl formation, probably through both ionic and radical pathways. We observed substantial increases in internal alkenyl signal with higher contamination levels (110-130 ppm), while α -alkenyl signals diminished (**Supplementary Figure 22**). This inverse correlation suggested that some internal alkenyls might derive from α -alkenyl, potentially through α -olefin polymerization, forming internal radicals, then underwent disproportionation or dehydrogenation (**Supplementary Scheme 1c, path III**). Additionally, allylic methylene groups in α -alkenyls have higher reactivity than typical methylenes, which could facilitate hydrogen abstraction, leading to conjugated alkene formation near chain termini.

The formation of internal alkenyl groups corresponded to the reactions of CH₃, CH₂, and CH groups (**Supplementary Figure 22**). Specifically, the CH₃ signal near 20 ppm decreased sharply, likely due to internal alkenyl generation, causing CH₃ signal split with new peaks emerging near 22-23 ppm and at 14 ppm (allylic methyl in conjugated systems). These internal alkenyls appear highly reactive, further converting into aryl structures. This transformation correlated with peak intensification at 125-131 ppm (aryls) and 32-37 ppm (benzylic methylenes), particularly pronounced at the lowest purity levels (**Supplementary Figure 22f**).

SI, Page 21: For real-life PP waste, impurities in real-life plastic waste complicated the quantification. With high impurity level, the CH intensity reduced significantly due to alkenyl formation, converting saturated

tertiary and quaternary carbons to unsaturated species, obscured branching analysis through signal overlap and signal shifts.

Nonetheless, branching represents a potential reaction during fractionated degradation, coexisting with radical coupling and polymerization, processes inherent to olefin radical chemistry. When alkyl radicals couple with other radicals or alkenes, they form tertiary or quaternary branch points (**Supplementary Scheme 1g**). Furthermore, olefin polymerization (**Supplementary Scheme 1g**) also generates branched structures.

Supplementary Figure 22. Side reactions and purity effects on olefin chemistry. (a) Fractionated degradation kinetics under simulated conditions (400°C, ambient pressure) using a Schlenk flask. (b) Enlarged view of **Supplementary Figure 10b** highlighting aryl and internal alkenyl correlations. (c) Flow tests of PP-T2-5h and PP-T2-24h on silicone gel plate. PP-T2-24h shows yellowish color and higher viscosity than PP-T2-5h, suggesting polymerization. (d) Standardized purity analysis of Jieshou samples W1-W3 and their yields in laboratory-scale fractionated degradation using three-fraction reactors.⁷ The other category includes minor PET plastics, PVC plastics, and undetermined components. *NOTE, PP or PE plastics are PP polymer with additives.* (e,f) ^{13}C NMR (10-50 ppm) of fractionated products from W1-W3 and PP-T2 using

three-fraction reactors; structural variations in PP-T2, T2W1, T2W2, and T2W3 indicate purity-dependent trends. (g) Spectral evolution in 110-150 ppm region showing α -alkenyl and internal alkenyl (including aryl) evolution. (h) Evolution of internal alkenyl and aryl groups (red) and α -alkenyl (black line).

Reviewer #2

Author reply: We appreciate this reviewer for evaluating our manuscript and providing valuable suggestions. In the revised version, we have carefully addressed all comments and improved the manuscript accordingly, please refer to responses to Reviewer 1 and Reviewer 3 for details.

Reviewer#3

This a relevant study that pursues surfactants production from waste plastics. The proposed strategy consist of two steps, selective conversion of plastics in suitable hydrocarbon fractions (olefins of different chain lengths), then, these hydrocarbons are further transformed by sulfonation. The paper is in general clear and well written, moreover, it has some original aspects as the polymer degradation couples with products fraction. However, there is a need for improvement and clarification before considering publication.

Author reply: We are grateful to the reviewer for the positive evaluation and constructive comments on our manuscript. Below, please review our point-by-point responses to each of the raised comments.

Comment #1: *The process was studied feeding PP, however, the properties and typical product distribution obtained in different polyolefins degradation is similar. The authors should discuss the applicability of this approach to polyolefins mixtures, in fact, it could be easier to find waste polyolefins mixtures than pure PP.*

Author reply: We sincerely appreciate the reviewer's valuable comments. Indeed, given the structural similarities between PE and PP, their degradation product can be expected to show similar distributions. We agree that discussing the method applicability helps to illuminate the potential of our proposed pathway, and have therefore revised the manuscript to include specific consideration of PE/PP mixtures. Regarding the availability of pure PP waste, we would like to gently note that obtaining such material is actually not difficult in practice. This understanding comes from our field research conducted in 2024, which we have incorporated into the revised manuscript to provide better context.

In this work, the W-heze is a mixture of PE and PP, as we recently characterized (**Supplementary Figure 19c**). The mass ratio of PE (~10 wt.%) in W-heze was a rough evaluation. In addition, 1 wt.% rubber, 1.7 wt.% biomass were found from the mixture. The ST degradation of W-heze generated 87 wt.% liquid hydrocarbons, primarily composed of α -alkenes, with minor alkanes (3.5 mol%) originating from PE. The PP and PE mixture-derived sulfonate surfactant showed comparable performance to that obtained from pure PP (**Figure 3c,d**). With more PE in the mixture, the resulting alkane would be more dominant (40 mol% for pure PE, *Science*, **2023**, 381, 666-671). Alkanes can be sulfonated through a H-abstraction, alkene formation, sulfonation pathway (**Supplementary Scheme 2**). As the formed alkene could be internal, the pathway is less controllable than α -olefin sulfonation which generate alkenyl sulfonate near the chain end. In addition, there could be residue alkane in the resulting product, compromising the performance and safety of the surfactant. Therefore, the proposed upcycling is applicable for mixed polyolefin upcycling, yet it is more recommended to use a single plastic wastes to ensure upcycling process stability, optimal performance and safety. In China, it is niether difficult nor expensive to obtain single PE or PP waste. The price of high-pure PP (> 99.8%) is ~400 \$/t, similar to those in the European Union (Joint Report on Management of Plastic Waste in Europe, European Organization of Supreme Audit Institutions, 2022, **Supplementary Figure 2c**).

Additionally, we also conducted fractionated degradation of real-life PE waste. The product was sulfonated

and hydrolyzed into surfactant (PE-STW-S; **Figure R8**). However, we decided to exclude them from the manuscript: (1) PE and PP go through a slightly different mechanisms that PE degradation selectivity to α -olefin is low (~ 60 mol%; *Science*, **2023**, *381*, 666-671). The other 40% are mainly alkane which cannot be sulfonated as efficient as olefins (alkane sulfonation is possible but slow; *JAOCs*, **1991**, *68*, 8); (2) practical PE upcycling to sulfonate would involve acidic catalyst to improve selectivity of α -olefin or modified sulfonation processes (prolongated sulfonation period or higher reaction time) to improve alkane conversion. **Consequently, involving PE will double the data and make the manuscript too lengthy to read.**

To highlight the applicability of the proposed upcycling approach for polyolefin mixture, the manuscript was revised according to the reviewer's comment.

Main text, Page 10: PP-STW-6h was primarily composed of α -alkenes, with minor alkanes (3.5 mol%) originated from PE (~ 10 wt.%) and biomass (1.7 wt.%), which are typical polymeric impurities in the real-life waste (**Supplementary Figures 19-23**).

SI, Page 17:

Field research of PP waste supply and processing work flow

A field research was conducted in a major plastic circulation industry park in Jieshou, China at July 23, 2024. To learn the status-of-quo of plastic supply and processing work flow, the field research was conducted through non-participant observation and interview to the park supervisors and enterprise owners. We visited 5 companies in the park, including two large enterprises (100k ton per year), one specialized for PE and PP, and one for general plastics. The interviewing questions included (1) what product do you produce, (2) where do you obtain waste plastic, and (3) how do you convert the plastic waste to product; (4) what is the market price of PP wastes.

Jieshou is a major contributor of plastic waste circulation in China. According to China Xinhua News (<https://www.ah.chinanews.com.cn/news/2023/0211/312924.shtml>), an official media of China, the annual capacity of Jieshou in 2024 was 2.6 million metric tons, corresponding to $\sim 15\%$ of plastic wastes recycled in China, and $\sim 8\%$ of the world. Therefore, the operation situation in Jieshou is representative to the plastic circulation system in China, and partially representative of the world.

A potential general work flow for real-life PP waste is recovered based on the field research (**Supplementary Figure 2**). The initial pre-sorting is the critical step in the flow. The pre-sorting by plastic characteristics (*e.g.*, plastic labels, transparency, tactility, initial functions, etc.) helps skillful workers to determine the plastic type. In a company specialized for polyolefins, plastic waste are ordered by plastic types and colors (**Supplementary Figure 2a**; *e.g.*, hard bottles, caps, shopping bags, woven bags, etc.) to improve the

purity and color uniformity of their products. The sorted wastes can be baled for low-quality regenerated plastics (*e.g.*, disposable product, counterweight), or delivered for cleaning and refinement for improved waste quality. Sink-flow system is generally equipped to remove the heavy components (stones, sand, ashes, heavy polymers). The purified plastic wastes are dried for sell, or sorted again mechanically (AI-assisted and spectroscopic systems) for higher purity (**Supplementary Figure 22**, W1, ~ 99 wt. %), typically utilized for food-grade materials.

The product value is determined by the waste purity. A rough value margin of 128-400 \$/ton was quoted for beaconing the typical pricing of polyolefin wastes after primary sorting (mechanical sorted wastes are not included; **Supplementary Figure 2**). Although, the actual prices of each were not revealed, it possibly elevated after each purifications. To be conservative, the highest price of 400 \$/ton was utilized in TEA.

SI, Page 28:

Applicability for mixed and contaminated polyolefin wastes

Properly sorted, cleaned, and purified real-world PP wastes (W-heze, W1, W2; **Figure 2d**, **Supplementary Figures 19 and 22**) yielded dominantly α -olefins. The resulting PP-STW, PP-T2W1, and PP-T2W2 exhibit minimal contamination and trace internal and aryl structures (**Supplementary Figure 22**). The W-heze (87 wt.% PP, 10 wt.% PE, 0.7 wt.% rubber, 1.7 wt.% biomass, 1.4 wt.% other, **Supplementary Figure 19c**) exemplifies the process applicability to mixed polyolefin with contamination, yielded PP-STW-6h (87 wt%) with dominantly α -alkenes and minor alkanes (~3.5 mol%). Critically, the metallic impurities level of PP-STW-6h was low (**Supplementary Table 11**), while metal could precipitate sulfonates, it showed negligible impact on the specific case of W-heze upcycling to sulfonate (**Figures 3c, 3d**).

In contrast, severely contaminated PP waste (*e.g.*, PP-T2W3: Al ~1.7 g/kg, Fe 1.7 g/kg, Ba 0.5 g/kg; **Supplementary Table 11**) or PP waste with high PE content, may compromise upcycling applicability. Impurities (metals and organic impurities) reduced α -olefin dominance during fractionated degradation, generating greater unsaturation through internal alkenes and aryl compounds (**Supplementary Figure 22**). High-PE feedstocks further exacerbate side reactions, producing substantial alkanes (~40 mol% for pure PE),¹ which reacts less efficiently than α -olefins. The alkane portion may be converted to internal alkenes via hydrogen abstraction (**Supplementary Scheme 2, Pathway III**), and then promotes polysulfonate formation, polymerization, coke, and internal sulfonates (**Supplementary Schemes 1e, 2**). Metals (*e.g.*, Ca, Mg, Fe) intensify coloration and precipitate sulfonate (*e.g.*, Ca-sulfonates).⁴⁸ Crucially, while hydrolysis treatments remove metals and some organics, they cannot prevent irreversible persulfonate and sulfonated byproducts that degrade detergency, disrupt hydrophilic-lipophilic balance (HLB), reduce solubility, darken color, and impair emulsification.

Hence, the proposed method demonstrates applicability for upcycling mixed polyolefin and contaminated wastes into surfactants, though real-world utility remains application-specific. When processing PP-rich wastes (>95 wt.%), or PE-rich streams using modified conditions (elevated degradation temperature, acidic catalyst),^{1,49,50} the approach yields high-performance surfactants comparable to the commercial benchmarks. However, for severely contaminated wastes, product complexity may compromise viability of our method in high-value applications like household cleaning, enhanced oil recovery, precision emulsifiers, and corrosion inhibitors.⁵¹ These materials, nonetheless, may serve effectively in industrial cleaning, heavy metal removal, and low-end detergents where blended sulfonates meet performance requirements.

References

- 48 Zhang, G., Wei, G., Liu, Z., Oliver, S. R. J. & Fei, H. A Robust sulfonate-based metal–organic framework with permanent porosity for efficient CO₂ capture and conversion. *Chem. Mater.* **28**, 6276–6281 (2016).
- 49 Ke, L. *et al.* Polyethylene upcycling to aromatics by pulse pressurized catalytic pyrolysis. *J. Hazard. Mater.* **461**, 132672 (2024).
- 50 Gan, L. *et al.* Beyond conventional degradation: catalytic solutions for polyolefin upcycling. *CCS Chemistry* **6**, 313-333 (2024).
- 51 Shaban, S. M., Kang, J. & Kim, D.-H. Surfactants: recent advances and their applications. *Compos. Commun.* **22** (2020).

Figure R8. Digital image of a PE-STW-S foaming in hard water.

Supplementary Figure 2. Field research and market investigation. (a) field research of plastic circulation industry park in Jieshou (July 2024). The flow chart represent a potential process flow of plastic wastes to obtain PP waste of different purity. (b) Quoted prices of real-life PP waste during the field research. Note that the prices are rough estimations and could fluctuate. W1, W2, W3 are purchased from the Jieshou plastic circulation industry park to investigate potential effect of impurity on products. W-heze was purchased from the local municipal recycling depot in Heze city. The degradation of W-heze in our scale-up reactor generated PP-STW. The box chart of M.I. represented prices of waste PP cited from <https://jiage.zz91.com/suliao>, (July 14, 2025). (c) Bruent oil price (Nov. 2024) per ton was recalculated from barrel; Virgin PE and PP were obtained from IMARC reports;^{59,60} Approximate prices of waste polyolefins were obtained from EU joint research center report;⁶¹ Prices of benzene, toluene, and *p*-xylene were obtained from ECHEMI, Market price&Insight website (Nov. 2024); Price of jet fuel was obtained from IATA Jet Fuel Price Monitor website (Nov. 15, 2024); Approximate prices of anionic surfactants in EU, CN, and US were obtained from ChemAnalyst report.⁶²

Comment #2: *The authors should clarify how were determined the carbon distributions shown in Figure 1, the explanations are unclear.*

Author reply: We sincerely thank the reviewer for the valuable comments and apologize for our missing. As we have introduced in the manuscript, GC-MS provides excellent separation, quantification, and detection performance for volatile hydrocarbons; however, it cannot accurately detect low-volatility hydrocarbons for column condensation. Instead, APCI-MS is more suitable for detecting medium-to-high molecular weight compounds. Therefore, heavy fractions (boiling point $>300^{\circ}\text{C}$, $\sim\text{C}_{19}$) were quantified using APCI-MS to overcome this constraint. Accordingly, we have revised the manuscript to clearly describe the distribution determination.

Main text, Page 6: To accurately determine the distribution and number-average molar mass (M_n), hydrocarbons were characterized by both GC-mass spectrometry (GC-MS) and atmospheric pressure chemical ionization mass spectroscopy (APCI-MS; **Supplementary Figures 8-9** and **Equations 2-3**). GC provided precise composition information for volatile mixtures, though its sensitivity was limited for low-volatility hydrocarbons due to column condensation.³⁵ Therefore, heavy fractions (boiling point $>$ GC column temperature) were quantified using APCI-MS.¹³

SI, Page 8: The GC-MS column operated at 300°C can accurately detect hydrocarbons below C_{19} . We, therefore, reconstructed mass distribution profiles of PP-T0-5h and PP-T1-5h directly from GC peak integration areas. For hydrocarbons above C_{20} , APCI-MS delivered better accuracy than GC. Therefore, mass distribution profiles of PP-T2-5h and -ST-5h combined GC-MS ($\text{C}_{15}\text{-C}_{19}$; **Supplementary Figure 8**) and APCI-MS (above C_{20} ; **Supplementary Figure 9a**). As APCI-MS functions as a molar detector, mass contributions were calculated by multiplying peak intensities by molar masses. PP-NT-5h was dominated by components above C_{19} and was analyzed only via APCI-MS.

Comment #3: *Results discussion is weak, the main contribution of this study is the incorporation of the fractionation system, accordingly, its role on products composition should be further discussed. Moreover, the authors should clearly explain the real advantages of this fractionation strategy, is it better than a typical products separation after oil condensation (i.e., a conventional distillation).*

Author reply: We sincerely thank the thoughtful and constructive comment. To enhance clarity and accessibility to readers, we have revised the manuscript. We would now like to outline the benefits of fractionated degradation below, particularly in contrast to conventional degradation-distillation process.

(1) Selective preparation of PP-derived α -olefins with tunable and narrow CLD at high yield and less

wastes. Fractional degradation synthesizes a tailored "cake" of specific hydrocarbons, whereas degradation-distillation partitions an existing "cake" for useful portions. In other words, degradation-distillation processes separate the pyrolysis oil into fractions, with desired components utilized for downstream processes and undesired portions treated as waste. These highlight two key advantages of fractional degradation: 1. enhanced product yields; 2. reduced generation of undesired hydrocarbons.

(2) Reduced operational costs and energy requirements. The fractionated degradation system represents an example of process intensification that combines degradation and fractionation within a single integrated equipment, reducing manufacturing expenses, facility floor area, and energy demands.

In summary, relative to conventional degradation-distillation methods, our proposed fractionated degradation approach demonstrates advantages in energy efficiency, product selectivity, yields, and environmental sustainability. These collective benefits suggest potential for establishing a scalable, high-value upcycling pathway for polymer waste streams.

According to the reviewer comment, we have revised our manuscript as follows.

Main text, Page 14: Degradation-distillation (D) scenario distilled PP degradation products to isolate target fractions, followed by sulfonation and hydrolysis (**Figure 4a**). Degradation-distillation could not selectively produce PP-derived α -olefins with tunable chain lengths at high yield. Nonetheless, for conservative analysis, the material flows of B and D were assumed equivalent, which overestimated the benefits of D scenarios (**Figure 4b**).

Main text, Page 14: In contrast, D-SD reduced IRR to 32.7% and emitted 40% more CO₂ than B-SD, confirming the superiority of fractionated degradation in sustainability and profitability than degradation-distillation.

Main text, Page 17: Fractional degradation represents a advantageous methodology for controlling C_{#n} and CLD in polyolefin degradation (**Supplementary Table 17**), improving energy efficiency, product selectivity, yields, and sustainability than direct distillation of pyrolysis-oil.

SI, Page 19:

2. Advantage of fractionated degradation over direct distillation.

a. Selective preparation of PP-derived α -olefins with tunable and narrow CLD at high yield. Fractional degradation synthesizes a tailored fraction of specific hydrocarbons, whereas direct distillation partitions an existing mixture for useful portions. In our three-fraction reactor system, C₃-C₇ (gaseous), C₆-C₁₅

(liquid), and C₁₅-C₃₀ (liquid) α -olefins were successfully obtained with narrow CLD (CV < 0.6). These distributions appeared notably narrower than the literature (**Supplementary Figure 7a**). In comparison, direct distillation only separates the desired fractions from an existing mixture for downstream processes, with the undesired fraction wasted, leading to a lower yield than fractionated degradation.

b. Reduced operational costs and energy requirements. The degradation-distillation approach entails additional equipment costs, maintenance, and energy (~40% higher CO₂ emissions, **Figure 4, Supplementary Table 16**). In contrast, the fractionated degradation system represents a process intensification strategy that combines degradation and fractionation within an integrated equipment, reducing manufacturing expenses, facility floor area, and energy demands (**Supplementary Table 13**).²²

Reference

- 22 Stankiewicz, A. & Moulijn, J. A. Process intensification: transforming chemical engineering. *Chem. Eng. Prog.* **96**, 22-34 (2000).

Comment #4: *What is the interest and relevance of the modeling section? What about the consistency of the kinetic parameters of this model? The use of suitable kinetic parameters is a critical aspect in that kind of models, sometimes, this is very challenging as long as reaction rate is controlled by heat transfer and not by reaction rate.*

Author reply: We sincerely apologize for any confusion caused by the lack of details in our original manuscript. We have revised the manuscript to explain the significance of C_{#n} profile by T_r-C_{#n} correlation. More details were added to explain the kinetic study. Supporting discussion of non-ideality now addresses deviations from intrinsic kinetic parameters, with methodological suggestions to assist future researchers. Please allow us to discuss your constructive comment as follows.

Interest and relevance.

This work proposes a fractionated degradation framework for PP, utilizing a “classifier” unit to separate molecules by size based on side-dependent properties. Reactors were specifically designed to exploit latent heat for this purpose. Within the reactor, complex interacting processes, including VLE, chain scission kinetics, and heat and mass transfer collectively govern PP fractionated degradation, particularly C_{#n} profile. These interactions manifest as a linear temperature gradient along the mass transfer direction (**Figure 2a**). Critically, this temperature profile intrinsically correlates with the C_{#n} profile. Therefore, rather than modeling using parameters in the complicated interactions with unknown accuracy, we established the direct correlation between temperature and C_{#n} near equilibrium (**Figure 2b**). This correlation is practical for four reasons:

(1) **avoiding C_{#n} profile deviation caused by parametric uncertainties inherent to mechanistic models.** As noted, obtaining intrinsic kinetic parameters are experimentally challenging under industrial

conditions, while ancillary properties (viscosity, diffusivity, non-ideal activity coefficients) introduce compounding errors in stepwise VLE models. By directly linking temperature to $C\#_n$, our approach eliminates error propagation from imperfect parameterization, ensuring practicability for establishing $C\#_n$ profile for a similar system.

(2) **revealing the indicator of $C\#_n$ profile.** The temperature gradient provides insight into the $C\#_n$ profile, acting as an integrated indicator driven by VLE thermodynamics, reaction kinetics, heat transfer, mass transfer, and others.

(3) **bypassing data-intensive computations.** Establishing the empirical $C\#_n$ profile eliminates the need for resource-intensive simulations of the underlying complex processes (VLE, kinetics, transport, hydrodynamics, etc.), reducing computational demands for accurate physiochemical data.

(4) **enabling potential practical implementation.** The correlation provides a portable tool for real-time monitoring of $C\#_n$ and rough CLDs during operation *via* simple temperature measurement.

While the $C\#_n$ profile does not utilize kinetic parameters, the robustness of the T_f - $C\#_n$ correlation was validated across experiments in different reactors and supported by ASPEN computation (**Figure 2b**), with only minor deviations. This robust T_f - $C\#_n$ correlation confirms the capturing of the primary thermodynamic drivers of $C\#_n$ regulation, despite some simplifications.

1. **Kinetic consistency.** Obtaining intrinsic kinetic parameters is challenging due to imperfect reaction conditions (*J. Energy Chem.*, **2023**, *86*, 84-117; uneven heating, rough reactor surface, poor thermal conductivity, etc.) even for state-of-art devices. For polymer chain scission, it is so difficult that even impossible to obtain intrinsic kinetic of degradation through experiments.

First, the polymer chain scission reaction is “polluted” by complicated side reactions (Supplementary Figure 22a). The non-linearity of the kinetic curve is the consequence of competing side reactions (polymerization, branching, termination, coking, crosslinking; see Supporting Discussion of *β -scission and side reactions*), and cannot be represented by a simple kinetic parameter. Such behavior is expected for polymer degradation chemistry, as with the going of degradation, reduced viscosity promotes the chance of radical-monomer reaction (*Polym. Degrad. Stab.*, **1997**, *57*, 113-125). Therefore, even with ideal heat transfer, the kinetic curve near the start or end of the reaction could be “polluted” by side reactions, causing deviation from a simple mechanism.

Second, the determination of absolute molar mass (from polymer to small molecules) is challenging. Due to the high molar mass of commercial plastic (> 20 kDa), accurate determination of the whole molar mass evolution by a single approach is challenging. At the initial stage, the degradation was minor. High-temperature gel-permeation chromatography (HT-GPC) can accurately determine the average molar mass using standard PP

references. The accuracy dropped significantly for heavy hydrocarbons as the hydrodynamic volume between small molecules is minimal and GPC column cannot separate them efficiently. The topological variation (cause by side reaction) from the reference further compromises the accuracy. Hence, the molar mass evolution would be highly unreliable for the interest region (< 500 Da) in this work.

Therefore, one can only approach the intrinsic kinetic parameters of polyolefin random scission. To do so, we made the following attempts and it represents our most rigorous approximation approach currently achievable:

(1) specialized setup with small and thin reactor ($d = 5.3$ cm, wall thickness = 1.02 mm) to minimize temperature gradients (**Figure R9**, mass/heat transfer limitation mitigation);

(2) carefully controlled and monitored reaction temperature and pressure ($\sim 400^\circ\text{C}$ and ~ 1 atm; see SI or below for experimental procedure); the condition simulated the condition in the degradation unit of the fractionated reactors;

(3) thick asbestos thermal insulation to minimize the heat loss and suppress temperature gradient;

(4) direct product characterization using single characterization technique (APCI-MS) to reduce systematic error;

(5) post-model fitting of random scission kinetic in the linear region (**Supplementary Figures 15 and 22a**). In complex kinetics (like fractionated degradation), the initial and end period often follows a different, non-linearizable behavior from random scission. Identifying and fitting the linear region (0.5 h to 4 h) is crucial for approaching the intrinsic parameters.

In addition, the author would like to humbly correct a misunderstanding. Fractionated degradation process is governed by multi-physic processes. Heat transfer is the primary factor at the initiation stage, influencing plastic melting and decomposition. For example, no significant decomposition was observed in the first 10 minutes (no entrainment). Once the polymer melted, decomposition proceeded rapidly (**Supplementary Figure 22a**), which is mainly controlled by random scission (**Supplementary Figure 22a**, equilibrium). Near the termination, the kinetic exhibited a decline as the result of the competing polymerization reactions and other side reactions (see Supporting Discussion of side reaction). Therefore, mass transfer should be a critical influencer near the end of the reaction.

Figure 2. Modeling, simulation, and scale-up experiments. (a) T_f and T_s distribution at varying tray heights in the three-fraction reactor. All temperature values are an average of five points per fraction (**Supplementary Figure 13**). (b) Correlation between T_f and $C_{\#n}$. The $C_{\#n}$ profile showed a linear correlation between $\ln(T_f)$ and $e^{\frac{C_{\#n}}{14B}}$, as validated by experimental and ASPEN simulated data. (c) Schematic of a scale-up reactor (left) and product yield using W-heze. (d) Elemental analysis of PP-STW-6h and PP-ST-5h. All error bars represent standard deviations.

Supplementary Figure 22a. Fractionated degradation kinetics under simulated conditions (400°C, ambient pressure) using a Schlenk flask.

Figure R9. Reaction setup for PP degradation kinetics under 400°C isothermic and near ambient pressure.

Accordingly, we have revised the manuscript as follows.

Main text, Page 8: To elucidate the $C\#_n$ regulation in fractionated reactors, a $C\#_n$ profile model was developed (Figure 2, Supplementary Figures 12-14 and Equation 5-13), supported by ASPEN HYSYS simulation (Figure 2, Supplementary Figures 15-17, Supplementary Tables 3-7). Polyolefin degradation released vaporized fragments that migrated across reactor trays. Under near-equilibrium and constant molar flow conditions, the interplay of heat/mass transfer and chain scission kinetics produced a linear temperature profile (Figure 2a) along the hydrocarbon transfer axis, corroborated by T_f and external surface temperature (T_s). Therefore, a direct correlation between temperature and hydrocarbon $C\#_n$ can be developed (Figure 2b) to bypass computational-intensive vapor-liquid equilibrium (VLE) calculations and avoid potential parametric inaccuracy, such as intrinsic kinetic parameters and unknown physiochemical properties. The $C\#_n$ profile unveiled a linear relationship between $\ln(T_f)$ and $\frac{C\#_n}{e^{14B}}$, and validated by experimental data cross different fractionated reactors (Figure 2b). The ASPEN simulated temperature and $C\#_n$ in an eight-fraction reactor (True Boiling Point and Engler distillation modules; Supplementary Figure 17a) showed similar linearity by the $C\#_n$ profile. The minor nonlinearity can be attributed to non-ideal VLE and varying chemical composition throughout the reactor. The robust correlation between T_f and $C\#_n$ demonstrated that T_f serves as the primary indicator for $C\#_n$ profile in fractionated degradation.

SI, Page 12:

3. Chain scission kinetics.

The initial molecular weight of laboratory-grade PP was determined by HT-GPC (Supplementary Figure 15a). To determine the chain scission rates, laboratory-grade PP powder (< 5 mg) was added into a small thin Schlenk flask (78 mL, diameter 5.3 cm, thickness 1.02 mm). The flask was degassed and refilled with Ar three times to evacuate oxygen. Constant Ar flow was supplied to the reactor from the side-tube through a flexible long needle when heating the reactor to the target temperature (360 or 400 °C; ~ 50 °C min^{-1}). The temperature was

monitored with a thermocouple, placed in the center of the reactor. After ~10 minutes, the temperature reached and stabilized near the target temperature. The long needle and thermocouple was removed and the Teflon cap was closed. After a few hours, the reaction was halted by immersing the flask into water. Deuterated solvent (10 mL) was injected into the flask to dissolve the products for APCI-MS characterization (**Supplementary Figures 15b, 15c, and Supplementary Table 3**). To avoid interruption by polymer melting and side reactions, linear regions (0.5 - 4 h) were fitted by random scission model (**Equation S17 and Supplementary Figure 15d**).¹⁴ Using Arrhenius equation, $\ln(A)$ and activation energy (E_a) were predicted to be 19.7 h^{-1} and $134.4 \text{ kJ mol}^{-1}$.

SI Page 13: The kinetic parameters of PP under fractionated degradation conditions (isothermic heating and ambient pressure at ~ 1 atm) were determined using random scission model and utilized for ASPEN reactor setup (E_a , A , and reaction order). Feed streams were defined by their composition, flow rate, and temperature, as shown in **Supplementary Tables 4-6**.

SI, page 25:

3. Kinetic non-ideality.

Intrinsic kinetic. Kinetic parameters from polymer degradation studies often reflect apparent rather than intrinsic kinetic values due to coupled factors, including side reactions, heat transfer limitations, mass transfer effects, etc.³⁸ For example, thermal gradients in the reactor slow down the degradation reaction and could cause deviations from the intrinsic kinetics. Even with minimal heat and mass transfer limitations, kinetic curves may show deviations, attributable to the side reactions (**Supplementary Figure 22a**). In addition, during the fractionated degradation, characterizing molar mass evolution (polymer to small molecules) using a single analytical approach is challenging, while using integrated techniques may introduce systematic errors.

Acquiring intrinsic kinetic parameters presents critical experimental challenges. To approach the intrinsic value of polymer degradation, the kinetic experiments employed judiciously designed setup and processes. A small thin-walled Schlenk flask was heated with thick asbestos insulation under controlled conditions (~400 °C, ~1 atm). Molar mass characterization employed APCI-MS and GC-MS, with kinetic analysis focused specifically on the linear region (0.5-4 hours). Through mitigation of mass/heat transfer limitations, simulation of degradation unit conditions, and rigorous post-model fitting, the derived kinetic parameters can optimally approach the intrinsic values of PP random scission. To better determine intrinsic chain scission kinetics, judiciously designed microreactors featuring sub-mm channels might achieve near-isothermal conditions and further reduce diffusion timescales. Additionally, material forms (thin films, fibers, or nanoparticles) can be optimized to accelerate heat transfer. Computational deconvolution methods may provide valuable insights as well, for example, iterative solutions of energy/mass balances coupled with reaction equations using computational fluid dynamics.

Reference.

38 Liu, J., Zhang, S., Wang, W. & Zhang, H. Photoelectrocatalytic principles for meaningfully studying photocatalyst properties and photocatalysis processes: from fundamental theory to environmental

applications. *J. Energy Chem.* **86**, 84-117 (2023).

Comment #5: *The overall process proposed (Figure 4) is interesting; however, the applicability of this process under real process conditions should be commented. Thus, real waste plastics contains several different impurities, what is their potential role on different reaction steps and on the quality and application of final product (surfactants). Please, clarify.*

Author reply: The authors deeply appreciate the reviewer for the valuable comments. In the proposed process, after fractionated degradation, the products are sulfonated with SO₃ using industrial-preferred drop-film reactor. The sulfonated product is then hydrolyzed and dried either by solar drying (SD) or spray drying (P). When the distribution of the fractionated degradation product is well-regulated, the sulfonation and post-treatment processes can be adaptable by current sulfonation industry. However, there is one important variable, as reviewer has suggested, the composition of the real-life waste could be contaminated by impurities, which affect the degradation reaction and surfactant performance. Therefore, an evaluation of applicability is given below and in the revised manuscript for reference.

When properly sorted, cleaned, and purified real-life PP waste samples (W-heze, W1, W2; **Figure 2d**, **Supplementary Figures 19 and 22**) undergo fractionated degradation, the resulting products (PP-STW, PP-T2W1, PP-T2W2) exhibit minimal contamination from alkaline earth metals, transition metals, and other elements (**Supplementary Figure 11**). These materials are dominated by α -olefin structures (**Supplementary Figure 22**), with minor internal or aryl signals. Although metallic impurities can coordinate with sulfonate groups to form complexes, the scale-up experiments using W-heze demonstrated minor impact on surfactant performance (**Figures 3c, 3d**). Therefore, at high purity levels (W-heze, W1, and W2; >95 wt.% are polyolefins), surfactant quality remains largely unaffected.

Nonetheless, when real-life plastic waste is severely contaminated, the value and target of upcycling should be reconsidered. Our analysis of impurity effects on the fractionated degradation (reply to **Comment #32**) reveals that plastic waste with lower purity levels generated more unsaturated hydrocarbons: internal alkenes and alkyl aromatics (**Figure S22**; see “*Analysis of side reaction and impact of contamination on fractionated degradation*”). According to fundamental sulfonation principles (de Groot, W.H. *Sulphonation Technology in the Detergent Industry*, Springer, 1991), the unsaturated species complicate sulfonation chemistry. The process yields byproducts, such as polysulfonated compounds, polymerized molecules, and coke (**Supplementary Scheme 1**; **Supplementary Scheme 2**).

Metal impurities pose dual challenges in surfactant production. Metals could catalyze side reaction, reducing sulfonation yields, and form insoluble salts. While the effects of impurity metals were minimal in W-heze, W1, and W2 (**Supplementary Figures 22f and 22g**), the excessive metals in W3 (**Supplementary**

Table 12, e.g., Al ~1.7 g/kg, Fe 1.7 g/kg, Ba 0.5 g/kg) may promote side reactions during sulfonation and cause surfactant precipitation. Calcium is an example of sulfonate precipitation (*Chem. Mater.*, **2016**, 28, 6276), forming stable alkaline earth salts. The chemistry is exploited in ion-exchange resins for water purification.

While hydrolysis treatments (aqueous hydroxide and sulfate solutions) reduce metal and some organic impurities, they cannot prevent the irreversible formation of byproducts (persulfonates and sulfonated polymers), critically degraded detergent performances, such as detergency, hydrophilic-lipophilic balance (HLB), solubility, color, and emulsification capacity, ultimately diminishing compatibility for high-value household cleaning purposes.

Accordingly, we have revised the Supporting Discussion. The potential effect of impurity on the method applicability is included in the section “*Applicability for mixed and contaminated polyolefin wastes*”.

SI, Page 28:

Applicability for mixed and contaminated polyolefin wastes

Properly sorted, cleaned, and purified real-world PP wastes (W-heze, W1, W2; **Figure 2d, Supplementary Figures 19 and 22**) yielded dominantly α -olefins. The resulting PP-STW, PP-T2W1, and PP-T2W2 exhibit minimal contamination and trace internal and aryl structures (**Supplementary Figure 22**). The W-heze (87 wt.% PP, 10 wt.% PE, 0.7 wt.% rubber, 1.7 wt.% biomass, 1.4 wt.% other, **Supplementary Figure 19c**) exemplifies the process applicability to mixed polyolefin with contamination, yielded PP-STW-6h (87 wt%) with dominantly α -alkenes and minor alkanes (~3.5 mol%). Critically, the metallic impurities level of PP-STW-6h was low (**Supplementary Table 11**), while metal could precipitate sulfonates, it showed negligible impact on the specific case of W-heze upcycling to sulfonate (**Figures 3c, 3d**).

In contrast, severely contaminated PP waste (e.g., PP-T2W3: Al ~1.7 g/kg, Fe 1.7 g/kg, Ba 0.5 g/kg; **Supplementary Table 11**) or PP waste with high PE content, may compromise upcycling applicability. Impurities (metals and organic impurities) reduced α -olefin dominance during fractionated degradation, generating greater unsaturation through internal alkenes and aryl compounds (**Supplementary Figure 22**). High-PE feedstocks further exacerbate side reactions, producing substantial alkanes (~40 mol% for pure PE),¹ which reacts less efficiently than α -olefins. The alkane portion may be converted to internal alkenes via hydrogen abstraction (**Supplementary Scheme 2, Pathway III**), and then promotes polysulfonate formation, polymerization, coke, and internal sulfonates (**Supplementary Schemes 1e, 2**). Metals (e.g., Ca, Mg, Fe) intensify coloration and precipitate sulfonate (e.g., Ca-sulfonates).⁴⁸ Crucially, while hydrolysis treatments remove metals and some organics, they cannot prevent irreversible persulfonate and sulfonated byproducts that degrade detergency, disrupt hydrophilic-lipophilic balance (HLB), reduce solubility, darken color, and impair emulsification.

Hence, the proposed method demonstrates applicability for upcycling mixed polyolefin and contaminated wastes into surfactants, though real-world utility remains application-specific. When processing PP-rich wastes (>95 wt.%), or PE-rich streams using modified conditions (elevated degradation temperature, acidic catalyst),^{1,49,50} the approach yields high-performance surfactants comparable to the commercial benchmarks. However, for severely contaminated wastes, product complexity may compromise viability of our method in high-value applications like household cleaning, enhanced oil recovery, precision emulsifiers, and corrosion inhibitors.⁵¹ These materials, nonetheless, may serve effectively in industrial cleaning, heavy metal removal, and low-end detergents where blended sulfonates meet performance requirements.

Comment #6: *The authors should provide additional details regarding the reactor operation. How was the reactor heated? what about temperature control? What is the heating rate used?*

Author reply: We appreciate the reviewer's comment. The detailed description regarding the heating method, temperature control, and heating rate are updated in the revised Manuscript and Supporting Information.

Main text, Page 5: Under inert atmosphere, pulverized PP (10.0 g) was heated to ~ 400°C (50 °C min⁻¹) in a heating mantle, generating hydrocarbon vapors fractionated by the trays.

SI page 1:

1. Reaction temperature calibration and pressure monitoring.

The reaction temperature of reactors with three-, single-, and none-fraction were calibrated using a thermocouple thermometer. Specifically, the thermocouple probe was placed at the bottom of the reactor (**Supplementary Figures 4 and 5**). The heating mantle temperature was gradually increased, and the thermocouple readings were monitored until the temperature stabilized at the target value for 30 minutes. The reactor pressure during the reaction was directly measured using a digital pressure meter equipped with needle probes.

2. PP degradation in three-fraction reactors.

Laboratory-grade PP (10 g) or real-life waste PP (W1, W2, W3) was loaded into a three-fraction reactor consisting of a degradation unit (~100 mL) and trays (~20 mL each, **Supplementary Figure 5**). The reactor was seamlessly molded to ensure airtightness and was set up in an Ar-filled glovebox to avoid oxygen contamination. Inside the glovebox, the top tray was connected to a total condenser via a frosted neck and sealed with vacuum grease. The condenser was plugged with a fluorinated stopper and connected to a gas bag. The setup was moved

out of the glovebox and placed in a heat mantle. To optimize thermal insulation and concentrate heat within the degradation unit, a thick asbestos layer was installed atop the electric heating jacket. The system maintained a heating rate of $\sim 50\text{ }^{\circ}\text{C min}^{-1}$. Temperature monitoring was achieved through a thermocouple integrated into the heating assembly, providing continuous verification that operational parameters remained within the prescribed range throughout the reaction duration. After a few hours, the reactor was cooled to room temperature for characterizations. After reaction, the setup was cleaned in a muffle furnace and then washed with aqua regia. The cleaned reactor was rinsed with DI water and stored in a desiccator for next use.

Comment #7: *The degradation temperature of 400 °C is quite low, accordingly, reaction rate is low and the process requires very long reaction times. Why not using higher temperatures?*

Author reply: We thank the reviewer for the valuable comments. Based on our thermogravimetric analysis (TGA, Supplementary Figure 16a), the commercial polypropylene (PP) used in our study begins to lose weight around 350°C, with the maximum rate of weight loss occurring near 400°C. This temperature is ideal for promoting the thermal degradation of PP, while avoiding the excessive formation of light gases such as methane, ethane, or hydrogen at higher temperatures. The reaction rate under 400°C is efficient enough. According to the experiment, the fractionated degradation in the three-fraction reactor took 5 h for complete reaction. Even in the scale-up reactor, the reaction period was 6 h (heating period was excluded). With properly designed mass flow, 3 to 4 shifts of degradation can be completed within a day. Although, higher reaction temperature accelerates the reactions, it also becomes more energy-intensive, along with side reactions. For example, research indicates that higher degradation temperature can accelerate gasification (*Energy Rep.*, **2020**, *6*, 202-207), generating light hydrocarbons (C₂–C₇), with limited applicability in the downstream synthesis of fine chemicals.

In addition, there is an increased possibility of aromatic formation and coking at temperatures > 500°C (*J. Hazard. Mater.*, **2024**, *461*, 132672), as the polymer is more likely to undergo secondary reactions like polymerization and crosslinking (see supporting discussion “*β-Scission, side reactions, and impact of contamination*”).

Therefore, we specifically operated the thermal pyrolysis at 400°C to avoid the excessive side reactions.

Supplementary Figure 16a. Thermal properties of PP.

Comment #8: *The pilot scale process operates in batch regime, moreover, the heating rate is very slow and therefore the time required to complete a reaction is of several ours. This approach is not suitable for the process scale up as long as the efficiency and process throughput are low. Can this process be feasible at full scale? Plastics thermochemical valorization processes should operate in continuous regime to ensure profitability. Please, comment.*

Author reply: We sincerely appreciate the reviewer's constructive comment regarding process efficiency and throughput, which are critical factors for industrial profitability. We wish to clarify that our scale-up experiment was primarily designed to evaluate size selectivity and maximum yield rather than to benchmark the operational efficiency of the fractionated degradation process. As the reviewer rightly notes, the pivotal concerns are whether the process can achieve greater efficiency and whether it could be adapted to continuous operation for improved economic viability.

We should note that our current scale-up reactor was a self-constructed prototype without performance optimization. From chemical engineering perspectives, the operation efficiency is addressable. Encouragingly, our reported data suggest potential for both higher efficiency and profitability at full scale, whether in batch or continuous modes. We would be pleased to discuss these possibilities below.

To be rigorous, we target $M_n < 300$ in the fractionated degradation, the validated upper limit for preparing surfactant.

1. Heating efficiency can be improved significantly with professional industrial equipment. Our self-built 10-kg-scale reactor (**Supplementary Figure 19a and Figure R5**) used barrel-mounted tubular heaters, ensuring temperature precision and uniformity but achieving only ~ 3 °C min^{-1} due to poor insulation and low number of heater density (reduce cost). Industrial systems typically achieve 5-10 °C min^{-1} (slow degradation) or ~ 100 °C min^{-1} (fast degradation, *Fuel*, **2022**, 324, 124777), enabling heating period < 40 min and < 7 h in total. Our designed ton-scale reactor (120 kW, **Figure R2e**) projects ~ 6 °C min^{-1} for 800kg PP (1h to 400°C). The heating efficiency can be further improved by engineering modifications, such as using internal tube heaters (higher

heat surface area and higher heat transfer coefficient, **Table R4**), helical turbulators (enhanced heat transfer), feedstock pre-heaters (reducing heating period).

2. Continuous mode improves the reaction efficiency. Adopting continuous operation is possible for the fractionated degradation. Unlike fractionated degradation in batch mode, which requires time to achieve thermodynamic equilibrium, continuous systems capitalize on dynamic equilibrium. Although we did not experimentally validate the continuous mode in the current study, polymer degradation is similar to petroleum refining practices which is operated contentiously (*Journal of Energy*, **2023**, 2023, 821129). Therefore, it is possible to modify the batch process into continuous process.

3. Profitability does not exclusively depend on reaction mode (continuous versus batch). While continuous operation offers clear efficiency benefits as recommended in EU and Chinese regulatory frameworks, it is worth noting that batch reactions (including semi-batch configurations) have demonstrated successful commercial implementation in the recycling industry. The commercial achievement of Resynergi's modular microwave-assisted plastic degradation illustrates how batch reactor array can attain high efficiency. Similarly, clustered thermal reactors for fractionated degradation present compelling advantages. The compact structure reduces spatial requirements, while integrated clustering enhances both heating efficiency and thermal conservation. **Our conservative techno-economic analysis (Figure 4c, 4d) confirms the economic viability of batch operations, indicating approximately \$14.8 million annual revenue and \$2.1 million net profit. These results remain valid even when applying a 50% discount to product selling prices.**

A discussion about full scale production is updated in the outlook section.

SI, Page 30:

2. From laboratory to industry. The transition from prototype reactors to industrial-scale operations requires significant improvements in efficiency and profitability. First, implementing professional engineering designs, such as internal tube heaters and helical turbulators to improve heat transfer, and feedstock pre-heating systems to shorten heating period.

Second, adopting continuous operation represents a advancement in production efficiency. Unlike fractionated degradation in batch mode, which requires 4-5 h to achieve thermodynamic equilibrium, continuous systems capitalize on dynamic equilibrium conditions. Although not experimentally validated in the current study, the similarity with petroleum refining practices suggests considerable potential for continuous production. This approach would maximize reactor utilization and increasing output capacity.

It should be noted that while continuous processing aligns with regulatory preferences and offers inherent efficiency advantages, commercially successful batch and semi-batch systems remain viable for plastic degradation, exemplified by the Resynergi's microwave-assisted approach. Following the example, clustered

reactors for fractionated degradation may demonstrate advantages in thermal efficiency and energy conservation. Our conservative techno-economic analysis (**Figure 4**) confirms the economic robustness of batch operations, indicating that batch and semi-batch modes are valid alternatives when continuous mode proved unnecessary, impossible, or challenging.

Comment #9: *A weak point of the paper is the lack of comparison with previous literature. The authors should provide a complete comparison of product yields and their composition with those reported in the literature with other plastics pyrolysis technologies and approaches.*

Author reply: We thank the reviewer for the valuable comment. A distribution comparison was given in **Supplementary Figure 7a**. Following the comment, we summarized the reaction information of some representative methods in the literature and tabulated in **Supplementary Table 17** and provided a more thorough outlook section.

Main text, Page 16: The work herein provides a basic reactor framework, quantitative distribution descriptor, and a promising green surfactant for general cleaning purposes. Fractional degradation represents a advantageous methodology for controlling $C\#_n$ and CLD in polyolefin degradation (**Supplementary Table 17**), improving energy efficiency, product selectivity, yields, and sustainability than direct distillation of pyrolysis-oil.

SI, Page 30:

Outlook

1. Comparison with representative methods for distribution controlling. Current approaches to size-selective degradation predominantly dependent on catalytic and chemical engineering pathways. While catalytic hydrogenolysis represents a common methodology (**Supplementary Table 17** and **Supplementary Figure 7a**), which necessitates sophisticated catalyst design and preparation involving costly noble metals like ruthenium and platinum.^{40,53-57} Fundamental limitations including high-pressure operational requirements, poor impurity tolerance, and expensive preparation protocols continue to restrict practical implementation of these systems.^{1,58} The production of alkane-dominated outputs further constrains downstream utility for fine chemical synthesis. Alternative strategies employing engineered reactors enable direct degradation under ambient pressure without hydrogen or catalysts, yielding olefin-rich products from PP and PE that are suited for functional chemical manufacturing.^{1,20} Nevertheless, both catalytic and non-catalytic methods generate hydrocarbons or oligomers with broad carbon distributions, or narrow-distributed light hydrocarbons. The

divergent reactivity between short- and long-chain olefins compromises reaction uniformity in downstream processing.

The fractionated degradation demonstrates promise for advancing this field. By converting PP into narrowly distributed C₆-C₁₅ and C₁₅-C₃₀ α -olefins, this methodology enables precise valorization pathways such as sulfonate surfactant production. The ambient-pressure operation and catalyst-free operation position this approach as a potentially transformative solution for circular plastic economies. More importantly, this catalyst-free platform establishes a robust and scalable foundation for transforming plastic waste into functional hydrocarbons. Beyond surfactants, this technology opens high-value routes to specialty chemicals including petroleum derivatives, advanced lubricants, battery separator materials, and pesticides.

Supplementary Table 17. Representative methods for controlling product distributions.

Ref.	Method	Catalyst/System	Operation conditions	Yield (wt%)	Main Products	Distribution
53	Hydrogen olysis	UiO-66-RuH ₂	200°C, 35 bar H ₂ , 20 h	90	alkanes	C ₅ -C ₁₉
54		mSiO ₂ /Pt-1.7/Si O ₂	300°C, 0.89 Mpa H ₂ , 15 h	73.6	alkanes	C ₉ -C ₃₆
55		mSiO ₂ /Pt/MCM -48	300°C, 2.06 Mpa H ₂ , 6 h	43.6	hydrocarbons	C ₉ -C ₅₂
40		L-ZrO ₂ @mSiO ₂	300°C, 0.992 Mpa H ₂ , 20 h	28	paraffin	C ₈ -C ₃₆
57		Ru-doped ZrO ₂	250 °C, 3Mpa H ₂	71	alkanes	C ₄ -C ₁₇
56	Ball milling	γ -Al ₂ O ₃	ball milling, r.t., ambient pressure	77.2 (wax)	hydrocarbon, alcohols, and ketons	C ₇ -C ₄₅
1	Thermal gradient degradation	w/o catalyst	360°C ambient pressure	90 (wax or liquid)	alkenes	C ₁₂ -C ₆₈₊
20						
This work	Fractionated degradation	w/o catalyst	400°C ambient pressure	85 (liquid)	α -olefins	C ₆ -C ₁₅ , C ₁₅ -C ₃₀

In addition to the above revision suggested by the reviewers, we also made the following important revisions to include more details and improve the quality of our work.

Main Text, Page 2:

Global polypropylene (PP) production rose sharply from ~62 million tons in 2015 to ~80 million tons in 2023 (~20% of total plastics; **Supplementary Figure S1a,b**). Driven by disposable packaging,¹ rapid PP turnover exacerbated plastic waste and GHG emissions (**Supplementary Figure 1c**). The recycling of polyolefins remains challenging due to inefficient logistics, severe contamination in plastic waste (**Figure 2**), and poor profitability, leading to low recycling rate of PP and other plastics.¹⁻⁴ Chemical recycling mitigates environmental impacts and valorizing waste,⁵⁻⁸ converting polyolefins into high-grade fuels,⁹ monomers,^{7,10} aromatics,^{11,12} and functionalized hydrocarbons.^{13,14} Among these, surfactants stand out for their high market volume and value (**Supplementary Figure 2**). Catalytic oxidative degradation, biological fermentation, and tandem strategies, can produce synthetic fatty acids from polyolefins.¹⁵⁻¹⁸ However, consumer preference for natural fatty acids restricts synthetic alternatives from body care, food, and medical markets,¹⁹ which dominate global demand (> 90 \$%).²⁰

Reference

- 2 Regulation (EU) 2025/40 of the European Parliament and of the Council, on packaging and packaging waste, amending Regulation (EU) 2019/1020 and Directive (EU) 2019/904, and repealing Directive 94/62/EC. (European Commission, Brussels, 2025).

Main Text, Page 12:High-temperature HMBC-NMR spectra (50°C) further identified olefinic sulfonates structures (**Figure 3a** and **Supplementary Figure 26**), with minor shifts near δ 5.2 (¹H NMR) indicating other internal alkenyls, originating from side reaction during sulfonation (**Supplementary Scheme 2**).

Main Text, Page 12: Foaming heights in hard water were assessed using the Ross-Miles method (**ASTM D1173–23**; **Figure 3c** and **Supplementary Figure 27**), which were 14.5 and 13.7 cm for PP-T2-S and PP-STW-S, respectively.

Main Text, Page 13: Despite this, the branched structure facilitated wetting and permeation to textures (**Supplementary Figure 28**), and thus improved detergency (**ISO 2267:1986**; **Figure 3d**).

Main Text, Page 13: Traditional branched alkyl sulfonates (BAS), such as tetrapropylbenzene sulfonate, are

resistant to biodegradation due to branched structures inhibiting ω -hydroxylation. Despite their superior detergency, BAS was banned and replaced by biodegradable linear alkyl sulfonates (*e.g.*, US Clean Water Act, 1972). To evaluate the ecological safety of PP-derived surfactants in natural water bodies, biodegradability was assessed using natural sludge in the Gaoguan River at the Qinling National Reserve (sampled at Sep., 2024).

Main Text, Page 14: Therefore, B-P could serve as an efficient and flexible substitute. The economic and ecological performance of B-P was similar to those of B-SD, with 20% higher fixed capital investment and 7% less CO₂ reduction (Supplementary Table 16).

Main Text, Page 16: In summary, this work demonstrated a fractionated degradation, leveraging size-dependent latent heat, for tuning hydrocarbon chain length and CLD during the degradation of polyolefins. Experiment, modeling, and simulation showed a T_r - $C_{\#n}$ correlation, while $F\#$ modulated CLDs through VLE and re-degradation, progressively narrow the distribution per tray. Consequently, narrow α -olefins distribution ($CV < 0.6$) was obtained using a three-fraction reactor, with CLD outperformed those obtained using reactors with lower $F\#$ and in literature. The fractional degradation was further validated by scaling-up to 10 kg using real-life plastic waste, showing product yields, chemical structures, and elemental profile affected by plastic waste purity. The α -olefins derived from virgin or real-life plastic waste were further valorized into sulfonates *via* sulfonation and hydrolysis. The resulting sulfonates exhibited enhanced foaming, detergency, and biodegradability performance comparable to commercial surfactants. Industrial-scale TEA and LCA showed that the upcycling of PP into sulfonates possesses improved sustainability and profitability than the industrial preparation of sulfonate surfactants.

The work herein provides a basic reactor framework, quantitative distribution descriptor, and a promising green surfactant for general cleaning purposes. Fractional degradation represents a advantageous methodology for controlling $C_{\#n}$ and CLD in polyolefin degradation (Supplementary Table 17), improving energy efficiency, product selectivity, yields, and sustainability than direct distillation of pyrolysis-oil. We anticipate the process to be amendable to other plastics, and can be potentially improved by integrating with catalytic processes, tray engineering, and operation optimization, enabling finer manipulation in adjusting chain length and distribution to promote real-life plastic upcycling.

Main Text, Page 19: The authors also acknowledge the Analyses and Testing Center at Shandong University and Shandong Normal University, especially the support by NMR and Spectroscopy Laboratories.

Figure 3. Chemical structure and performance of PP-derived surfactants. (a) HT-HMBC-NMR spectrum of PP-T2-S. (b) Time-lapse images show the stability of PP-T2-S foam from 0 to 16 h. (c) Foaming heights determined by Ross-Miles methods. The error bars represent standard deviations. (d) detergency against sebum and protein stains on standard cotton fabrics. The PP-derived surfactants are compared against commercial surfactants SLS and SDBS. (e) Biodegradation of PP-T1-S, PP-T2-S, and commercial SDBS in natural water collected from Qinling National Reserve.

Supplementary Figure 7. (a) Distributions of PP polymer and hydrocarbon small molecules characterized by calibrated variance (CV), polydispersity index (PDI), standard deviation (σ), and entropy (S) with literature data for β +Pt@Hie-TS-1,⁴¹ L-ZrO₂@mSiO₂,⁴⁰ UiO-66-RuH₂,⁵³ 25-mSiO₂/Pt/SiO₂,⁵⁵ mSiO₂/Pt-5.0/SiO₂,⁵⁶ and γ -Al₂O₃.⁵⁴ Lines are included for visual guidance. (b) Illustrative examples of hydrocarbons and a polymer exhibiting identical PDI values.

Supplementary Figure 26. HT-HMBC-NMR spectra of (a) PP-T1-5h-S and (b) PP-STW-6h-S. The sulfonates were dissolved in D₂O and characterized at 50°C to enhance solubility and signal resolution. Similar to the spectra shown in **Figure 3a**, both spectra exhibit signals corresponding to alkenyl sulfonates. The sulfonation reaction resulted in the formation of new alkenyl groups **other than the β-alkenyl**, attributed to a dehydrogenation reaction induced by SO₃, a potent Lewis acid.³⁴